# Strong synergy between gold nanoparticles and cobalt porphyrin induces highly efficient photocatalytic hydrogen evolution

Huixiang Sheng[1], Jin Wang[1], Juhui Huang[1], Zhuoyao Li[1], Guozhang Ren[1], Linrong Zhang[1], Liuyingzi Yu[1], Mengshuai Zhao[1], Xuehui Li[1], Gongqiang Li [1], Ning Wang [2], Chen Shen [3] & Gang Lu [1,4] ✉

The reaction efficiency of reactants near plasmonic nanostructures can be enhanced significantly because of plasmonic effects. Herein, we propose that the catalytic activity of molecular catalysts near plasmonic nanostructures may also be enhanced dramatically. Based on this proposal, we develop a highly efficient and stable photocatalytic system for the hydrogen evolution reaction (HER) by compositing a molecular catalyst of cobalt porphyrin together with plasmonic gold nanoparticles, around which plasmonic effects of localized electromagnetic field, local heating, and enhanced hot carrier excitation exist. After optimization, the HER rate and turn-over frequency (TOF) reach 3.21 mol g$^{-1}$ h$^{-1}$ and 4650 h$^{-1}$, respectively. In addition, the catalytic system remains stable after 45-hour catalytic cycles, and the system is catalytically stable after being illuminated for two weeks. The enhanced reaction efficiency is attributed to the excitation of localized surface plasmon resonance, particularly plasmon-generated hot carriers. These findings may pave a new and convenient way for developing plasmon-based photocatalysts with high efficiency and stability.

The hydrogen evolution reaction (HER) from the photocatalytic reduction of water is an effective and green approach to produce clean energy, helping to solve the energy crisis and to reduce environmental pollution[1–4]. Developing highly efficient photocatalysts locates in the central of hydrogen evolution. To date, the widely used catalysts for photocatalytic HER include organic molecules and inorganic semiconductors, among which molecular photocatalysts possess the advantage of easy modulation via substitution[5]. For instance, metal porphyrins are widely used in the photocatalytic HER, and their catalytic performances can be easily modulated by changing the core metal and/or the linked functional groups[6–8]. However, molecular photocatalysts suffer from their poor light-absorbing ability and low photostability[7]. Composing these molecular photocatalysts with organic or inorganic composites may improve the light-absorbing and photostable properties.

Localized surface plasmon resonance (LSPR), the collective oscillation of free electrons in metal nanostructures, can result in extraordinary optical properties[9]. Noble metal nanoparticles, such as Au, Ag, and Cu, show LSPR in the visible and/or near infrared spectral regions and a strong light absorption coefficient (~$10^9$ M$^{-1}$ cm$^{-1}$), which is four orders higher than that of traditional photosensitizers[10]. Under light illumination, localized electromagnetic field, local thermal heating, and excited hot carriers can be generated in the vicinity of plasmonic nanostructures[11–13]. Because of these extraordinary properties, plasmonic nanoparticles have been used to trigger and accelerate many reactions, such as the reduction of p-nitrophenol, reduction of

[1]Key Laboratory of Flexible Electronics (KLoFE) and Institute of Advanced Materials (IAM), Nanjing Tech University (NanjingTech), 30 South Puzhu Road, Nanjing 211816, China. [2]School of Physics, University of Electronic Science and Technology of China, Chengdu 610054, P. R. China. [3]Institute of Materials Science, Technical University of Darmstadt, Darmstadt 64287, Germany. [4]National Laboratory of Solid State Microstructures, Nanjing University, Nanjing 210093, China. ✉e-mail: iamglv@njtech.edu.cn

carbon dioxide, and photocatalytic water splitting[14–16]. However, plasmon-generated hot carriers cannot be separated effectively in single-component plasmonic nanostructures, limiting their utilization in chemical reactions[17]. To solve this problem, developing plasmonic composites is becoming mainstream in plasmon-mediated photocatalysis. For example, gold nanoparticles (AuNPs) have been combined with semiconductors, such as g-$C_3N_4$ and $TiO_2$, to further improve the photocatalytic hydrogen evolution reaction[18,19]. Nevertheless, compositing plasmonic nanostructures with inorganic materials is challenging in many cases due to the difficulties in designing and preparing.

As reported, molecules near plasmonic nanostructures become much more active in many chemical reactions due to the improved light absorption, electromagnetic field, local temperature, and hot carrier excitation[20–22]. In addition, the lifetime of plasmon-generated hot carriers can be prolonged obviously at the plasmon-molecule interface[22,23]. What will happen if molecular catalysts are positioned in the vicinity of plasmonic nanostructures? We propose that the catalytic activity of these molecular catalysts can be enhanced dramatically because of plasmonic effects. In this design, the plasmonic nanostructures have a strong light-absorbing ability, and the excited LSPR may activate/promote the molecular catalysts for highly efficient catalytic reactions. Moreover, the generated hot carriers may be utilized in reactions more effectively due to the existence of a plasmon-molecule interface. Note that the preparation of these plasmon-molecule composites is highly simple and convenient, making the composites promising in many practical applications. Thus, the advantages of molecular catalysts can be maintained, and plasmonic effects can help to improve the activity of molecular catalysts.

Herein, a highly efficient HER reaction is realized by mixing plasmonic AuNPs and cobalt porphyrin (5,10,15,20-meso-tetrakis(4-pyridyl)porphyrin, CoTPyP) together. Tuning the morphology and/or aggregation of the AuNPs leads to a superior HER rate of 3.21 mol g$^{-1}$ h$^{-1}$ and a high TOF of 4650 h$^{-1}$ under visible light illumination. In addition, this photocatalytic system is stable after 45-hour catalytic cycles. The catalytic activity and stability of our composite photocatalyst are superior to those of reported state-of-the-art molecular catalysts. The contribution of plasmonic effects to the superior HER efficiency is then discussed comprehensively.

## Results
### Preparation and characterization of the AuNP@CoTPyP nanostructures

Metalloporphyrin catalysts, which have been intensively applied in photocatalytic and electrocatalytic HER, were adsorbed on plasmonic nanostructures to enhance their photocatalytic performance in HER. AuNPs (average diameter is ~15 nm) were used as the plasmonic nanostructures because of their high chemical stability and LSPR in the visible spectrum[12]. CoTPyP molecules, a variant of metalloporphyrin, were used in this work since the pyridine groups can form strong coordination bonds with heavy metals, such as gold[24]. Therefore, the CoTPyP molecules can be easily adsorbed on the surface of AuNPs, forming an organic–inorganic hybrid nanostructure, referred to as AuNP@CoTPyP. Under light illumination, the strong coupling between the plasmonic AuNP and CoTPyP molecules can lead to a high catalytic activity in the HER (Fig. 1a).

The AuNP@CoTPyP nanostructure can be easily prepared by mixing AuNP colloid with CoTPyP solution. As known, there are four pyridine groups that can bond with gold in a CoTPyP molecule. Because of the steric effect and geometry configuration, one CoTPyP molecule may link two AuNPs together to form aggregates (Supplementary Fig. 1). This aggregation was successfully observed by scanning transmission electron microscopy (STEM) imaging (Fig. 1b). The energy-dispersive X-ray spectrum (EDS) mapping (Fig. 1b) of AuNP@CoTPyP further confirms that the distributions of carbon, nitrogen,

and cobalt elements overlaps with that of gold, suggesting that CoTPyP molecules are uniformly adsorbed on the AuNP surface. The aggregation was also confirmed by the ultraviolet–visible (UV–Vis) spectrum (Fig. 1c), in which a very broad peaks and strong background peak at >620 nm appeared due to the coupling mode of LSPR in aggregates. In addition, the UV–Vis spectra at higher CoTPyP concentrations (Supplementary Fig. 2) showed that the peak of CoTPyP redshifted from 424 nm to 430 nm, suggesting that adsorption on AuNPs may slightly shorten the LUMO-HOMO gap of the CoTPyP molecules, which will be discussed later. Moreover, the Raman peaks of CoTPyP molecules shifted slightly after being adsorbed onto the surface of AuNPs (Supplementary Fig. 3), also suggesting an interaction between AuNPs and CoTPyP molecules.

Then, X-ray photoelectron spectroscopy (XPS) measurements were carried out to investigate the interaction between the AuNPs and CoTPyP molecules. The Au $4f_{5/2}$ and $4f_{7/2}$ peaks at 87.3 and 83.6 eV shifted negatively to 87.1 and 83.4 eV, respectively (Fig. 1d), implying successful binding of CoTPyP molecules with AuNPs. In addition, the shape of the N 1 s peak changed obviously after the CoTPyP molecules were adsorbed on AuNPs (Fig. 1e). After deconvolution, it was revealed that the intensity of pyridinic N decreased and that of metal-coordinated pyridinic N increased obviously after the CoTPyP molecules were adsorbed on AuNPs, suggesting that a large amount of pyridinic N in CoTPyP is bonded to AuNPs. The strong interaction between CoTPyP molecules and AuNPs may lead to a huge increase in the efficiency of the photocatalytic HER.

### High catalytic activity and stability of AuNP@CoTPyP

An ultrahigh HER rate of 3.21 mol g$^{-1}$ h$^{-1}$ was achieved on the AuNP@CoTPyP nanostructures. As shown in Fig. 2a, a high HER rate of ~0.71 mol g$^{-1}$ h$^{-1}$ was observed within the first 0.5 h of light illumination with a 300 W Xenon lamp. This HER rate increased obviously to 3.21 mol g$^{-1}$ h$^{-1}$ after 1.5 h of light illumination. This HER activity is tens to hundreds of times higher than the reported state-of-the-art photocatalytic HER rates (Fig. 2b and Supplementary Table 1)[25–39]. The TOF of our system was determined as 4650 h$^{-1}$ by using the amount of CoTPyP as the reference. The ultrahigh HER activity in this work suggests a strong synergy between the AuNPs and CoTPyP molecules, which will be discussed later.

To understand the synergy between the AuNPs and CoTPyP, the HER rates of AuNPs or CoTPyP only were also investigated. With AuNPs only, the HER reaction was hardly observed, indicating that AuNPs are catalytically inert for photocatalytic HER (Fig. 2a). Although plasmonic nanostructures have been reported to be catalytically active in electrocatalytic HER reactions[40], few photocatalytic HER reactions have been demonstrated on AuNPs, possibly due to the difficulty in extracting plasmon-generated hot electrons. With CoTPyP molecules only, the photocatalytic activity was still low. A very low HER rate of ~0.09 mol g$^{-1}$ h$^{-1}$ was observed (Fig. 2a), partially due to the low light-utilization ability of CoTPyP molecules. Therefore, the strong catalytic activity observed in the AuNP@CoTPyP system suggests a strong synergy between the AuNPs and CoTPyP in the photocatalytic HER process.

In addition to the reaction rate, the catalytic stability of the AuNP@CoTPyP hybrid nanostructures was also high. We performed cyclic photocatalysis tests to study the stability of our hybrid photocatalyst. The cleaned AuNP@CoTPyP collected by centrifugation were used for cyclic measurements. It was found that the AuNP@CoTPyP nanostructures can maintain stable catalytic activity after 45 h of cyclic photocatalytic HER tests, which corresponds to a turnover number (TON) of 13950 each cycle (3 h). Besides, the catalytic performance hardly changed (Fig. 2c). The TEM images indicate that the morphology of the AuNP@CoTPyP structures barely changed after 45 h of photocatalytic reaction (Supplementary Fig. 4), confirming a high morphological stability during the photocatalytic reaction. In addition,

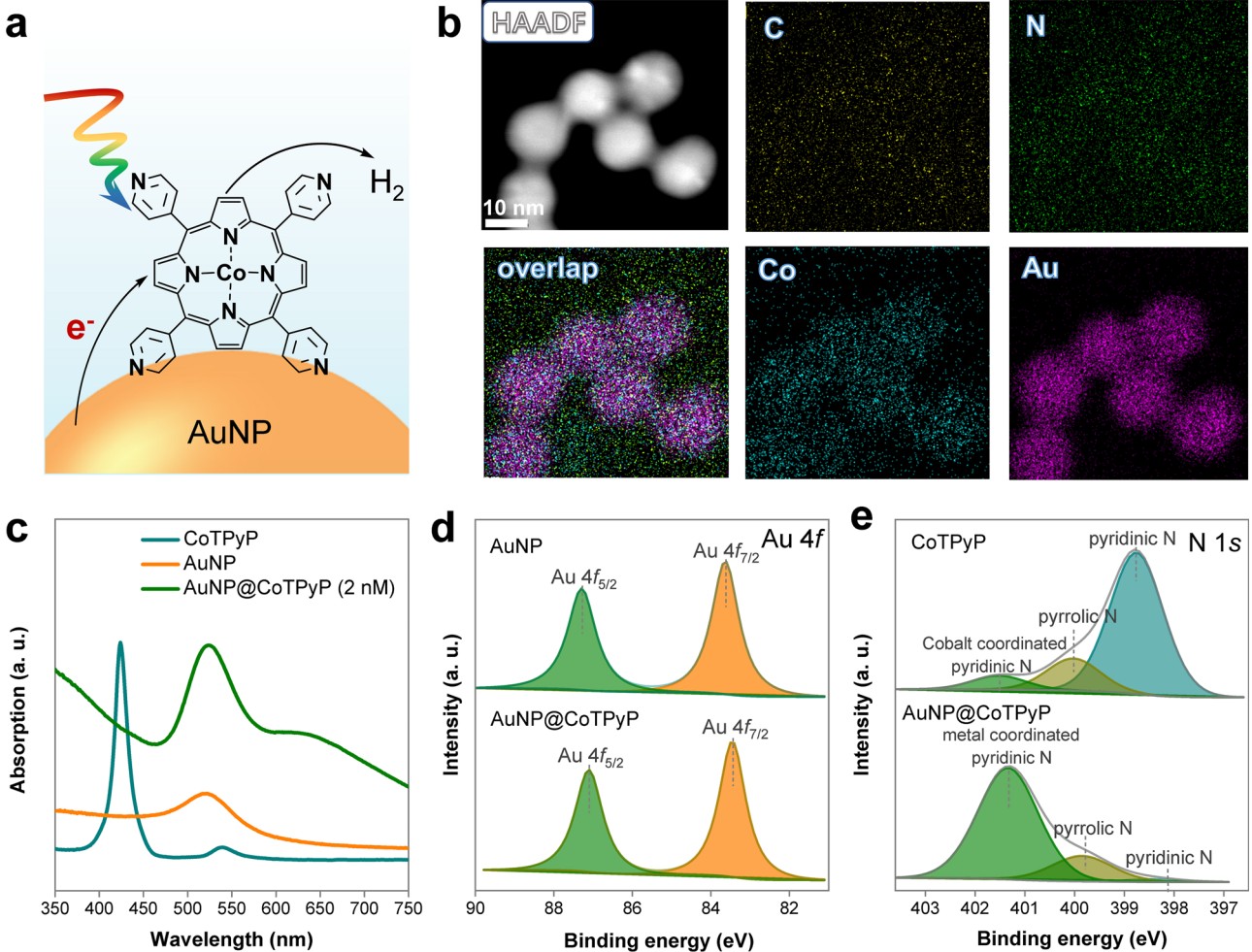

**Fig. 1 | Schematic illustration and characterization of the AuNP@CoTPyP nanostructures. a** Schematic illustration of the enhanced photocatalytic HER in AuNP@CoTPyP. **b** STEM image of AuNP@CoTPyP and corresponding EDS mapping images. **c** UV−Vis extinction spectra of AuNPs, CoTPyP (50 nM) and AuNP@CoT-PyP (CoTPyP concentration = 2 nM). High-resolution XPS (**d**) Au 4 f and (**e**) N 1 s spectra of AuNP@CoTPyP.

the UV−Vis extinction spectrum barely changed after 45 h of reaction (Supplementary Fig. 5), indicating that no obvious further aggregation occurred during the photocatalytic reaction. In addition to morphology, the surface state of the AuNP@CoTPyP nanostructures was also stable during the photocatalytic HER process, since no noticeable change in the XPS spectrum was observed after 45 h of reaction (Supplementary Fig. 6). Furthermore, the catalytic performance of the AuNP@CoTPyP nanostructures was still stable after two weeks of exposure to light illumination (Fig. 2d), suggesting a high photo- and catalytic stability of our hybrid photocatalyst. The stability here is much better than that of traditional organic photocatalysts[3], possibly due to the introduction of photo- and chemically stable AuNPs.

### Reaction rates at different CoTPyP concentrations

Interestingly, we found that the photocatalytic activity of our system is highly dependent on the concentration of CoTPyP molecules. As discussed, the AuNP@CoTPyP system possessed a high HER rate of 3.21 mol g$^{-1}$ h$^{-1}$ at a CoTPyP concentration of 2 nM. At this low CoTPyP concentration, very broad peaks and strong background appeared at >620 nm in UV-Vis spectrum (Fig. 3a and Supplementary Fig. 7a), suggesting a significant aggregation of AuNPs, which was confirmed by TEM image (Supplementary Fig. 7b). This aggregation is caused by the interconnection of AuNPs and CoTPyP molecules, since one CoTPyP molecule can link up to two AuNPs simultaneously. This aggregation leads to the formation of a large amount of gap-mode plasmonic

hotspots[41,42], which may contribute to the enhancement of photocatalytic HER activity. Excitation/activation of the CoTPyP molecular catalysts may be promoted by the excitation of LSPR, resulting in an enhanced photocatalytic HER.

When the concentration of CoTPyP molecules increased to 20 nM, the catalytic activity of the system obviously decreased to 0.14 mol g$^{-1}$ h$^{-1}$ (Fig. 3b, c), which can be explained by the following two reasons. First, the higher concentration of CoTPyP resulted in less serious aggregation of AuNPs, which was confirmed by the UV−Vis spectrum (Fig. 3a) and TEM image (Supplementary Fig. 8). This less aggregation will reduce the amount of formed gap-mode plasmonic hotspots, leading to less enhancement of photocatalytic activity. Second, more CoTPyP molecules are positioned far from the AuNPs because of the increase in CoTPyP concentration; therefore, a smaller proportion of the CoTPyP molecules are activated by the excitation of LSPR. Further increasing the concentration of CoTPyP molecules led to a further decrease in photocatalytic activity (Fig. 3c). When a high CoTPyP concentration of 2 μM was applied, no coupling mode of LSPR was observed in the UV−Vis spectrum (Fig. 3a), suggesting that the AuNPs did not aggregate obviously under this condition. As a result, the catalytic activity decreased significantly to 0.048 mol g$^{-1}$ h$^{-1}$ (Fig. 3b, c), even though this activity is still much higher than that of bare AuNPs or bare CoTPyP molecules. These results double confirm the great contribution of LSPR excitation in the photocatalytic HER.

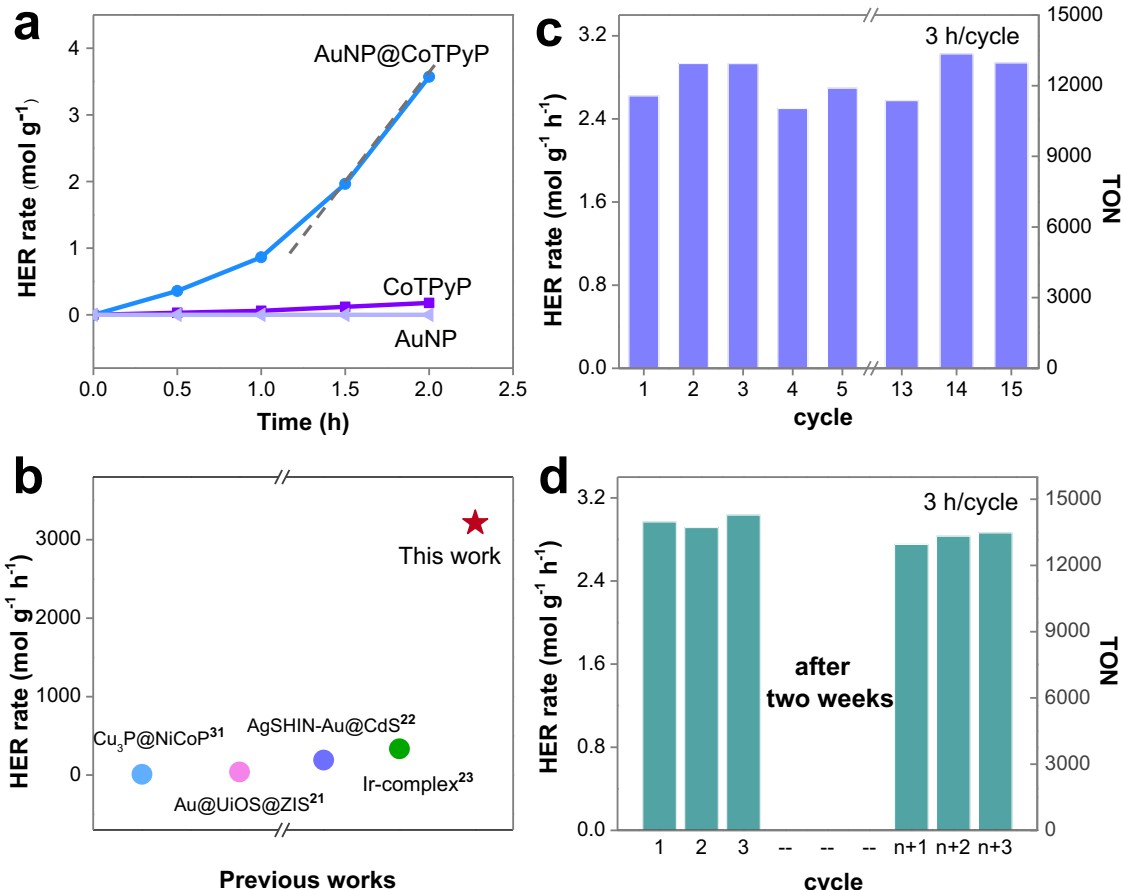

**Fig. 2 | Highly efficient and stable HER of the AuNP@CoTPyP nanostructures.** **a** Photocatalytic HER curves of AuNP, CoTPyP and AuNP@CoTPyP. **b** Photocatalytic HER rates of recently reported photocatalysts. **c** Photocatalytic HER cycles and corresponding TON of AuNP@CoTPyP. **d** Photocatalytic HER activity and corresponding TON of AuNP@CoTPyP after two weeks. Conditions: CoTPyP = 2.0 nM, $CH_3OH$ = 0.5 μM.

To exclude the effect of nonadsorbed catalyst molecules, we also performed the photocatalytic experiments after washing away excess CoTPyP molecules. At the CoTPyP concentration of 2 nM, the amount of produced hydrogen was basically unchanged after washing. While the amount of produced hydrogen slightly decreased after washing at higher CoTPyP concentrations of 20 and 200 nM (Supplementary Fig. 9). These results indicate the great contribution of AuNP aggregation in HER enhancement. The conclusion here is consistent with the cases without washing. Inductively coupled plasma-optical emission spectroscopy (ICP-OES) was used to obtain the accurate Co:Au atomic ratios after washing (Supplementary Table 2) for evaluation of the accurate TON values. According to the ICP-OES analysis, the Co:Au atomic ratio was 1:1600 for the AuNP@CoTPyP structure prepared at the typical CoTPyP concentration of 2 nM. This Co:Au ratio matches perfectly with the one calculated based on the amount of input CoTPyP, since all molecules were bounded on AuNP surface. Therefore, the previously obtained TON values should be accurate.

**Hybrid nanocatalysts based on other plasmonic nanostructures**
The excitation of LSPR is highly dependent on the morphology of plasmonic metals[3,42]. It is feasible to modulate the plasmon-related chemical reactions by tuning the morphology of plasmonic nanostructures[41]. Herein, we replaced the spherical AuNPs with gold nanorods with a length of 100 nm and an aspect ratio of 2:1, which were synthesized by following a reported method[43], and the obtained gold nanorods were highly uniform in shape and size (Supplementary Fig. 10a). The UV–Vis spectrum of the gold nanorods (Supplementary Fig. 10b) showed two plasmonic bands at ~526 and ~670 nm, indicates

that the whole visible spectrum can be effectively utilized by using these gold nanorods. After the adsorption of CoTPyP molecules, the UV–Vis spectrum (Supplementary Fig. 11a) shows that the CoTPyP-induced aggregation of gold nanorods is less significant than that of spherical AuNPs. Moreover, as shown in EDS mapping results, the spatial distributions of Co and N elements were highly consistent with that of Au (Supplementary Fig. 12), suggesting a successful binding of CoTPyP molecules on gold surface in spite of the presence of the cetyltrimethylammonium bromide (CTAB) capping agent in gold nanorod colloid. Compared with that on spherical AuNPs, the HER on gold nanorods showed a slightly decreased rate of 0.2 mol g$^{-1}$ h$^{-1}$ (Supplementary Fig. 11b), possibly due to the smaller amount of gap-mode plasmonic hotspots formed in this case. Silver nanoparticles (AgNPs) can also be applied in this highly efficient photocatalytic HER. The morphology, surface functionalization, and extinction spectra of AgNPs@CoTPyP were shown in Supplementary Fig. 13. A HER rate of 1.45 mol g$^{-1}$ h$^{-1}$ was observed in the AgNP@CoTPyP organic–inorganic hybrid nanostructures (Supplementary Fig. 14). The slightly lower HER rate observed here is possibly attributed to the poorer light absorption in the visible spectrum, different materials, large particle size of AgNPs (~50 nm), and different CoTPyP-induced aggregation, which are not good for catalytic reactions.

**Contribution of LSPR in AuNP@CoTPyP-catalyzed HER**
The above results have already demonstrated a great contribution of LSPR to the high activity of the AuNP@CoTPyP nanostructures in the photocatalytic HER. Then, we further investigated the role of LSPR in the AuNP@CoTPyP-catalyzed HER reaction. LSPR excitation possesses

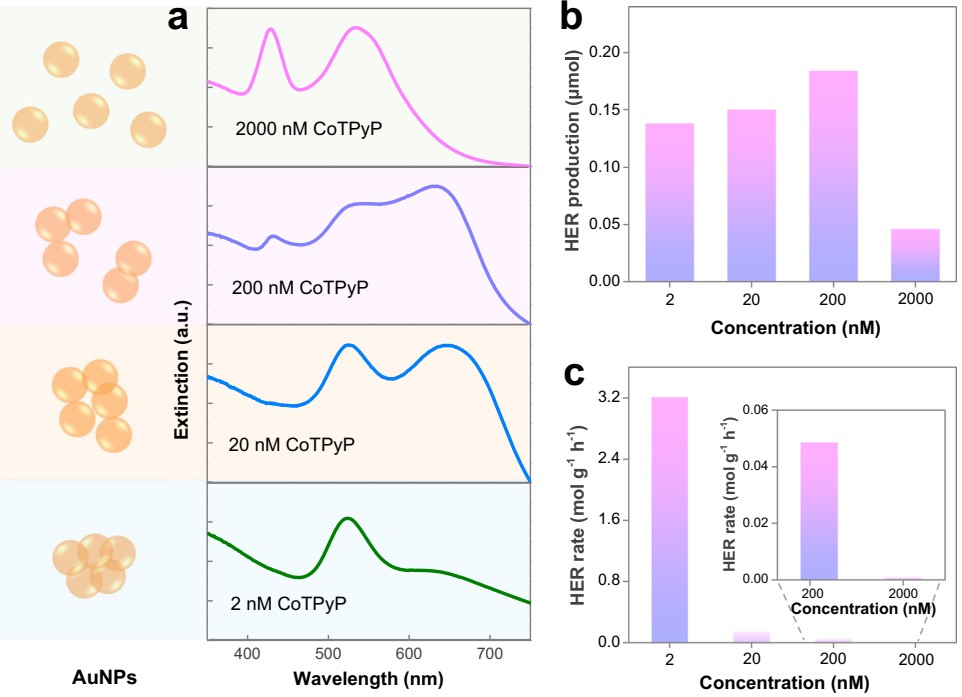

**Fig. 3 | Effect of CoTPyP concentration to the improved HER. a** Schematic illustration of AuNPs and UV−Vis extinction spectra of the AuNP@CoTPyP suspensions at different concentrations of CoTPyP. **b-c** HER production and HER rates at different concentrations of CoTPyP.

high spatial heterogeneity[44]. It is reasonable to investigate the contribution of LSPR by studying the spatial heterogeneity of the reaction around AuNP@CoTPyP nanostructures, which can be investigated directly by using single-molecule fluorescence microscopy (SMFM) (scheme shown in Fig. 4a), an effective tool for catalysis mapping at high spatial resolution[45,46]. Resazurin molecules were used as probes to monitor the generation and distribution of hot electrons, during which resorufin molecules are produced to give bursts of fluorescent intensity. The observed fluorescent bursts indicate the precise location of the generated hot electrons by fitting with a two-dimensional (2D) Gaussian function; thus, the catalysis distribution can be revealed at a high spatial resolution. In our experiment, the AuNP@CoTPyP nanostructures were spin-coated on a piece of cleaned glass slide for catalysis mapping. As observed, the distribution of catalytic sites (Fig. 4b, c) was consistent with that of gold nanostructures, suggesting that the catalytic sites are located mainly in the vicinity of gold nanostructures. To eliminate the possible contribution of plasmon-enhanced fluorescence, we also tried to monitor the catalytic activity of gold nanostructures only, and fluorescent bursts were hardly observed (Supplementary Fig. 15), indicating that the previously observed fluorescent bursts are indeed from the catalytic activity of AuNP@CoTPyP nanostructures. Moreover, this result also proves that AuNPs only cannot catalyze the reaction effectively.

LSPR excitation is highly related to the excitation wavelength. It is necessary to investigate the HER performance under monochromatic light illumination (Fig. 4d). The photocatalytic activity of the AuNP@ CoTPyP system was high under illumination with monochromatic light at 550 nm and 600 nm, which are close to the transverse and longitudinal LSPR peaks of the aggregated AuNPs, respectively. These results suggest that LSPR excitation is crucial in our photocatalytic process. Similar results were also observed in AgNP@CoTPyP (Supplementary Fig. 16), double confirming the great contribution of LSPR excitation in HER enhancement.

Then, we tried to investigate the contribution of plasmonic effects in our photocatalytic HER. First, the finite difference time domain (FDTD) simulation results indicate that the aggregation of AuNPs

effectively increases the electromagnetic field intensity, especially in the nanogap region. Quantitatively, the electromagnetic field enhancement for isolated AuNPs is only 3.8-fold under 650 nm light illumination, in contrast to the enhancement as high as 20.8-fold for AuNP aggregates (Supplementary Fig. 17). The enhanced electromagnetic field may explain the enhancement of the photocatalytic HER. Second, plasmonic heating may also contribute to the enhancement of photocatalytic activity around AuNP@CoTPyP nanostructures. This possible contribution was investigated by continuously measuring the temperature during the photocatalytic HER process. In the case of AuNP@CoTPyP, the temperature increased continuously along with the HER reaction and reached 70 °C after 2 h, which was more serious than the case of CoTPyP only and CoTPyP combined with the traditional photosensitizer Ru(bpy)$_2$ (Fig. 4e, f). To further investigate the contribution of plasmonic heating, we carried out photocatalytic HER at a fixed temperature of 50 °C in the AuNP@CoTPyP and Ru(bpy)$_2$/CoTPyP systems (Light absorption was controlled to be the same in these two systems). As shown, the reaction rate in the AuNP@CoTPyP system was still 4.6-fold higher than that in the Ru(bpy)$_2$/CoTPyP system (Fig. 4f). Therefore, plasmonic heating contributes to the enhanced photocatalytic HER; however, it is not the main reason.

## Interface charge transfer

Many reports have already revealed that plasmon-generated hot carriers participate in many chemical reactions[47,48]. In our case, the plasmon-excited hot electrons may contribute mainly to the enhanced photocatalytic activity of AuNP@CoTPyP nanostructures. As discussed, some pyridinic N in CoTPyP is strongly linked to the gold surface via coordination bond. The unlinked pyridine groups in CoTPyP may ionize to make the molecule positively charged. Therefore, the plasmon-generated hot electrons in AuNPs can easily transfer to the adsorbed CoTPyP molecules to excite/activate them for highly efficient HER. Furthermore, electrochemical impedance spectroscopy (EIS) was performed to investigate the charge transfer at the AuNP-CoTPyP interface. The diameter of the semicircle in an EIS spectrum

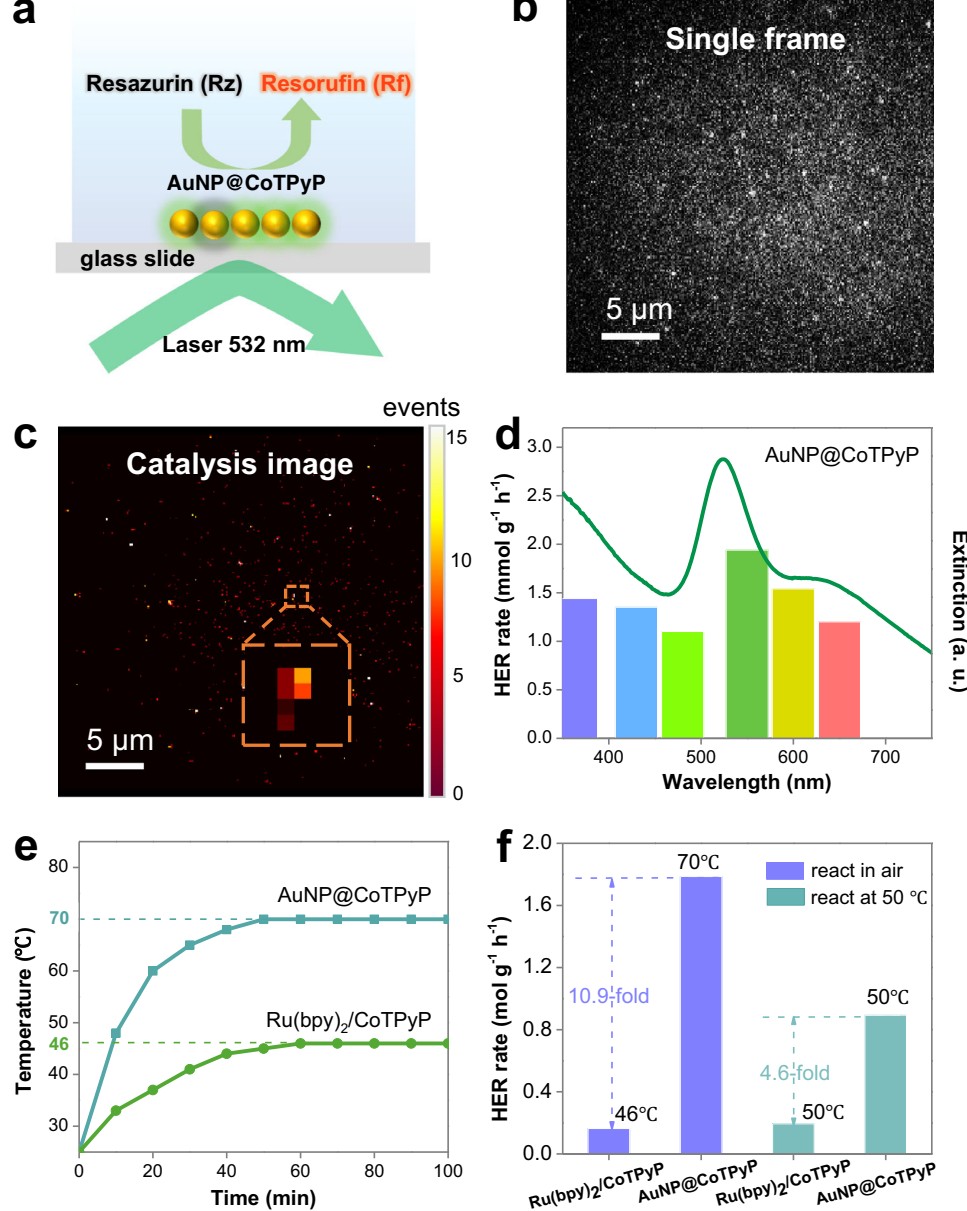

**Fig. 4 | Characteristics of plasmon-enhanced catalysis on AuNP@CoTPyP nanostructures. a** Scheme of the AuNP@CoTPyP-catalyzed resazurin reduction in SMFM. **b** Single frame of the AuNP@CoTPyP nanostructure during SMFM. **c** Reconstructed image of the catalytic active events ($10^4$ frames were acquired within 200 s). **d** UV−Vis extinction spectrum of AuNP@CoTPyP (CoTPyP concentration = 2 nM) and the HER rate under monochromatic light with different individual wavelengths. The power was set as 5.2 W at all wavelengths. **e** Temperature change of Ru(bpy)$_2$/CoTPyP and AuNP@CoTPyP in air. **f** HER rates of Ru(bpy)$_2$/CoTPyP and AuNP@CoTPyP in air and at 50 °C.

indicates the charge transfer resistance ($R_{ct}$), and a smaller diameter implies a favored charge transfer[49]. The model of Randles equivalent circuit (inset in Fig. 5a) was used to analyze the charge transfer at interface[50]. $R_s$ and $R_{ct}$ are the solution and charge transfer resistances, $C_W$ is the Warburg impedance, and $C_{DL}$ is the double-layer capacitance. In our case, the sample of AuNP@CoTPyP showed a much smaller charge transfer resistance than that of the CoTPyP sample (Fig. 5a), suggesting a favored charge transfer and separation at the AuNP-CoTPyP interface. This favored separation of hot carriers results in an improved photocatalytic HER performance. In the photocurrent response spectra, photocurrent of the prepared AuNP@CoTPyP under illumination was clearly larger than that of the bare AuNPs, suggesting an improved charge transfer at the interfaces (Fig. 5b). The charge flow from AuNPs to adsorbed CoTPyP molecules under illumination could be determined based on the configuration of the measurement setup.

Then, we further studied the charge transfer dynamics in AuNP@CoTPyP by using transient absorption spectroscopy. First, we tested the transient absorption spectra of the AuNPs and CoTPyP molecules as controls. As shown in Fig. 5c, a negative peak with two positive wings appeared at ~520 nm in the sample of AuNP colloid, which is attributed to the plasmonic band of AuNPs[51]. In the transient absorption spectrum of the CoTPyP solution, a weak bleaching peak around at ~537 nm (Fig. 5d), corresponding to ground state absorption of CoTPyP molecules, matched perfectly with the UV−Vis absorption peak of CoTPyP molecules (Fig. 1c). Note that the peak related to the main absorption peak was missing in transient absorption spectra, because it partially overlaps with the pump wavelength (430 nm). A broad and positive absorption band also appeared within the range of 560−720 nm, possibly from the light absorption of a new species. To verify this species, the spectroelectrochemical experiments were

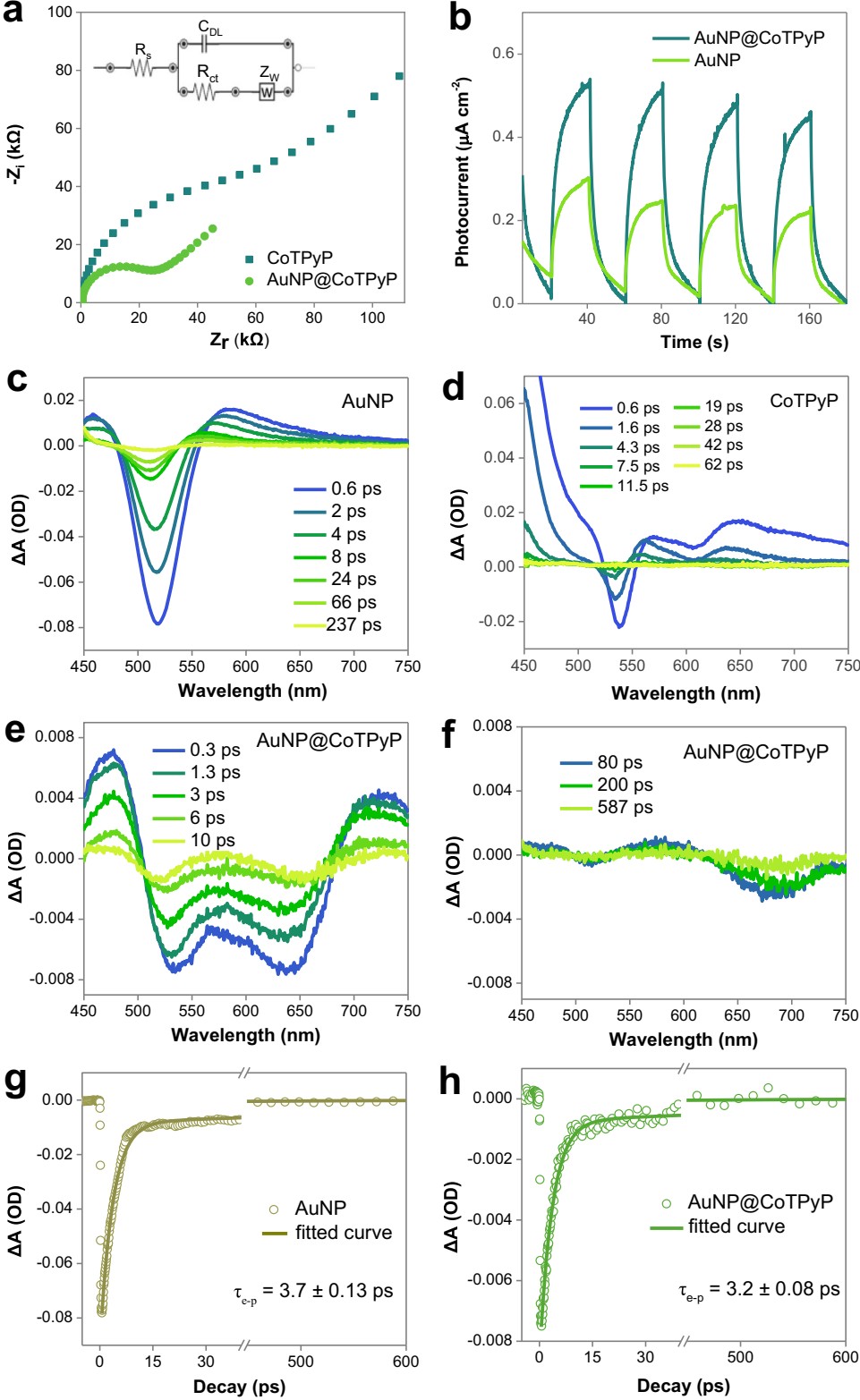

**Fig. 5 | Contribution of the enhanced hot carrier transfer in AuNP@CoTPyP photocatalysis. a** Nyquist plots of CoTPyP and AuNP@CoTPyP in $H_2SO_4$ solution (pH = 4). **b** Photocurrent measurements of the AuNPs and AuNP@CoTPyP (CoTPyP concentration = 2 nM). The samples were periodically illuminated with green light (550 ± 25 nm filter was applied to Xenon lamp). **c** Ultrafast transient absorption spectra of the AuNP excited by a 430 nm pump beam (pulse density = 17 µJ·cm$^{-2}$). **d** Ultrafast transient absorption spectra of the CoTPyP molecules (20 µM) excited

by a 430 nm pump beam (pulse density = 90 µJ·cm$^{-2}$). **e–f** Ultrafast transient absorption spectra of AuNP@CoTPyP (CoTPyP concentration = 2 nM) excited by a 430 nm pump beam (pulse density = 25 µJ·cm$^{-2}$). All the transient absorption experiments were performed in water solvent added with 5% methanol. **g–h** Transient absorption decay curves and corresponding fitting of AuNP (at 520 nm) and AuNP@CoTPyP (at 530 nm), respectively.

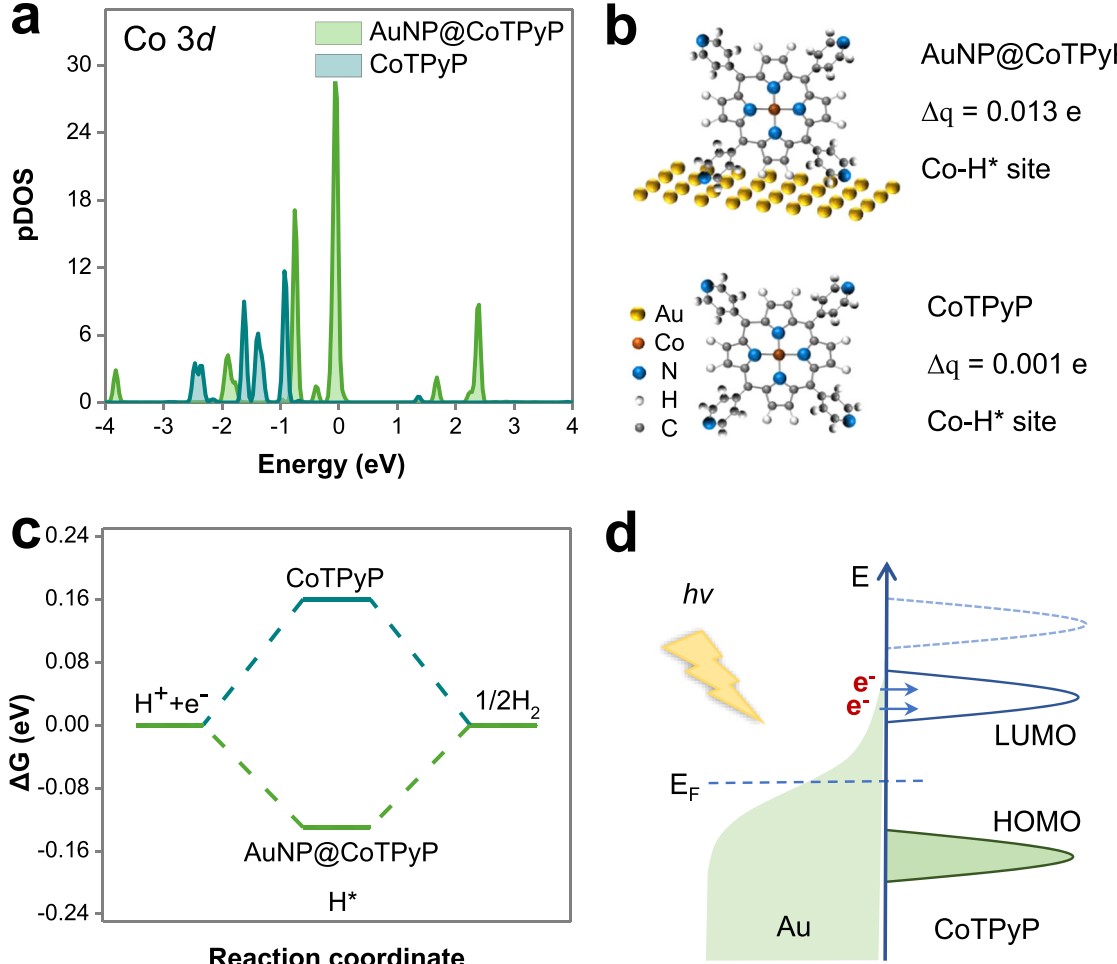

**Fig. 6 | Theoretical DFT calculations of the AuNP@CoTPyP system. a** Partial density of states (pDOS) of CoTPyP and AuNP@CoTPyP. **b** Differential charge densities of H* at CoTPyP and AuNP@CoTPyP. **c** Gibbs free energy of H* absorption on different catalyst sites. **d** Schematic illustration of the charge transfer processes in AuNP@CoTPyP.

performed in $N_2$ atmosphere. The transient absorption spectra of CoTPyP matched well with the shape of the UV-Vis differential absorption spectra of the reduced CoTPyP and were different from that of the oxidized CoTPyP (Supplementary Fig. 18), suggesting that the reductive quenching pathway should be a dominant process and the new species may be the reduced state of CoTPyP[36,52]. In contrast, when AuNPs were adsorbed with CoTPyP molecules, except for the plasmonic peak of AuNPs at ~530 nm, a new negative peak appeared at ~640 nm (Fig. 5e), which could be attributed to the bleaching of the plasmonic band of aggregated AuNPs, consistent with the UV–Vis spectrum of AuNP@CoTPyP nanostructures (Fig. 3a). Meanwhile, a new negative peak appeared at ~670 nm from 80 ps (Fig. 5f) and gradually shifted to ~705 nm from 80 to 1500 ps (Fig. 5f and Supplementary Fig. 19). This peak could be attributed to the stimulated emission from the CoTPyP molecules. Note that two peaks showed up at ~665 and ~710 nm in the photoluminescence spectrum of CoTPyP (Supplementary Fig. 20). Due to the different lifetimes of these two photoluminescence events, they showed up in the transient spectra at different time scales, well explaining the observed features in 630-730 nm region.

To further study the interaction between the AuNPs and CoTPyP molecules, we plotted the decay kinetics of AuNPs and AuNP@CoTPyP (Fig. 5g, h). During the decay of LSPR, hot carriers are formed and then consumed via e-p scattering and chemical reaction[22]. Therefore, a decrease in lifetime of hot carriers usually suggests an inhabitation of radiative decay and a favored chemical reaction. In our case, by fitting

the decay curves to a two-term exponential model[53], it is revealed that the lifetime of plasmon-generated hot carriers was $3.7 \pm 0.13$ ps in the naked AuNP sample, and this lifetime decreased to $3.2 \pm 0.08$ ps when CoTPyP molecules were adsorbed. Before CoTPyP adsorption, the lifetime is mainly affected by the e-p scattering which consumes hot carriers. After CoTPyP adsorption, the plasmon-generated hot carriers can transfer to the adsorbed CoTPyP molecules for catalytic reactions, during which hot carriers are consumed. Therefore, the radiative decay pathway is inhibited and thus the HER rate is increased.

## DFT calculation

The AuNP@CoTPyP system was also studied theoretically via density functional theory (DFT) calculations. First, the partial density of state (pDOS) was calculated to explore the effect of AuNPs on the electronic structure of CoTPyP molecules (Fig. 6a). The center of the Co 3d orbital in AuNP@CoTPyP shifted toward the Fermi level compared with that in the CoTPyP molecule, indicating that AuNPs favor the excitation of the CoTPyP molecule. In addition, the differential charge densities at the H* site were calculated in both AuNP@CoTPyP and CoTPyP (Fig. 6b). The charge transfer value from the Co atom to H* is only 0.001 e in the bare CoTPyP molecule. In strong contrast, this charge-transfer value increases significantly to 0.013 e in AuNP@CoTPyP. The increase in the charge transfer value here suggests that it is much easier for the electron to transfer from the Co center to H*, helping to produce $H_2$ molecules, which is in agreement with the observed results. Moreover, Gibbs free energies were also calculated to investigate the

contribution of AuNPs to the CoTPyP-catalyzed HER reaction. In bare CoTPyP, the Gibbs free energy for H* adsorption is 0.16 eV, which decreased to −0.13 eV in AuNP@CoTPyP (Fig. 6c), favoring the HER reaction. Thus, the HER rate on AuNP@CoTPyP is higher than that on the bare CoTPyP molecule. In addition, adsorbing CoTPyP molecules to the AuNP surface can also lead to a change in the HOMO and LUMO levels, resulting in a reduction in the HOMO-LUMO gap from 3.24 eV to 3.22 eV (Supplementary Table 3), which is consistent with the UV−Vis absorption result (Supplementary Fig. 1). The above results and calculations indicate that plasmon-generated hot carriers can transfer effectively to the LUMO of CoTPyP molecules (Fig. 6d), and thus, the excited CoTPyP molecules can lead to a more favorable HER reaction. Therefore, the AuNP@CoTPyP system can work as a highly effective photocatalyst for the HER.

## Discussion

In summary, a highly efficient and stable photocatalyst for the HER was developed by combining CoTPyP with AuNPs. The high HER efficiency in the AuNP@CoTPyP system is attributed to the strong synergy between AuNPs and CoTPyP molecules. It has been experimentally and theoretically revealed that the lifetime of plasmon-generated hot carriers is prolonged at the AuNP-CoTPyP interface, and the transferred hot carriers to the LUMO of CoTPyP molecules favor catalytic HER. This research provides a brand-new approach, which is highly simple, effective and cost-efficient, for the design and preparation of highly efficient hybrid nanocatalysts. This method may also be extended to preparing many other photocatalytic systems for various reactions, including the reduction of carbon dioxide and fixation of nitrogen.

## Methods

### Chemicals

Sodium citrate ($Na_3Ct$, 99.5%) and ascorbic acid (AA) was purchased from Aladdin Pte Ltd. Gold chloride trihydrate ($HAuCl_4·3H_2O$) was purchased from Shanghai Macklin Biochemical Co., Ltd. Cobalt (II) acetate tetrahydrate ($C_4H_6CoO_4·4H_2O$) and 5,10,15,20-tetra-pyridylporphyrine (TPyP) were purchased from J&K Scientific Ltd. N,N-dimethylformamide (DMF, ≥ 99.5%) and methanol ($CH_3OH$, AR) were purchased from China National Pharmaceutical Group (Shanghai, China). All chemicals were used without further purification, and ultrapure water (18.2 MΩ·cm at 25 °C, MZY-U10V, Miaozhiyi) was used in all experiments.

### Synthesis of AuNPs

The AuNPs were synthesized by following the procedures in previous report[54,55]. First, the mixture of 1 mL $Na_3Ct$ solution (1 wt% in water) and 20 mL water were heated to boil in a cleaned flask. Then, 0.4 mL $HAuCl_4$ solution (1 wt% in water) was quickly injected and this mixture was boiled for another 20 min. The obtained gold sol was naturally cooled to room temperature.

### Synthesis of AgNPs

The AgNPs were synthesized by following the procedures in previous report[20]. First, 3 mL $AgNO_3$ solution (10 mM) was quickly injected into 30 mL of boiling water stirred at 650 rpm. After boiling the solution again, 1.2 mL $Na_3Ct$ solution (1 wt %) was injected. The mixture was naturally cooled to room temperature after 1 h reaction under boiling. All above process was kept in dark.

### Synthesis of AuNRs

The AuNRs were synthesized by following the procedures in previous report[43]. First, 0.03 mL of fresh $Na_4BH_4$ solution (0.1 mM) was quickly injected to the mix solution of 0.25 mL of 0.5 mM $HAuCl_4$ and 0.25 mL of 0.2 M CTAB solution under vigorous stirring (1200 rpm). After 2 min, the solution turned from yellow to brownish yellow, obtained the seed solution, which was aged at room temperature for 30 min

before usage. To prepare the growth solution, 0.09 g CTAB and 0.015 g NaOL were first dissolved in 2.5 mL of warm water (~50 °C) in a 20 mL glass tube. After cooling down the above solution to 30 °C, 40 μL $AgNO_3$ (4 mM) and 2.5 mL $HAuCl_4$ (1 mM) solutions were added. The mixture became colorless after 90 min of stirring at 700 rpm, and then 10 μL HCl solution (12 M) and 39 μL AA (0.064 M) solution were injected consequently under vigorous stirring for 30 s. To prepare the AuNRs, 0.005 mL seed solution was injected into the growth solution, and the resultant mixture was stirred for 30 s and left undisturbed at 30 °C for 12 h.

### Synthesis of CoTPyP

The CoTPyP was synthesized by following a previous report[56]. Briefly, 220 mg (0.36 mmol) TPyP and 360 mg (1.4 mmol) Co(ac)$_2$ were both dissolved in 20 mL DMF, and the above mixture was refluxed for 5 h. Then, the solid product of CoTPyP was precipitated by adding cold water and keeping the above solution in an ice bath. The obtained solid was filtered and washed with water three times, after which the product was dried under vacuum. The UV−Vis spectra (Supplementary Fig. 22) showed the typical Soret band red-shifted by 8 nm to 425 nm and the typical Q band at 537 nm showed up after reaction, suggesting a successful metalation.

### Synthesis of AuNP@CoTPyP

CoTPyP powder was dissolved in 1 mL of 0.1 M hydrochloric acid to obtain a final concentration of 2 nM. Then, 140 μL of the prepared CoTPyP solution was rapidly injected in to a bottle containing 5 mL gold colloid (0.488 mM) and 15 mL water under magnetic stirring at 300 rpm, and this solution was stirred for additional 2 min. The pH value of the final solution was 3.9.

### Photocatalytic hydrogen evolution

The reaction system was prepared in a 40 mL reactor by rapidly mixing above solution and 500 μL methanol at 300 rpm. Then, photocatalytic hydrogen generation experiments were carried out under illumination of a 300 W Xenon lamp mounted with a long pass filter (λ ≥ 400 nm, UV400CUT), and the gas analysis was carried out every 0.5 h on an offline gas chromatograph (GC-9860 5CNJ, Nanjing Hope Analytical Equipment Co., Ltd). In the stability test, 100 μM polyvinyl pyrrolidone (PVP) was added to the cleaned AuNP@CoTPyP system to prevent the unexpected aggregation of AuNPs during long-term photocatalysis. After each cycle, the system was degassed and left in dark for one hour. In some cases, the temperature of the reaction system was controlled by using a constant-temperature water bath and the reaction suspension was stirred continuously.

### Characterization of AuNP@CoTPyP

The absorption spectra were obtained on a UV−Vis spectrometer (Lambda 950, PerkinElmer). X-ray photoelectron spectroscopy (XPS) measurements of the samples were carried out on a photoelectron spectrometer (ESCALAB 250XI, Thermo). The XPS spectrum was corrected according to the binding energy of C 1 s at 284.8 eV. The morphology and EDS mapping of AuNP@CoTPyP were characterized by transmission electron microscopy (TEM, 2100 Plus, JEOL). Electrochemical impedance spectroscopy (EIS) was performed on a portable electrochemical workstation (Plamsens4, PlamSens BV) in a three-electrode system. FTO slide (15×5×2.2 mm³), platinum mesh, and Ag/AgCl electrode were used as the working, counter, and reference electrodes. The 10 μL AuNP@CoTPyP was drop-casted on the FTO slide to serve as the working electrode. The measurement was carried out in dilute sulfuric acid (pH = 4). The frequency during the measurement ranged from 0.01 to 100 000 Hz, and the sample was illuminated with a 300 W Xenon lamp (Perfectlight, Beijing, China). The photocurrent test was performed on an electrochemical analyzer (CHI 630E, CH Instrument) in a standard three-electrode electrochemical

cell filled with 1 M PBS buffer. Carbon paper (1×1.5 cm$^2$) was used as the working electrode, and 5 mL of AuNP (0.488 mM) was drop-casted, followed by thermal annealing at 80 °C. The annealed sample was then soaked in CoTPyP solution (typically 2 nM) for ten minutes. A piece of Pt plate served as the counter electrode, while Ag/AgCl served as the reference electrode. A green light beam was used for sample excitation by applying a 550 ± 25 nm filter to a Xenon lamp.

## Catalysis imaging of the AuNP@CoTPyP film

The as-prepared AuNP@CoTPyP was spin-cast on a piece of pre-cleaned cover glass. The above sample coated on cover glass was sealed with a homemade flow cell made of PDMS elastomer. The hydrogen-saturated resazurin (50 nM) solution in phosphate buffer (pH = 7.2, 0.1 M) was used as the fluidic flow during the SMFM measurements, and the flow rate was controlled as 20 μL min$^{-1}$ via a syringe pump (TYD01-01, LeadFluid, China). The AuNP sample was prepared with a similar procedure as a control. The sample sealed with a flow cell was then mounted on an inverted optical microscope (Ti2-U, Nikon) in TIR configuration and illuminated with an ~20 mW circularly polarized continuous-wave 532 nm laser beam (MGL-III-532, CNI, China) through a 100× oil-immersion objective (NA = 1.49, Nikon). The fluorescence signals were collected by the same objective, and fluorescent images were recorded with an electron-multiplying charge-coupled device (EMCCD, Ultra 897, Andor) camera operated at an acquisition speed of 20 frames per second (fps) and an EM gain of 500.

## Data analysis of single-molecule results

The image sequences acquired by the EMCCD camera were analyzed with home-written scripts. The fluorescent bursts representing single turnover catalytic reactions were isolated from the surrounding background and then fitted with a 2D Gaussian function. The fitting-obtained coordinates were then used for reconstructing a single image to show the catalysis mapping. The details have been described in our previous reports[57].

## FDTD simulation

In FDTD simulations, the diameter of the AuNP was set as 15 nm, and two AuNPs were connected to simulate aggregation to minimize calculation resources. The length and diameter of the Au NRs were 50 nm and 25 nm, respectively.

## Measurement of transient absorption spectroscopy

The ultrafast laser used in our experiment was generated by a Ti:sapphire ultrafast laser (Astrella 800 nm, Coherent) at a repetition rate of 1 kHz. The pulse width was ~100 fs, and the pulse energy was typically ~7 mJ·pulse$^{-1}$. The laser beam was split into two beams. One of the beams was sent into an optical parametric amplifier (OPerA Solo, Coherent) to generate a pump light centered at 400 nm, and the other beam was focused on a sapphire crystal to obtain a white-light continuum (450–750 nm) as a probe light to measure the absorption change of the samples. The delay between the pump and probe beam was varied by a mechanical delay line at a minimum step of 14 fs. The sample was placed in a quartz colorimetric dish with a thickness of 1 mm. The pump and probe beams were both focused onto the sample solution, and the measurements were finished under ambient conditions.

## Theoretical calculation

DFT calculations were performed using a Vienna Ab initio simulation package (VASP), and the exchange-correlation potential was described using the generalized gradient approximation of Perdew-Burke-Ernzerhof. The projector augmented-wave (PAW) method was used to analyze the interactions between ion cores and valence electrons, and a plane-wave cutoff energy of 500 eV was applied. The structural models were relaxed until the Hellmann−Feynman forces were smaller than −0.02 eV/Å and the energy change was smaller than $10^{-5}$ eV. The Brillouin zone was represented by a Γ centered k-point grid of 1×1×1 during the relaxation. Spin polarization was considered in all calculations. The DFT-D3 empirical correction method was used to describe van der Waals interactions. The lengths of the x-axis and y-axes of the lattice were 20.18 Å and 19.97 Å, respectively. The periodic image interaction between two nearest neighbor unit cells was avoided by setting the vacuum to be 30 Å in the z-direction. The sub-monolayer CoTPyP molecule was adsorbed on Au(111) surface to obtain a surface coverage of ~10%. The free energy (ΔG) in each elementary step was calculated based on the standard hydrogen electrode model determined as:

$$\triangle G = \triangle E + \triangle E_{ZPE} - T\triangle S - neU \qquad (1)$$

where ΔE and ΔS are the changes in reaction energy and entropy, respectively; $\triangle E_{ZPE}$ is the difference in zero point energy between the adsorbed and gas phase molecules; U is the applied bias; and n is the electron transfer number involved in the reaction. U = 0 V for free energies was used in the diagrams demonstrated in this paper. For the frequency calculation, we considered the adsorbate *H, 2H*, and the Co atoms where intermediates are adsorbed. We calculated the ΔG for both the Volmer−Heyrovsky and Volmer−Tafel pathways. It was obtained that the rate-determining step (RDS) of Heyrovsky process possesses a lower reaction energy than that of Volmer-Tafel process (Supplementary Fig. 21). Therefore, the Volmer−Heyrovsky mechanism is the main pathway of HER in our system.

## Data availability

The data generated in this study are provided within the manuscript and Supplementary Information file. Any additional information needed is available from the corresponding author upon request. Source data are provided with this paper.

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

## Acknowledgements

This work was supported by the National Natural Science Foundation of China 11974180 (to G.L.), the Six Talent Peaks Project in Jiangsu Province XCL – 038 (to G.L.), and the Postgraduate Research & Practice Innovation Program of Jiangsu Province KYCX21_1095 (to Z.L.). DFT computations for this research were undertaken with the assistance of Phadcalc (www.phadcalc.com) resources. The authors would like to thank shiyanjia lab (www.shiyanjia.com) for the support of HRTEM test.

## Author contributions

H.S. and G.L. conceived and designed this project. H.S. and J.H. performed the experiments. H.S., Z.L., G.R. and L.Z. analysed the data. J.W. performed the SEM measurements. N.W. and C.S. carried out the DFT calculations. L.Y., M.Z., X.L. and G.Li interpret the final data of the revised manuscript. H.S. and G.L. wrote and revised the article with the help from the other authors.

## Competing interests

The authors declare no competing interests.
