## [Peer review file · Nature Communications]

REVIEWER COMMENTS

Reviewer #1 (Remarks to the Author):

The manuscript by Gang Lu and co-workers reports a hybrid photocatalyst for HER based-on Au-nanoparticles and a Co-porphyrin catalyst that operates through localized surface plasmon resonance. The authors demonstrate that under visible light irradiation, plasmon generated hot carriers (electrons) can be transferred from Au-NP to the molecular catalyst, CoTPyP, enables the photocatalytic HER in aqueous medium. The paper is an important contribution in the growing field of plasmon driven photocatalysts for sustainable transformations. The photocatalytic HER studies are well complemented by experimental and theoretical investigations including SMFM analysis, temperature dependence of the catalysis, TAS measurements, and DFT. However, there are few things that need to be addressed before the manuscript can be considered for publication in Nature Communications.

- 1) It is unclear in the introduction whether there are any earlier works on combining plasmonic metal nanoparticles with molecular catalysts. If not, the authors should highlight this as it presents an important finding in this area.
- 2) What was rationale for using pyridine anchors for anchoring Co-porphyrins on Au-NP? Wouldn't thiols be more suitable for immobilisation of Au surface?
- 3) Page 5, line 104: one CoTPyP molecular can link up to four AuNPs, not just two. However, the number of AuNPs in the aggregate might be limited by the size of porphyrin molecule and dimension of AuNPs. The authors should check if four AuNPs can be accommodated around a porphyrin molecule.
- 4) Page 6: the UV-vis of AuNP@CoTPyP shows an intense peak at ~550 nm. What is the origin of this band?
- 5) N1s XPS spectra of both CoTPyP and AuNP@CoTPyP show a peak assigned to graphitic N and the authors claim that the increase in this peak (with concomitant disappearance of pyridinic N) indicate anchoring of CoTPyP on Au surface. However, I am not quite sure why they are termed as 'graphitic N'. Do the authors mean metal coordinated pyrrolic N in CoTPyP? This should be clarified.
- 6) The ratio of molecular catalyst and AuNPs used in photocatalysis experiments should be clearly discussed. The rate of HER clearly depends on the catalyst loading, which is rationalized based on formation of aggregates. However, the size of aggregates will likely be controlled by the ratio of CoTPyP molecules and number of AuNPs in the mixture. Highly dilute CoTPyP condition will favour 1:4 (Co:AuNP) aggregates, while high CoTPyP concentration can lead to 1:3, 1:2, 1:1 aggregates. This neither clearly described in results section nor in the experimental methods.
- 7) Page 15, line 289: the authors claim that electron transfer from AuNP to CoTPyP is facilitated by the positive charge of the CoTPyP molecule. This requires further clarification. If we assume that all four pyridines are coordinated to AuNPs (1:4 ratio), then they pyridines cannot be protonated to provide positively charged groups. In that case, the above statement is incorrect and there won't be any additional contributing factor due to charge of the molecule.

8) Can the authors clarify the results shown in Figure 5f? The discussion on this figure is unclear, in particular, the authors describe 'a new positive peak also appeared at 630-730 nm' which isn't clear in the figure.

9) Can the authors clarify how the time scale of e-p coupling is related to the lifetime of the plasmon-generated hot carriers? The time scale gets shorter when CoTPyP is immobilized on AuNP. How does that translate into slower dissipation of the energy?

10) Experimental methods: (i) further characterisation (e.g., elemental analysis) of CoTPyP is recommended; otherwise, original publication on CoTPyP should be cited. (ii) details of the EIS analysis (electrode preparation, circuit used to fit data) should be included. (iii) detailed preparation of AuNP@CoTPyP material for characterization should be included.

Reviewer #2 (Remarks to the Author):

The manuscript submitted by Gang Lu and colleagues sure reports on an interesting approach for HER and the system shows amazing performance and stability. Nevertheless, especially in the data analyzing the mechanism in this system, the data does not well support the claims in the current state and a major revision is needed before this manuscript can be considered for publication in nature communications.

In detail I have the following comments, corrections and questions:

Page 6 XPS X-ray photoelectron spectroscopy not photoluminescence spectroscopy

Figure 1: could the authors also show the absorption spectrum of just the AuNP. Simply add it to panel c. This would better illustrate the appearance of the aggregation induced resonance.

Figure 3a: The authors argue in the text at page 10 that increasing the CoTPyP concentration leads to less aggregation of the AuNPs and refer to the changes in the absorption spectra. The band at 620 nm was assigned to be an aggregation induced resonance. If the aggregation is highest at 2 nM of the porphyrin, shouldn't this aggregation induced feature be strongest at this concentration and decrease with increasing concentration of the porphyrin? Can the authors explain why this feature in Figure 3 panel a is increasing first with increasing concentration and then suddenly seems to disappear at the highest concentration?

Caption Figure 4 f does not match the panel. Please check carefully.

Does the binding of the porphyrin to the Au particles change the molecular structure? Have the authors tried to collect SERS spectra to observe changes in vibrational modes upon binding?

The rates are given in mol/g h. What is the mass which is used as reference here? Could the authors also give TONs and TOFs. These values also are very often used to evaluate HER and it would be helpful to have here also an idea of the order of magnitude to compare with other systems reported in literature.

The authors compare the rates for Au and Ag particles and argue the lower activity of the Ag system to be caused by lower light absorption in the visible spectrum by the Ag particles. I assume that these experiments which are compared here were performed under broadband illumination? Later the authors describe HER evolution experiments with defined irradiation wavelengths. Was such an experiment also performed for Ag? How does Au and Ag compare upon irradiation of Ag and Au in resonance with the respective LSPR?

How was the temperature controlled in the plasmonic system. How can the authors make sure, that there is no local heating event under the "controlled" temperature conditions?

The authors claim, that besides heating there might be a transfer of hot electrons from the Au particles to the porphyrin reaction center. Hence the Au particles kind of act as sensitizer if I understood correctly. To trace the interfacial charge transfer transient absorption EIS was performed and these results indicate a decreased resistance in this systems hinting to a charge transfer event, also PL spectroscopy has been performed. Here is not entirely clear how the PL of the excited porphyrin is quenched by a hot electron transfer event. Can the authors please elaborate on the PL experiment a bit more in detail. Also to really evaluate emission intensity QYs have to be determined which take into account only photons absorbed by the porphyrin. Have the authors in their comparison of the emission intensities taken care of this inner filter effect by the Au particles?

In which solvent have the transient absorption experiments been performed? For the decay curves it is unclear which probe wavelength is plotted and which feature is observed. I assume it is the decay of the bleach of the LSPR band which occurs due to hot electron? I can not follow the conclusion from the decay kinetics that the lifetime of the hot electrons is prolonged as the authors claim here. Clearly in presence of the porphyrin the decay is faster, this means the hot electrons live shorter.

The assignment of the spectral features in the TA spectra also needs to be reevaluated. The authors show in Figure 5 d the TA spectra of just the porphyrin. The authors claim the positive feature to be due to the reduced porphyrin. How should the reduced porphyrin have formed? Is there electron donor

present in the solution? Do the authors have data from spectroelectrochemistry to support their claim? Assuming that this is a feature of the reduced porphyrin, I would expect to observe this feature also in the spectra of the Au-porphyrin system if indeed charge transfer to the porphyrin occurs. Why is this not the case. And what is the source of the negative feature at 680 nm in panel f (Figure 5)? Further the authors describe a positive peak appearing at 630-730 nm in the AU-porphyrin system, which increases in the first 10 ps. This I can not observe in the data. All positive peaks are decaying on this timescale. Also, this new peak is ascribed to an "excited state". What excited state is this supposed to be?

Although the described system is for sure interesting, especially the results from spectroscopy need major revision.

Reviewer #3 (Remarks to the Author):

In this work, the authors have been developed a system form by cobalt porphyrin bounded to plasmonic nanoparticles for photocatalytic hydrogen evolution. The results showed in this manuscript could be interested for the field. However, several issues could compromise the findings achieved. Thus, in view of main concerns described below I do not recommend the publication of this work at this time.

Comments:

*Regarding the preparation of the AuNP@CoTOyP nanostructures, it is not clear why the authors do not clean the excess of porphyrins that are not attached to the AuNPs. This could be easily do it by centrifugation, which additionally could be used to address the number of porphyrins per AuNP. This is of main importance, since the free porphyrin do not contribute to the catalytic activity of the system.

*The HER rate reported in this work it is much higher to the ones reported with other photocatalytic systems. However, it is not clear how this value it is calculated. Thus, it seems that this value was addressed just considering the mass of the porphyrin without taking into consideration the mass of AuNPs. In order to compare this value with others of the literature they should take into account the mass of the catalytic system composed of AuNP and porphyrin. If they take the mass of AuNPs into consideration the HER values obtained could be probably worse than the literature values.

*Taking in consideration the UV-vis spectra of the AuNP@CoTOyP dispersions with different concentrations of CoTPyP, author's claim that with the lower amount of porphyrin the degree of aggregation is higher, increasing the number of hot spots. However, from the UV-vis spectra can be clearly observed that the shift to the red of the plasmon band is much higher for 20nM and 200nM than for 2nM of porphyrin, which probably is a consequence of a much higher degree of aggregation for higher concentrations. Additionally this could mean that a higher number of hot spots it is expected for higher concentrations of porphyrins.

*The HER rate for the different porphyrin concentrations, should be do it after washing the samples and removing the excess of porphyrin molecules that are not attach to the AuNPs. Thus, increasing the porphyrin concentration increase mainly the free porphyrin in solution that are playing a minor catalytic role. Additionally, this issue it is also critical to justify the claimed role of the hot spots as main responsible for the high activity of the system.

*The mechanism proposed and conclusions obtained for the plasmonic catalytic enhancement of the system developed it is rather speculative. The authors compare the 15 nmAuNPs system with Au NRs and Ag NPs. The characterization of those systems are missing. As an example, in the case of Au NRs, those are probably synthesized in the presence of CTAB. Thus, it is difficult to expect that porphyrins are going to reach the surface of AuNPs. The conclusions obtained with this system are not clear. Regarding the Ag NPs, there is also no details about its surface functionalization. Indeed, the plasmon band of Ag nanoparticle overlaps with the absorption of porphyrin, and it could be expected a higher catalytic activity for Ag NPs in terms of hot electron injection or plasmon induced energy transfer.

*The synthesis of the porphyrin-coated plasmonic nanoparticle hybrid nanocatalyst should be further improved to be able to address the main concerns explained above.

Point by Point Response to Referees' Comments

From Referee #1:

The manuscript by Gang Lu and co-workers reports a hybrid photocatalyst for HER based on Au-nanoparticles and a Co-porphyrin catalyst that operates through localized surface plasmon resonance. The authors demonstrate that under visible light irradiation, plasmon generated hot carriers (electrons) can be transferred from Au-NP to the molecular catalyst, CoTPyP, enables the photocatalytic HER in aqueous medium. The paper is an important contribution in the growing field of plasmon driven photocatalysts for sustainable transformations. The photocatalytic HER studies are well complemented by experimental and theoretical investigations including SMFM analysis, temperature dependence of the catalysis, TAS measurements, and DFT. However, there are few things that need to be addressed before the manuscript can be considered for publication in Nature Communications.

Response: We thank the reviewer for the rigorous and valuable comments, which largely improve the quality of our manuscript.

Q1. It is unclear in the introduction whether there are any earlier works on combining plasmonic metal nanoparticles with molecular catalysts. If not, the authors should highlight this as it presents an important finding in this area.

Reply: Thank you very much for the valuable comment. To the best of our knowledge, our report is the first one on photocatalysis based on composite of plasmonic metal and molecular catalysts. This information has been included in the revised Introduction part by adding following sentence. “To the best of our knowledge, this is the first report on highly efficient photocatalysis based on composite of plasmonic metal and molecular catalysts.”
(Lines 21-23, Page 4)

Q2. What was rationale for using pyridine anchors for anchoring Co-porphyrins on AuNP? Wouldn't thiols be more suitable for immobilisation of Au surface?

Reply: Thank you very much for the valuable question. The pyridine anchors were used simply because of the strong binding between pyridine nitrogen and gold. This strong binding will favor the charge transfer between gold and Co-porphyrins. In addition, the

pyridine-containing porphyrin is easily bought from commercial market. As you indicated, thiols may also be used as anchors in this system. However, the thiol-containing porphyrins are difficult to buy from commercial market. The thiol anchors will be investigated in the near future after successful synthesis of porphyrins with thiol anchors.

Q3. Page 5, line 104: one CoTPyP molecular can link up to four AuNPs, not just two. However, the number of AuNPs in the aggregate might be limited by the size of porphyrin molecule and dimension of AuNPs. The authors should check if four AuNPs can be accommodated around a porphyrin molecule.

Reply: Thank you very much for the valuable comment. Although there are four pyridine anchors in a single CoTPyP molecule, this molecule cannot be simultaneously linked to four different AuNPs because of the steric effect and geometry configuration. First, according to DFT calculation and XPS spectra, two pyridine anchors can be simultaneously linked to one AuNP. Second, the size of CoTPyP molecule is ~1.1 nm, while the size of AuNPs is ~15 nm (Figure R1). When two AuNPs were linked by one CoTPyP molecule, the third AuNP cannot be linked to the same CoTPyP molecule anymore, because the surface of third AuNP cannot reach the pyridine anchor in CoTPyP molecule (Figure R1). Therefore, one CoTPyP molecule can only link up to two AuNPs. The related description of “Because of the steric effect and geometry configuration, one CoTPyP molecule may link two AuNPs together to form aggregates (Figure S1 in the Supporting Information).” has been supplied in the revised manuscript (Lines 20-22, Page 5).

Figure R1. Schematic diagram showing the linking of AuNPs with CoTPyP molecules. The third AuNP (right one) cannot be linked to the CoTPyP molecule (marked green) between first and second AuNPs.

Q4. Page 6: the UV-vis of AuNP@CoTPyP shows an intense peak at ~550 nm. What is the origin of this band?

Reply: Thank you very much for the question. As shown, the LSPR peak of AuNPs in aqueous solution locates at 520 nm (Figure 1e in the revised manuscript). It is known that the position of LSPR peak is highly related to the refractive index around plasmonic nanoparticles.¹ If abovementioned AuNPs were adsorbed with CoTPyP molecules, the LSPR peak red-shifted to ~526 nm (not ~550 nm) because of the change in local refractive index induced by molecular adsorption. The broad peak at ~620 nm is attributed to the aggregation of AuNPs, which is induced by the addition of CoTPyP molecules.

Q5. N1s XPS spectra of both CoTPyP and AuNP@CoTPyP show a peak assigned to graphitic N and the authors claim that the increase in this peak (with concomitant disappearance of pyridinic N) indicate anchoring of CoTPyP on Au surface. However, I am not quite sure why they are termed as 'graphitic N'. Do the authors mean metal coordinated pyrrolic N in CoTPyP? This should be clarified.

Reply: Thank you very much for the helpful comment. As suggested, the “graphitic N” has been changed to “metal coordinated pyridinic N” in the revised manuscript (Line 18, Page 6). The related Figure 1e has been corrected in the revised manuscript (Page 7)

Q6. The ratio of molecular catalyst and AuNPs used in photocatalysis experiments should be clearly discussed. The rate of HER clearly depends on the catalyst loading, which is rationalized based on formation of aggregates. However, the size of aggregates will likely be controlled by the ratio of CoTPyP molecules and number of AuNPs in the mixture. Highly dilute CoTPyP condition will favour 1:4 (Co:AuNP) aggregates, while high CoTPyP concentration can lead to 1:3, 1:2, 1:1 aggregates. This neither clearly described in results section nor in the experimental methods.

Reply: Thanks for your valuable comment. As shown in the reply to Q3, one CoTPyP molecule can link up to two AuNPs. Moreover, the aggregation of AuNPs can be realized by more than one CoTPyP molecules and the aggregation is usually not uniform (Figure R2), which is usually the case for the aggregation induced by molecular linking.² Therefore, instead of Co:AuNP ratio, the concentration of CoTPyP was discussed in our manuscript.

For easier understanding, Figure R2 has been added as Figure S7 in the revised Supporting Information and the preparation of AuNP@CoTPyP structures has been included in the revised Experimental part.

Figure R2. (a) SEM and (b) TEM images of the aggregation of AuNPs.

Q7. Page 15, line 289: the authors claim that electron transfer from AuNP to CoTPyP is facilitated by the positive charge of the CoTPyP molecule. This requires further clarification. If we assume that all four pyridines are coordinated to AuNPs (1:4 ratio), then they pyridines cannot be protonated to provide positively charged groups. In that case, the above statement is incorrect and there won't be any additional contributing factor due to charge of the molecule.

Reply: Thank you very much for the valuable comment. As discussed in the reply to Q3, one CoTPyP molecule can be linked up to two AuNPs. Some of the CoTPyP molecules may have their four pyridine groups simultaneously linked to gold surface (Figure R2), which usually leads to AuNP aggregation. However, the CoTPyP molecules not linked to AuNP gaps will have two pyridine groups exposed. These exposed pyridine groups can be protonated to provide positively charge groups. The description has been added in the revised manuscript. “The unlinked pyridine groups in CoTPyP may ionize to make the molecule positively charged.” (Lines 6-7, Page 16).

Q8. Can the authors clarify the results shown in Figure 5f? The discussion on this figure is unclear, in particular, the authors describe ‘a new positive peak also appeared at 630-730 nm’ which isn't clear in the figure.

Reply: Thank you very much for the valuable comment. The original description on Figure 5f was not accurate. The negative peak (not positive peak) appeared at 630–730 nm after 80 ps may be attributed to the emission from adsorbed CoTPyP molecules. It was reported that the molecules adsorbed on metal surface will experience a change in energy levels, usually narrow down of energy gap.³ Therefore, the observed peak at 630–730 nm might be attributed to the red-shifted emission of the adsorbed CoTPyP molecules. We have tried our best to improve the discussion in the revised manuscript. “... a new negative peak also appeared at 630–730 nm after 80 ps (Figure 5f). This new peak might be attributed to the emission from the adsorbed CoTPyP molecules.” (Lines 20-22, Page 18).

Q9. Can the authors clarify how the time scale of e-p coupling is related to the lifetime of the plasmon-generated hot carriers? The time scale gets shorter when CoTPyP is immobilized on AuNP. How does that translate into slower dissipation of the energy?

Reply: Thank you very much for the valuable comment. As indicated in many literatures,^{4,6} the decay of localized surface plasmon resonance (LSPR) follows following steps: (1) oscillation of electron cloud under light illumination; (2) generation of hot carriers via e-e scattering; (3) relaxation of hot carriers via e-p scattering; (4) dissipation of heat in crystal lattice (Figure R3). The e-p scattering happens within the time scale of 100 fs to 10 ps, and the energy of plasmon-generated hot carriers will be consumed in this e-p scattering process. Therefore, the lifetime of plasmon-generated hot carriers will be largely affected by e-p scattering in plasmonic nanostructures. Moreover, the lifetime of plasmon-generated hot carriers can also be shortened by chemical reactions which consume hot carriers (Figure R4).

Figure R3. Schematic illustration of the plasmon excitation and relaxation on the surface of a plasmonic metal nanoparticle. The straight yellow arrows represent the electron-electron scattering, while the curved ones represent the electron-phonon scattering. These hot carriers dissipate rapidly within 100 fs–1ps and 1–10 ps, respectively.⁷

In our case, the lifetime of plasmon-generated hot carriers was 3.7 ± 0.13 ps in the naked AuNP sample, and this lifetime decreased to 3.2 ± 0.08 ps when CoTPyP molecules were adsorbed. Before CoTPyP adsorption, the lifetime is mainly affected by the e-p scattering. After CoTPyP adsorption, the plasmon-generated hot carriers can transfer to the adsorbed CoTPyP molecules, leading to an inhibition of hot carrier recombination. Therefore, more hot carriers could be utilized in HER and the HER rate could be increased.

To make it easier to understand, the original discussion on transient absorption results were revised in the revised manuscript. “During the decay of LSPR, hot carriers are formed and then consumed *via* e-p scattering and chemical reaction. Therefore, a decrease in lifetime of hot carriers usually suggests an inhabitation of radiative decay and a favored chemical reaction. In our case, by fitting the decay curves to a two-term exponential model, it is revealed that the lifetime of plasmon-generated hot carriers was 3.7 ± 0.13 ps in the naked AuNP sample, and this lifetime decreased to 3.2 ± 0.08 ps when CoTPyP molecules were adsorbed. Before CoTPyP adsorption, the lifetime is mainly affected by the e-p scattering which consumes hot carriers. After CoTPyP adsorption, the plasmon-generated hot carriers can transfer to the adsorbed CoTPyP molecules for catalytic reactions, during which hot carriers are consumed. Therefore, the radiative decay pathway is inhibited and thus the HER rate is increased.” (Lines 24-25, Page 18 and Lines 1-10, Page 19).

Figure R4. Scheme showing the charge transfer at AuNP-CoTPyP interface.⁸

Q10. *Experimental methods: (i) further characterisation (e.g., elemental analysis) of CoTPyP is recommended; otherwise, original publication on CoTPyP should be cited. (ii)*

details of the EIS analysis (electrode preparation, circuit used to fit data) should be included. (iii) detailed preparation of AuNP@CoTPyP material for characterization should be included.

Reply: Thanks a lot for your valuable suggestions. We have carried out many more characterizations, and the description on experimental methods has been enriched in the revised manuscript.

(i) The synthesis of CoTPyP was reported by a previous literature, which has been cited as reference 59 in the revised manuscript. In our work, the TPyP molecules were bought from market (J&K Scientific Ltd.) and its $^1\text{H-NMR}$ spectrum was shown in Figure R5. After the synthesis of CoTPyP, we tried to analyze the structure of CoTPyP using NMR. However, the presence of cobalt interferes with the magnetic field, resulting in inaccurate chemical shift values. Luckily, the UV–Vis spectra can be used to analyze the reaction efficiency, and this method has been used in many literatures.⁹⁻¹¹ As shown, the typical Soret band red-shifted by 8 nm to 425 nm and the typical Q band at 537 nm showed up after reaction (Figure R6), suggesting a successful metalation. Moreover, Co element was clearly observed in EDS mapping of AuNP@CoTPyP nanostructure (Figure 1b), double confirming the successful synthesis of CoTPyP in previous step. The related details have been added in the revised manuscript. “The UV–Vis spectra (Figure S18 in the Supporting Information) showed the typical Soret band red-shifted by 8 nm to 425 nm and the typical Q band at 537 nm showed up after reaction, suggesting a successful metalation.” (Lines 8-10, Page 22).

Figure R5. $^1\text{H-NMR}$ spectrum of TPyP (400MHz, CDCl_3). $^1\text{H NMR}$ (400 MHz, CDCl_3) δ 9.08–9.07 (m, 1H), 8.88 (s, 1H), 8.18–8.17 (m, 1H).

Figure R6. UV–Vis absorption spectra of (a) TPyP and (b) CoTPyP.

Figure 1b. STEM image of AuNP@CoTPyP and corresponding EDS mapping images.

(ii) We have added the details of EIS analysis, including the electrode preparation and circuit fitting, in the revised manuscript. “The 10 μL AuNP@CoTPyP was drop-casted on the FTO slide to serve as the working electrode.” (Lines 11-12, Page 23). Detailed discussion on EIS spectra has been added in the revised manuscript, “The model of Randles equivalent circuit (inset in Figure 5a) was used to analyze the charge transfer at interface. R_s and R_{ct} are the solution and charge transfer resistances, C_w is the Warburg impedance, and C_{DL} is the double-layer capacitance.” (Lines 12-15, Page 16).

(iii) The details of the synthesis of AuNP@CoTPyP have been included in the revised manuscript. “*Synthesis of AuNP@CoTPyP.* CoTPyP powder was dissolved in 1 mL of 0.1 M hydrochloric acid to obtain a final concentration of 2 nM. Then, 140 μL of the prepared CoTPyP solution was rapidly injected in to a bottle containing 5 mL gold colloid (0.488 mM) and 15 mL water under magnetic stirring at 300 rpm, and this solution was stirred for additional 2 min. The pH value of the final solution was 3.9.” (Lines 11-15, Page 22).

From Referee #2:

The manuscript submitted by Gang Lu and colleagues sure reports on an interesting approach for HER and the system shows amazing performance and stability. Nevertheless, especially in the data analyzing the mechanism in this system, the data does not well support the claims in the current state and a major revision is needed before this manuscript can be considered for publication in nature communications. In detail I have the following comments, corrections and questions:

Response: Thank you very much for the valuable comments and suggestions, based on which we have improved the quality of our manuscript.

Q1. Page 6 XPS X-ray photoelectron spectroscopy not photoluminescence spectroscopy

Reply: Thank you very much for the correction. The full name of XPS has been corrected to “X-ray photoelectron spectroscopy” in the revised manuscript (Line 12, Page 6).

Q2. Figure 1: could the authors also show the absorption spectrum of just the AuNP. Simply add it to panel c. This would better illustrate the appearance of the aggregation induced resonance.

Reply: Thank you very much for the valuable suggestion. The absorption spectrum of pristine AuNPs has been added in Figure 1c in the revised manuscript (Page 7).

Figure 2c. UV–Vis extinction spectra of AuNPs, CoTPyP and AuNP@CoTPyP (CoTPyP concentration = 2 nM).

Q3. Figure 3a: The authors argue in the text at page 10 that increasing the CoTPyP concentration leads to less aggregation of the AuNPs and refer to the changes in the absorption spectra. The band at 620 nm was assigned to be an aggregation induced resonance. If the aggregation is highest at 2 nM of the porphyrin, shouldn't this aggregation induced feature be strongest at this concentration and decrease with increasing concentration of the porphyrin? Can the authors explain why this feature in Figure 3 panel a is increasing first with increasing concentration and then suddenly seems to disappear at the highest concentration?

Reply: Thank you very much for the valuable comment. At the CoTPyP concentration of 2 nM, the AuNPs aggregated seriously since one CoTPyP molecule can link up to two AuNPs simultaneously. The interconnection between AuNPs and CoTPyP molecules leads a serious aggregation. This aggregation has been confirmed by many experimental results. First, the color of the AuNP colloid changed seriously, implying a serious aggregation. Second, new peaks around ~625 and ~955 nm was observed after the introduction of 2 nM CoTPyP (Figure R7a). The appearance of these two new peaks suggests a serious aggregation.¹² Third, this aggregation has also been confirmed by TEM images (Figure R7b). This aggregation is obviously more serious than the case with higher CoTPyP concentration (Figure R7b-c). The related description has been added in the revised manuscript. "... new peaks appeared at ~625 and ~955 nm in UV-Vis spectrum (Figure 3a, Figure S7a in the Supporting Information), suggesting a significant aggregation of AuNPs, which was confirmed by TEM image (Figure S7b in the Supporting Information)." (Lines 3-6, Page 10).

Figure R7. (a) UV-Vis extinction spectrum and the photograph of the AuNP@CoTPyP prepared by using 2 nM CoTPyP solution. (b-c) TEM images of AuNP@CoTPyP.

Q4. Caption Figure 4f does not match the panel. Please check carefully.

Reply: Thank you very much for the valuable comment. The labels of “Ru(bpy)₂/CoTPyP” and “AuNP@CoTPyP” have been corrected in Figure 4f in the revised manuscript (Page 14).

Figure 4f. HER rates of Ru(bpy)₂/CoTPyP and AuNP@CoTPyP in air and at 50 °C.

Q5. Does the binding of the porphyrin to the Au particles change the molecular structure? Have the authors tried to collect SERS spectra to observe changes in vibrational modes upon binding?

Reply: Thank you very much for the insightful suggestion. The Raman and SERS spectra of CoTPyP (Figure R8) have been provided in the revised Supporting Information as Figure S2. The Raman peaks of CoTPyP experienced negligible shift after being adsorbed on AuNP, suggesting that the molecular structure can be inferred to be the same after adsorption.

The related description of “Moreover, the Raman peaks of CoTPyP molecules shifted slightly after being adsorbed onto the surface of AuNPs (Figure S3 in the Supporting Information), also suggesting an interaction between AuNPs and CoTPyP molecules.” has been supplied in the revised manuscript (Lines 8-11, Page 6). The description of “The ν_2 and ν_4 vibrational modes of CoTPyP were detected at 1555 and 1352 cm⁻¹, respectively. It is known that ν_2 mode arises from the $\nu(\text{C}_\beta\text{C}_\beta)$ stretching of the porphyrin ring together with a smaller contribution from the $\nu(\text{C}_\alpha\text{C}_m)$, while ν_4 mode is a response of the pyrrole half-ring symmetric stretching modes.¹³ After being adsorbed onto AuNPs, ν_4 vibrational

mode of CoTPyP shifted to a higher wavenumber at 1355 cm⁻¹, ν_2 modes shifted to a lower wavenumber at 1543 cm⁻¹.” has been added in the revised Supporting Information.

Figure R8. Raman (bottom curve) and SERS (top curve) spectra of CoTPyP.

Q6. The rates are given in mol/g h. What is the mass which is used as reference here? Could the authors also give TONs and TOFs. These values also are very often used to evaluate HER and it would be helpful to have here also an idea of the order of magnitude to compare with other systems reported in literature.

Reply: Thank you very much for the valuable suggestion. The catalytic rate is highly related to the mass of used photocatalyst. Therefore, in many literatures, the mass of catalyst is used for calculating the catalytic rate of HER.¹⁴⁻¹⁷ In our work, the catalytic activity was provided by CoTPyP, while the AuNPs were confirmed to be catalytically inert. As a result, we used the mass of CoTPyP as the reference in our work.

As suggested, we also calculated the TON and TOF of our catalytic system and then compared these values with those in literatures. Following equations are used for calculating TON and TOF, $TON = \frac{2n(H_2)}{n(catalyst)}$ and $TOF = TON/t$.¹⁸ However, in literatures, the calculation of TOF is chaotic and the obtained TOF cannot be simply compared across different literatures. In most cases, the amount of catalyst is used for calculating TOF. In some cases, the amount of photosensitizer is also used for calculating TOF. In our case, the amount of the catalyst (CoTPyP) was used for calculating TOF, since the activity is limited by the amount of CoTPyP in our case. The obtained TON is 6850 and TOF is 4890 h⁻¹ for our system. Our TOF value is very high compared with the values in literatures (Table R1).

Note that in some literatures the mass of photosensitizer was used as reference for calculating TOF (Table R2). Therefore, it is not reasonable to compare these values with our results.

The information of TOF has been included in the revised Abstract and Introduction. “... the HER rate and turn-over frequency (TOF) reach 3.21 mol·g⁻¹h⁻¹ and 4890 h⁻¹ ...” (Lines 9-10, Page 2), “... a superior HER rate of 3.21 mol g⁻¹h⁻¹ and a high TOF of 4890 h⁻¹ ...” (Line 24, Page 4), “The TOF of our system was determined as 4890 h⁻¹.” (Lines 13-14, Page 7).

Table R1. Comparison of TOFs of reported photocatalysts under visible light irradiation.

Catalyst	Light source (W)	HER rate (mmol g ⁻¹ h ⁻¹)	TOF (h ⁻¹)	reference
CoTPyP	300 (>420 nm)	3214	4890	This work
[Co ^{III} (dmgH) ₂ (py)Cl]	500	12	12	19
NiP	1 sun (λ > 300 nm)	0.4	41	15
Cobaloxime	AM 1.5 light	0.8	2.72	16
CoGGH	LED light	N.A.	62.9	17
Co ²⁺ catalysts	AM 1.5G	N.A.	833	20
Cat/PS blend nanofibres	500(>420 nm)	244	1400	14

N.A. refers to not available.

Table R2. Comparison of TOFs of reported photosensitizer under visible light irradiation.

photosensitizer	Light source (W)	HER rate (mmol g ⁻¹ h ⁻¹)	TOF (h ⁻¹)	reference
ZnDC(p-NI)PP	LED	35.7	120	21
TBPyZnP-Ir	OLED light	16.12	6.15	20
Ru-4	LED light (λ = 450 nm)	N.A.	378	22
Ir-4	175 (>420 nm)	17.8	6435	23

N.A. refers to not available.

Q7. The authors compare the rates for Au and Ag particles and argue the lower activity of the Ag system to be caused by lower light absorption in the visible spectrum by the Ag particles. I assume that these experiments which are compared here were performed under

broadband illumination? Later the authors describe HER evolution experiments with defined irradiation wavelengths. Was such an experiment also performed for Ag? How does Au and Ag compare upon irradiation of Ag and Au in resonance with the respective LSPR?

Reply: Thank you very much for the valuable comment. You made a good point. We did the experiments for Ag system under broadband illumination. Indeed, the wavelength is crucial in plasmon-mediated reactions. As you suggested, we performed the HER experiments for Ag system under monochromatic light illuminations, which was realized by applying band pass filters. A maximum HER rate of $1.21 \text{ mol g}^{-1}\text{h}^{-1}$ was observed at 430 nm, close to the plasmonic band of AgNPs at $\sim 422 \text{ nm}$, in the AgNP@CoTPyP hybrid system (Figure R9). Longer or shorter wavelengths both led to a drop in HER rate. Therefore, the excitation of Ag plasmonic mode is essential in the photocatalytic HER reaction. The related description of “**Similar results were also observed in AgNP@CoTPyP (Figure S16 in the Supporting Information), double confirming the great contribution of LSPR excitation in HER enhancement.**” has been supplied in the revised manuscript (Lines 3-5, Page 15).

However, the HER rate of AgNP system at 430 nm was still lower than that of AuNP system at 550 nm ($1.92 \text{ mol g}^{-1}\text{h}^{-1}$). The lower activity of AgNP@CoTPyP compared with AuNP@CoTPyP could be attributed to following reasons. First, the light absorption of AuNPs and AgNPs are different. Second, because clean surface is important in catalysis, the AuNPs and AgNPs were synthesized by using the simple Turkevich method which does not introduce long-chain ligands. The size of obtained AuNPs was $\sim 15 \text{ nm}$, while that of AgNPs was $\sim 50 \text{ nm}$. The larger particle size of AgNPs may explain the lower catalytic activity of AgNP@CoTPyP. Third, the aggregation is crucial in our system. The CoTPyP-induced aggregation of AgNPs is different to that of AuNPs, which may also contribute the lower catalytic activity of AgNP@CoTPyP. The related description of “**... large particle size of AgNPs ($\sim 50 \text{ nm}$), and different CoTPyP-induced aggregation, which are not good for catalytic reactions**” has been supplied in the revised manuscript (Lines 1-2, Page 13).

Figure R9. UV–Vis extinction spectrum of AgNP@CoTPyP (CoTPyP concentration = 2 nM) and the HER rates under illuminations of monochromatic light. The power was set as 5.2 W at all wavelengths.

Q8. How was the temperature controlled in the plasmonic system. How can the authors make sure, that there is no local heating event under the “controlled” temperature conditions?

Reply: Thank you very much for the valuable comment. The temperature of system was controlled by using a constant-temperature water bath, and the reaction suspension was stirred continuously for easier mass transport (Figure R10). Therefore, the temperature of the reaction suspension was kept same during the whole reaction due to the fast heat transfer. The related details “In some cases, the temperature of the reaction system was controlled by using a constant-temperature water bath and the reaction suspension was stirred continuously” have been supplied in the revised manuscript (Lines 23-24, Page 22).

Actually, the previous measurements can only indicate the macroscopic temperature of the reaction system. However, experimental determination of nanoscale temperatures in plasmonic systems is always a big challenge. When the reaction was carried out in water bath and under magnetic stirring, a fast heat transfer is guaranteed to minimize the non-uniform distribution of temperature in the catalytic system. In many literatures, researchers used water bath to minimize the possible contribution from plasmonic heating,^{24,25} which is very similar to what we used here.

Figure R10. Scheme showing the reaction cell for photocatalytic HER.

Q9. The authors claim, that besides heating there might be a transfer of hot electrons from the Au particles to the porphyrin reaction center. Hence the Au particles kind of act as sensitizer if I understood correctly. To trace the interfacial charge transfer transient absorption EIS was performed and these results indicate a decreased resistance in this systems hinting to a charge transfer event, also PL spectroscopy has been performed. Here is not entirely clear how the PL of the excited porphyrin is quenched by a hot electron transfer event. Can the authors please elaborate on the PL experiment a bit more in detail. Also to really evaluate emission intensity QYs have to be determined which take into account only photons absorbed by the porphyrin. Have the authors in their comparison of the emission intensities taken care of this inner filter effect by the Au particles?

Reply: Thank you very much for the valuable comment. As shown in our Results and Discussion, the AuNPs play multiple roles in photocatalytic HER. First, the light absorption is enhanced due to surface plasmon resonance. Second, more hot carriers are excited and transferred to the molecular catalyst of CoTPyP. Third, plasmonic heating also play a part in enhanced HER. These effects work simultaneously to enhance the HER rate. In terms of PL spectra, CoTPyP and AuNP@CoTPyP was excited under 430 nm illumination. The PL peaks of CoTPyP at 650 and 710 nm became weaker after being adsorbed to AuNPs (Figure 5b). This drop in PL intensity can be explained by the charge transfer between AuNP and CoTPyP. Without AuNPs, the excited electrons in CoTPyP molecules decay via radiative and non-radiative pathways, and the radiative pathway gives PL. When AuNPs are introduced, the PL intensity was obviously reduced, suggesting that non-radiative decay is enhanced. This non-radiative decay favors the charge transfer

between CoTPyP molecule and adsorbed hydrogen, and thus the HER rate was improved. To avoid possible misunderstanding, the related discussion has been revised in the revised manuscript. "... non-radiative decay pathway is improved and thus the hot carriers are more easily separated at the AuNP-CoTPyP interface." (Lines 22-23, Page 16).

As suggested, the details of PL measurements were enriched in the revised Experimental part. "A fluorescence spectrophotometer (F-7100, Hitachi, Japan) was used to record the fluorescence spectra of samples in a standard 10×10 mm² quartz cell and a Xenon lamp mounted with a 430 nm band pass filter was used as the excitation source." (Lines 14-17, Page 23). In terms of inner filter effect, it can be minimized by reducing the concentration of solutions (absorbance < 0.05). In our case, the AuNP concentration was very low (0.12 mM) and the CoTPyP concentration was also low (2 nM). Therefore, the inner filter effect should be very weak.

Q10. In which solvent have the transient absorption experiments been performed? For the decay curves it is unclear which probe wavelength is plotted and which feature is observed. I assume it is the decay of the bleach of the LSPR band which occurs due to hot electron? I cannot follow the conclusion from the decay kinetics that the lifetime of the hot electrons is prolonged as the authors claim here. Clearly in presence of the porphyrin the decay is faster, this means the hot electrons live shorter.

Reply: Thank you very much for your valuable comments. The transient absorption experiments were performed in water solvent added with 5% methanol. We plotted the decay curve of AuNP at 520 nm and AuNP@CoTPyP at 530 nm (Figure 5c-f). These information has been included in the revised manuscript. "All the transient absorption experiments were performed in water solvent added with 5% methanol." (Lines 1-2, Page 18). "... corresponding fitting of AuNP (at 520 nm) and AuNP@CoTPyP (at 530 nm), respectively." (Lines 3-4, Page 18).

As indicated in many literatures,^{4,6} the decay of localized surface plasmon resonance (LSPR) follows following steps: (1) oscillation of electron cloud under light illumination; (2) generation of hot carriers via e-e scattering; (3) relaxation of hot carriers via e-p scattering; (4) dissipation of heat in crystal lattice (Figure R3). The e-p scattering happens within the time scale of 100 fs to 10 ps, and the energy of plasmon-generated hot carriers

will be consumed during this e-p scattering process. Therefore, the lifetime of plasmon-generated hot carriers will be largely affected by e-p scattering in plasmonic nanostructures. Moreover, the lifetime of plasmon-generated hot carriers can also be shortened by chemical reactions which consume hot carriers. In our case, the lifetime of plasmon-generated hot carriers was 3.7 ± 0.13 ps in the naked AuNP sample, and this lifetime decreased to 3.2 ± 0.08 ps when CoTPyP molecules were adsorbed. Before CoTPyP adsorption, the lifetime is mainly affected by the e-p scattering. After CoTPyP adsorption, the plasmon-generated hot carriers can transfer to the adsorbed CoTPyP molecules, leading to an inhabitation of hot carrier recombination and a decrease in hot carrier lifetime. Therefore, more hot carriers could be utilized in HER and the HER rate could be increased. To make it easier to understand, the original discussion on transient absorption results has been changed to “During the decay of LSPR, hot carriers are formed and then consumed *via* e-p scattering and chemical reaction. Therefore, a decrease in lifetime of hot carriers usually suggests an inhabitation of radiative decay and a favored chemical reaction. In our case, by fitting the decay curves to a two-term exponential model, it is revealed that the lifetime of plasmon-generated hot carriers was 3.7 ± 0.13 ps in the naked AuNP sample, and this lifetime decreased to 3.2 ± 0.08 ps when CoTPyP molecules were adsorbed. Before CoTPyP adsorption, the lifetime is mainly affected by the e-p scattering which consumes hot carriers. After CoTPyP adsorption, the plasmon-generated hot carriers can transfer to the adsorbed CoTPyP molecules for catalytic reactions, during which hot carriers are consumed. Therefore, the radiative decay pathway is inhibited and thus the HER rate is increased.” in the revised manuscript (Lines 24-25, Page 18 and Lines 1-10, Page 19).

Figure R3. Schematic illustration of the plasmon excitation and relaxation on the surface of a plasmonic metal nanoparticle. The straight yellow arrows represent the electron-electron scattering, while the curved ones represent the electron-phonon scattering. These hot carriers dissipate rapidly within 100 fs–1ps and 1–10 ps, respectively.⁷

Figure R4. Scheme showing the charge transfer at AuNP-CoTPyP interface.⁸

Q11. The assignment of the spectral features in the TA spectra also needs to be reevaluated. The authors show in Figure 5 d the TA spectra of just the porphyrin. The authors claim the positive feature to be due to the reduced porphyrin. How should the reduced porphyrin have formed? Is there electron donor present in the solution? Do the authors have data from spectroelectrochemistry to support their claim? Assuming that this is a feature of the reduced porphyrin, I would expect to observe this feature also in the spectra of the Au-porphyrin system if indeed charge transfer to the porphyrin occurs. Why is this not the case? And what is the source of the negative feature at 680 nm in panel f (Figure 5)? Further the authors describe a positive peak appearing at 630-730 nm in the AU-porphyrin system, which increases in the first 10 ps. This I cannot observe in the data. All positive peaks are decaying on this timescale. Also, this new peak is ascribed to an “excited state”. What excited state is this supposed to be? Although the described system is for sure interesting, especially the results from spectroscopy need major revision.

Reply: Thank you very much for your valuable comments. There was a typo in the original manuscript. We mean “excited porphyrin”, not “reduced porphyrin”. Besides, we realized that the original spectra in Figure 5d are not good after consulting an expert on transient absorption. There was possibly an issue in instrument calibration. Therefore, we redid the transient absorption measurement for the CoTPyP molecules (2 μM), and 5% methanol was added in the solution as electron donor during measurement. Note that this concentration was used for acceptable signal-to-noise ratio. The new spectra have been

used to update the previous ones. The broad and positive absorption bands between 560 and 750 nm were identified as the light absorption of the excited CoTPyP. This feature cannot be observed in AuNP@CoTPyP (Figure 5f), probably due to the low CoTPyP concentration (The low CoTPyP concentration of 2 nM favors AuNP aggregation and HER) used in measurement. The original discussion on these spectra were changed to “A broad and positive absorption band also appeared within the range of 610–720 nm, possibly due to the light absorption of excited state of CoTPyP.” in the revised manuscript (Lines 14-16, Page 18).

Figure 5d. Ultrafast transient absorption spectra of the CoTPyP molecules excited by a 430 nm pump beam (pulse density = $90 \mu\text{J}\cdot\text{cm}^{-2}$).

The original description on Figure 5f was not accurate. The negative peak (not positive peak) appeared at 630–730 nm after 80 ps may be attributed to the emission from adsorbed CoTPyP molecules. It was reported that the molecules adsorbed on metal surface will experience a change in energy levels, usually narrow down of energy gap.³ Therefore, the observed peak at 630–730 nm might be attributed to the red-shifted emission of the adsorbed CoTPyP molecules. We have tried our best to improve the discussion in the revised manuscript. “... a new negative peak also appeared at 630–730 nm after 80 ps (Figure 5f). This new peak might be attributed to the emission from the adsorbed CoTPyP molecules.” (Lines 20-22, Page 18).

From Referee #3:

In this work, the authors have been developed a system form by cobalt porphyrin bounded to plasmonic nanoparticles for photocatalytic hydrogen evolution. The results showed in this manuscript could be interested for the field. However, several issues could compromise the findings achieved. Thus, in view of main concerns described below I do not recommend the publication of this work at this time.

Response: Thanks a lot for your professional comments. We have tried our best to improve the quality of our manuscript based on the given comments.

Q1. Regarding the preparation of the AuNP@CoTPyP nanostructures, it is not clear why the authors do not clean the excess of porphyrins that are not attached to the AuNPs. This could be easily do it by centrifugation, which additionally could be used to address the number of porphyrins per AuNP. This is of main importance, since the free porphyrin do not contribute to the catalytic activity of the system.

Reply: Thank you very much for the valuable suggestion. We would like to point it out that the ratio of AuNP:CoTPyP is difficult to determine. Although one CoTPyP molecule can link to up to two AuNPs, aggregates composed of multiple AuNPs can be obtained (Figure R1 & R2), because of the interconnection between AuNPs and CoTPyP molecules. Although the AuNP:CoTPyP ratio in solution can be calculated, it is actually a mixture of different AuNP:CoTPyP ratios. After washing, it is even more difficult to determine the AuNP:CoTPyP ratio, since the amount of the CoTPyP molecules washed away cannot be determined precisely.

Anyway, it is interesting to know the performance of the AuNP:CoTPyP catalyst after washing away the excess CoTPyP molecules. The AuNPs aggregate easily after washing away excess CoTPyP molecules. We tried out best to washing away excess CoTPyP molecules for three times and try to avoid further aggregation. After washing, the HER performance AuNP@CoTPyP at 2 nM hardly changed, since most of CoTPyP molecules were linked to AuNPs at this low concentration (Figure R11). Increasing the concentration of CoTPyP molecules to 20 and 200 nM, the HER performance decreased after washing away excess CoTPyP molecules. At these higher CoTPyP concentrations, more CoTPyP molecules should be linked to AuNPs than those at lower concentration, even after washing.

In addition, less aggregation was observed at higher CoTPyP concentrations (Figure 3a). The obtained HER performance was lower, suggesting the great contribution of AuNP aggregation. These AuNP aggregation produces many plasmonic hot spots, which favor the catalytic HER process. Further increasing the CoTPyP concentration to 2000 nM further decreased the HER performance after washing away excess CoTPyP molecules, double confirming the contribution of AuNP aggregation in HER enhancement. The conclusion here is consistent with the cases without washing. The related description of “To exclude the effect of non-adsorbed catalyst molecules, we also performed the photocatalytic experiments after washing away excess CoTPyP molecules. At the CoTPyP concentration of 2 nM, the amount of produced hydrogen was basically unchanged after washing. While the amount of produced hydrogen slightly decreased after washing at higher CoTPyP concentrations of 20 and 200 nM (Figure S9 in the Supporting Information). These results indicate the great contribution of AuNP aggregation in HER enhancement. The conclusion here is consistent with the cases without washing. However, it is difficult to calculate the HER rate and TOF after washing, because it is challenging to know the amount of CoTPyP residue in the systems.” has been added in the revised manuscript (Lines 4-12, Page 11).

Figure R11. HER production at different concentrations of CoTPyP after washing.

Q2. The HER rate reported in this work is much higher to the ones reported with other photocatalytic systems. However, it is not clear how this value it is calculated. Thus, it seems that this value was addressed just considering the mass of the porphyrin without taking

into consideration the mass of AuNPs. In order to compare this value with others of the literature they should take into account the mass of the catalytic system composed of AuNP and porphyrin. If they take the mass of AuNPs into consideration the HER values obtained could be probably worse than the literature values.

Reply: Thank you very much for the valuable comment. The catalytic rate is highly related to the mass of used photocatalyst. Therefore, in many literatures, the mass of catalyst is used for calculating the catalytic rate of HER.^{14,18} In our work, the catalytic activity was provided by CoTPyP, while the AuNPs were confirmed to be catalytically inert. As a result, we used the mass of CoTPyP as the reference in our work.

We also want to emphasize the novelty of our work. We composited the plasmonic AuNPs and molecular catalyst of CoTPyP together to obtain AuNP@CoTPyP nanostructure. As a result, the plasmonic effects could be used for improving the catalytic performance of the molecular catalyst. In addition, the catalytic stability of our system is also high. To the best of our knowledge, this is the first report on composite of plasmonic metal and molecular catalyst for high-performance photocatalytic HER.

We also calculated the TON and TOF of our catalytic system and then compared these values with those in literatures. Following equations are used for calculating TON and TOF, $TON = \frac{2n(H_2)}{n(catalyst)}$ and $TOF = TON/t$. In our case, the amount of the catalyst (CoTPyP) was used for calculating TOF, since the activity is limited by the amount of CoTPyP in our case. The obtained TON is 6850 and TOF is 4890 h⁻¹ for our system. Our TOF value is very high compared with the values in literatures (Table R1).

The information of TOF has been included in the revised Abstract and Introduction. "... the HER rate and turn-over frequency (TOF) reach 3.21 mol·g⁻¹h⁻¹ and 4890 h⁻¹ ..." (Lines 9-10, Page 2), "... a superior HER rate of 3.21 mol g⁻¹h⁻¹ and a high TOF of 4890 h⁻¹ ..." (Line 24, Page 4), "The TOF of our system was determined as 4890 h⁻¹." (Lines 13-14, Page 7).

Table R1. Comparison of TOFs of reported photocatalysts under visible light irradiation.

Catalyst	Light source (W)	HER rate (mmol g ⁻¹ h ⁻¹)	TOF (h ⁻¹)	reference
CoTPyP	300 (>420 nm)	3214	4890	This work
[Co ^{III} (dmgH) ₂ (py)Cl]	500	12	12	19
NiP	1 sun (λ > 300 nm)	0.4	41	15
Cobaloxime	AM 1.5 light	0.8	2.72	16
CoGGH	LED light	N.A.	62.9	17
Co ²⁺ catalysts	AM 1.5G	N.A.	833	20
Cat/PS blend nanofibres	500(>420 nm)	244	1400	14

N.A. refers to not available.

Q3. Taking in consideration the UV-vis spectra of the AuNP@CoTOyP dispersions with different concentrations of CoTPyP, author's claim that with the lower amount of porphyrin the degree of aggregation is higher, increasing the number of hot spots. However, from the UV-vis spectra can be clearly observed that the shift to the red of the plasmon band is much higher for 20nM and 200nM than for 2nM of porphyrin, which probably is a consequence of a much higher degree of aggregation for higher concentrations. Additionally this could mean that a higher number of hot spots it is expected for higher concentrations of porphyrins.

Reply: Thank you very much for the valuable comment. At the CoTPyP concentration of 2 nM, the AuNPs aggregate seriously since one CoTPyP molecules can link up to two AuNPs simultaneously. The interconnection between AuNPs and CoTPyP molecules leads a serious aggregation. This aggregation has been confirmed by many experimental results. First, the color of the AuNP colloid changed seriously, implying a serious aggregation. Second, new peaks around ~625 and ~955 nm was observed after the introduction of 2 nM CoTPyP molecules (Figure 7a). The appearance of these two new peaks suggests a serious aggregation.¹² In addition, this aggregation has also been confirmed by TEM images (Figure R7b). This aggregation is obviously more serious than the case with higher CoTPyP concentration (Figure R7b-c).

Figure R7. (a) UV–Vis extinction spectrum and the photograph of the AuNP@CoTPyP prepared by using 2 nM CoTPyP solution. (b-c) TEM images of AuNP@CoTPyP.

The related description has been added in the revised manuscript. “... new peaks appeared at ~625 and ~955 nm in UV-Vis spectrum (Figure 3a, Figure S7a in the Supporting Information), suggesting a significant aggregation of AuNPs, which was confirmed by TEM image (Figure S7b in the Supporting Information).” (Lines 3-6, Page 10).

Q4. The HER rate for the different porphyrin concentrations, should be do it after washing the samples and removing the excess of porphyrin molecules that are not attach to the AuNPs. Thus, increasing the porphyrin concentration increase mainly the free porphyrin in solution that are playing a minor catalytic role. Additionally, this issue it is also critical to justify the claimed role of the hot spots as main responsible for the high activity of the system.

Reply: Thank you very much for the valuable comment. As suggested, we studied the performance of catalyst after washing away the excess CoTPyP molecules. We tried out best to washing away excess CoTPyP molecules for three times. After washing, the HER performance AuNP@CoTPyP at 2 nM hardly changed, since most of CoTPyP molecules were linked to AuNPs at this low concentration (Figure R11). Increasing the concentration of CoTPyP molecules to 20 and 200 nM, the HER performance decreased after washing away excess CoTPyP molecules. At these higher CoTPyP concentrations, more CoTPyP molecules are linked to single AuNPs than those at lower concentration, even after washing. Therefore, the AuNP aggregation is less serious because of the avoided interconnection between AuNPs and CoTPyP molecules. However, the HER performance was lower, suggesting the great contribution of AuNP aggregation. These AuNP aggregation produces

many plasmonic hot spots, which favor the catalytic HER process. Further increasing the CoTPyP concentration to 2000 nM further decreased the HER performance after washing away excess CoTPyP molecules, double confirming the contribution of AuNP aggregation in HER enhancement. The conclusion here is consistent with the cases without washing.

Q5. The mechanism proposed and conclusions obtained for the plasmonic catalytic enhancement of the system developed it is rather speculative. The authors compare the 15 nm AuNPs system with Au NRs and Ag NPs. The characterization of those systems are missing. As an example, in the case of Au NRs, those are probably synthesized in the presence of CTAB. Thus, it is difficult to expect that porphyrins are going to reach the surface of AuNPs. The conclusions obtained with this system are not clear. Regarding the Ag NPs, there is also no details about its surface functionalization. Indeed, the plasmon band of Ag nanoparticle overlaps with the absorption of porphyrin, and it could be expected a higher catalytic activity for Ag NPs in terms of hot electron injection or plasmon induced energy transfer.

Reply: Thank you very much for the valuable suggestion. As suggested, morphology, surface functionalization, and extinction spectra of AuNRs and AgNPs were fully characterized and the results have been added in the revised Supporting Information as Figure S9-S12. In addition, the synthesis procedures for AuNRs and AgNPs have been included in the revised manuscript. “*Synthesis of AgNPs. The AgNPs were synthesized by following the procedures in our previous report.*” (Lines 23-24, Page 21). “*Synthesis of AuNRs. The AuNRs were synthesized by following the procedures in previous report.*” (Lines 1-2, Page 22).

The AuNRs were synthesized by using the widely used seed-mediated method²⁶ and CTAB ligands are indeed present. As shown, the obtained AuNRs were highly uniform in morphology (Figure R12a), and the length and width were ~100 and ~50 nm, respectively. The UV–Vis extinction spectrum of the AuNR@CoTPyP nanostructure showed two plasmonic bands at ~526 nm and ~668 nm, respectively (Figure R12b). Therefore, broadband light can be utilized in photocatalytic HER. Note that the CTAB ligands were mainly washed away via centrifugation, leaving limited amount of CTAB residue. The CTAB residue is physically adsorbed on the surface of AuNRs and can be easily replaced

by the CoTPyP molecules, which can be strongly adsorbed via the gold-pyridine bonds. As shown in EDS mapping results, the spatial distributions of Co and N elements were highly consistent with that of Au (Figure R12c), suggesting a successful binding of CoTPyP molecules on gold surface. The related descriptions have been added in the revised manuscript. “... and the obtained gold nanorods were highly uniform in shape and size (Figure S10a in the Supporting Information).” (Lines 6-7, Page 12). “The UV–Vis spectrum of the gold nanorods (Figure S10b in the Supporting Information) showed two plasmonic bands at ~526 and ~668 nm ...” (Lines 7-9, Page 12). “... as shown in EDS mapping results, the spatial distribution of Co and N elements were highly consistent with that of Au (Figure S12 in the Supporting Information), suggesting a successful binding of CoTPyP molecules on gold surface in spite of the presence of the cetyltrimethylammonium bromide (CTAB) capping agent in gold nanorod colloid.” (Lines 13-17, Page 12).

Figure R12. (a) SEM image of AuNR@CoTPyP. (b) UV–Vis extinction spectrum of AuNR@CoTPyP. (c) STEM image of AuNR@CoTPyP and corresponding EDS element mapping images.

The AgNPs with average diameter of ~50 nm were synthesized (Figure R13a) using sodium citrate reduction method,²⁷ which does not introduce long-chain ligands. Note that 15 nm AgNPs cannot be synthesized by using this method. Usually, organic solvent and long-chain ligand are needed for synthesizing ~15 nm AgNPs, and these AgNPs are not good for catalysis because of the existence of long-chain ligand. The UV–Vis extinction spectrum of the AgNP@CoTPyP nanostructure showed the plasmonic band of AgNPs at

~424 nm (Figure R13b). As indicated by EDS mapping results (Figure R13c), CoTPyP molecules were also uniformly adsorbed on the surface of AgNPs. The related description has been added in the revised manuscript. “The morphology, surface functionalization, and extinction spectra of AgNPs@CoTPyP were shown in Figure S12 in the Supporting Information.” (Lines 21-22, Page 12). Detailed description has been added in the revised Supporting Information.

Figure R13. (a) SEM image of AgNP@CoTPyP. (b) UV–Vis extinction spectrum of AgNP@CoTPyP. (c) STEM image of AgNP@CoTPyP and corresponding EDS element mapping images.

We performed the HER experiments for AgNP@CoTPyP under monochromatic light illuminations, which was realized by applying band pass filters. A maximum HER rate of $1.21 \text{ mol g}^{-1}\text{h}^{-1}$ was observed at 430 nm, close to the plasmonic band of AgNPs at ~422 nm, in the AgNP@CoTPyP hybrid system (Figure R8). Longer or shorter wavelengths both led to a drop in HER rate. Therefore, the excitation of Ag plasmonic mode is essential in the photocatalytic HER reaction. However, the HER rate of AgNP system at 430 nm was still lower than that of AuNP system at 550 nm ($1.92 \text{ mol g}^{-1}\text{h}^{-1}$). Note that the AuNPs and AgNPs were synthesized by using the simple Turkevich method, which does not introduce long-chain ligands. The size of obtained AuNPs was ~15 nm, while that of AgNPs was ~50 nm. The lower activity of AgNP@CoTPyP compared with AuNP@CoTPyP is possibly attributed to the large particle size of AgNPs, which is not good for catalytic reactions. The related descriptions “... and large particle size of AgNPs (~50 nm), and different CoTPyP-

induced aggregation, which are not good for catalytic reactions” has been supplied in the revised manuscript (Lines 1-2, Page 13).

Figure R8. UV-Vis extinction spectrum of AgNP@CoTPyP (CoTPyP concentration = 2 nM) and the HER rates under illuminations of monochromatic light. The power was set as 5.2 W at all wavelengths.

Q6. The synthesis of the porphyrin-coated plasmonic nanoparticle hybrid nanocatalyst should be further improved to be able to address the main concerns explained above.

Reply: Thanks a lot for your valuable comment. The porphyrin-coated plasmonic nanoparticle hybrid nanocatalyst was synthesized by using a simple solution-based method. Typically, the AuNPs were synthesized by using the simple Turkevich method, and then the obtained colloidal solution was mixed with CoTPyP solution to reach a final CoTPyP concentration of 2 nM. The colloidal solution and the CoTPyP concentration were varied in some cases. This information has been included in the revised Experimental part.

We also studied the performance of catalyst after washing away the excess CoTPyP molecules. The results have been shown in the response to Q1 from Reviewer 3.

References

1. Guo, L., Jackman, J.A., Yang, H.-H., Chen, P., Cho, N.-J., Kim, D.-H. Strategies for enhancing the sensitivity of plasmonic nanosensors. *Nano Today* **10**, 213-239 (2015).
2. Shein, J.B., Lai, L.M.H., Eggers, P.K., Paddon-Row, M.N., Gooding, J.J. Formation of Efficient Electron Transfer Pathways by Adsorbing Gold Nanoparticles to Self-Assembled Monolayer Modified Electrodes. *Langmuir* **25**, 11121-11128 (2009).

3. Zhang, Y., He, S., Guo, W., Hu, Y., Huang, J., Mulcahy, J.R., *et al.* Surface-Plasmon-Driven Hot Electron Photochemistry. *Chem. Rev.* **118**, 2927-2954 (2018).
4. Zhan, C., Chen, X.J., Yi, J., Li, J.F., Wu, D.Y., Tian, Z.Q. From plasmon-enhanced molecular spectroscopy to plasmon-mediated chemical reactions. *Nat. Rev. Chem.* **2**, 216-230 (2018).
5. Zhang, Y.C., He, S., Guo, W.X., Hu, Y., Huang, J.W., Mulcahy, J.R., *et al.* Surface-Plasmon-Driven Hot Electron Photochemistry. *Chem. Rev.* **118**, 2927-2954 (2018).
6. Zhang, Z.L., Zhang, C.Y., Zheng, H.R., Xu, H.X. Plasmon-Driven Catalysis on Molecules and Nanomaterials. *Acc. Chem. Res.* **52**, 2506-2515 (2019).
7. Zhu, Y.M., Guan, M.D., Wang, J., Sheng, H.X., Chen, Y.Q., Liang, Y., *et al.* Plasmon-mediated photochemical transformation of inorganic nanocrystals. *Appl. Mater. Today* **24**, 101125-101150 (2021).
8. Besteiro, L.V., Yu, P., Wang, Z., Holleitner, A.W., Hartland, G.V., Wiederrecht, G.P., *et al.* The fast and the furious: Ultrafast hot electrons in plasmonic metastructures. Size and structure matter. *Nano Today* **27**, 120-145 (2019).
9. Oldacre, A.N., Friedman, A.E., Cook, T.R. A Self-Assembled Cofacial Cobalt Porphyrin Prism for Oxygen Reduction Catalysis. *J. Am. Chem. Soc.* **139**, 1424-1427 (2017).
10. Adilina, I.B., Hara, T., Ichikuni, N., Shimazu, S. Oxidative cleavage of isoeugenol to vanillin under molecular oxygen catalysed by cobalt porphyrin intercalated into lithium taeniolite clay. *J. Mol. Catal. A: Chem.* **361-362**, 72-79 (2012).
11. Barbosa Neto, N.M., De Boni, L., Mendonça, C.R., Misoguti, L., Queiroz, S.L., Dinelli, L.R., *et al.* Nonlinear Absorption Dynamics in Tetrapyrrolyl Metalloporphyrins. *J. Phys. Chem. B* **109**, 17340-17345 (2005).
12. Hentschel, M., Saliba, M., Vogelgesang, R., Giessen, H., Alivisatos, A.P., Liu, N. Transition from Isolated to Collective Modes in Plasmonic Oligomers. *Nano Lett.* **10**, 2721-2726 (2010).
13. Lokesh, K.S., De Keersmaecker, M., Adriaens, A. Self Assembled Films of Porphyrins with Amine Groups at Different Positions: Influence of Their Orientation on the Corrosion Inhibition and the Electrocatalytic Activity. *Molecules* **17**, (2012).
14. Tian, J., Zhang, Y., Du, L., He, Y., Jin, X.-H., Pearce, S., *et al.* Tailored self-assembled photocatalytic nanofibres for visible-light-driven hydrogen production. *Nat. Chem.* **12**, 1150-1156 (2020).

15. Martindale, B.C.M., Hutton, G.A.M., Caputo, C.A., Reisner, E. Solar Hydrogen Production Using Carbon Quantum Dots and a Molecular Nickel Catalyst. *J. Am. Chem. Soc.* **137**, 6018-6025 (2015).
16. Banerjee, T., Haase, F., Savasci, G., Gottschling, K., Ochsenfeld, C., Lotsch, B.V. Single-Site Photocatalytic H₂ Evolution from Covalent Organic Frameworks with Molecular Cobaloxime Co-Catalysts. *J. Am. Chem. Soc.* **139**, 16228-16234 (2017).
17. Chakraborty, S., Edwards, E.H., Kandemir, B., Bren, K.L. Photochemical Hydrogen Evolution from Neutral Water with a Cobalt Metallopeptide Catalyst. *Inorg. Chem.* **58**, 16402-16410 (2019).
18. Qureshi, M., Takanabe, K. Insights on Measuring and Reporting Heterogeneous Photocatalysis: Efficiency Definitions and Setup Examples. *Chem. Mater.* **29**, 158-167 (2017).
19. Lazarides, T., Delor, M., Sazanovich, I.V., McCormick, T.M., Georgakaki, I., Charalambidis, G., *et al.* Photocatalytic hydrogen production from a noble metal free system based on a water soluble porphyrin derivative and a cobaloxime catalyst. *Chem. Commun.* **50**, 521-523 (2014).
20. Tritton, D.N., Bodedla, G.B., Tang, G., Zhao, J., Kwan, C.-S., Leung, K.C.-F., *et al.* Iridium motif linked porphyrins for efficient light-driven hydrogen evolution via triplet state stabilization of porphyrin. *J. Mater. Chem. A* **8**, 3005-3010 (2020).
21. Bodedla, G.B., Wong, W.Y., Zhu, X.J. Coupling of a new porphyrin photosensitizer and cobaloxime cocatalyst for highly efficient photocatalytic H₂ evolution. *J. Mater. Chem. A* **9**, 20645-20652 (2021).
22. Guo, S., Chen, K.-K., Dong, R., Zhang, Z.-M., Zhao, J., Lu, T.-B. Robust and Long-Lived Excited State Ru(II) Polyimine Photosensitizers Boost Hydrogen Production. *ACS Catal.* **8**, 8659-8670 (2018).
23. Wang, P., Guo, S., Wang, H.J., Chen, K.K., Zhang, N., Zhang, Z.M., *et al.* A broadband and strong visible-light-absorbing photosensitizer boosts hydrogen evolution. *Nat. Commun.* **10**, 3155-3157 (2019).
24. Pan, L., Zhu, Y., Wang, Z., Xu, X., He, H., Du, W., *et al.* Plasmonic Cocatalyst with Electric and Thermal Stimuli Boots Solar Hydrogen Evolution. *Solar RRL* **4**, 2000094 (2020).
25. Qiu, J., Wei, W.D. Surface Plasmon-Mediated Photothermal Chemistry. *J. Phys. Chem. C* **118**, 20735-20749 (2014).

26. Ye, X., Zheng, C., Chen, J., Gao, Y., Murray, C.B. Using Binary Surfactant Mixtures To Simultaneously Improve the Dimensional Tunability and Monodispersity in the Seeded Growth of Gold Nanorods. *Nano Lett.* **13**, 765-771 (2013).
27. Chen, Y., Zhu, Y., Sheng, H., Wang, J., Zhang, C., Chen, Y., *et al.* Molecular Coadsorption of p-Hydroxythiophenol on Silver Nanoparticles Boosts the Plasmon-Mediated Decarboxylation Reaction. *ACS Catal.* **12**, 2938-2946 (2022).

List of Changes

1. The information of TOF has been included in the revised Abstract and Introduction. "... the HER rate and turn-over frequency (TOF) reach $3.21 \text{ mol} \cdot \text{g}^{-1} \cdot \text{h}^{-1}$ and 4890 h^{-1} ..." (Lines 9-10, Page 2). "... a superior HER rate of $3.21 \text{ mol} \cdot \text{g}^{-1} \cdot \text{h}^{-1}$ and a high TOF of 4890 h^{-1} ..." (Line 24, Page 4). "The TOF of our system was determined as 4890 h^{-1} ." (Lines 13-14, Page 7).
2. As suggested by reviewer 1, the sentence of "To the best of our knowledge, this is the first report on highly efficient photocatalysis based on composite of plasmonic metal and molecular catalysts." has been added in the revised manuscript to highlight the novelty of our work (Lines 21-23, Page 4).
3. The discussion on AuNP aggregation has been changed to "Because of the steric effect and geometry configuration, one CoTPyP molecule may link two AuNPs together to form aggregates (Figure S1 in the Supporting Information)." in the revised manuscript (Lines 20-22, Page 5). "This aggregation is caused by the interconnection of AuNPs and CoTPyP molecules, since one CoTPyP molecule can link up to two AuNPs simultaneously." (Lines 6-8, Page 10). In addition, Figure S7 has been added in the revised Supporting Information.
4. Raman and SERS spectra of CoTPyP, as well as corresponding description, have been added in the revised manuscript and Supporting Information. "Moreover, the Raman peaks of CoTPyP molecules shifted slightly after being adsorbed onto the surface of AuNPs (Figure S3 in the Supporting Information), also suggesting an interaction between AuNPs and CoTPyP molecules." (Lines 8-11, Page 6). Figure S2 has been added in the revised Supporting Information.

5. The full name of XPS has been corrected to “X-ray photoelectron spectroscopy” in the revised manuscript (Line 12, Page 6).
6. The “graphitic N” was corrected to “metal coordinated pyridinic N” in the revised manuscript (Line 18, Page 6). The labels in Figure 1e have been corrected in the revised manuscript (Page 7).
7. Absorption spectrum of pristine AuNPs has been added in Figure 1c in the revised manuscript (Page 7).
8. The descriptions of AuNP aggregation at CoTPyP concentration of 2 nM has been revised in the revised manuscript. “... new peaks appeared at ~625 and ~955 nm in UV-Vis spectrum (Figure 3a, Figure S7a in the Supporting Information), suggesting a significant aggregation of AuNPs, which was confirmed by TEM image (Figure S7b in the Supporting Information).” (Lines 3-7, Page 10). In addition, Figure S7 has been added in the revised Supporting Information.
9. The TEM image of AuNP@CoTPyP (20 nM) was added in the Supporting Information. “... and TEM image (Figure S8 in the Supporting Information).” (Line 16, Page 10)
10. As suggested, the catalytic properties of AuNP@CoTPyP has been studied after washing away excess CoTPyP molecules. Following discussion has been added in the revised manuscript. “To exclude the effect of non-adsorbed catalyst molecules, we also performed the photocatalytic experiments after washing away excess CoTPyP molecules. At the CoTPyP concentration of 2 nM, the amount of produced hydrogen was basically unchanged after washing. While the amount of produced hydrogen slightly decreased after washing at higher CoTPyP concentrations of 20 and 200 nM (Figure S9 in the Supporting Information). These results indicate the great contribution of AuNP aggregation in HER enhancement. The conclusion here is consistent with the cases without washing. However, it is difficult to calculate the HER rate and TOF after washing, because it is challenging to know the amount of CoTPyP residue in the systems.” (Lines 4-12, Page 11). In addition, Figure S9 has been added in the revised Supporting Information.
11. The morphology, surface functionalization, and extinction spectra of AuNRs have been further characterized and corresponding discussion has been changed in the revised manuscript. “... and the obtained gold nanorods were highly uniform in shape

and size (Figure S10a in the Supporting Information).” (Lines 6-7, Page 12). “The UV–Vis spectrum of the gold nanorods (Figure S10b in the Supporting Information) showed two plasmonic bands at ~526 and ~668 nm, ...” (Lines 7-9, Page 12). “... as shown in EDS mapping results, the spatial distribution of Co and N elements were highly consistent with that of Au (Figure S12 in the Supporting Information), suggesting a successful binding of CoTPyP molecules on gold surface in spite of the presence of the cetyltrimethylammonium bromide (CTAB) capping agent in gold nanorod colloid.” (Lines 13-17, Page 12).

12. The morphology, surface functionalization, and extinction spectra of AgNPs have been further characterized and corresponding discussion has been changed. “The morphology, surface functionalization, and extinction spectra of AgNPs@CoTPyP were shown in Figure S13 in the Supporting Information.” (Lines 21-22, Page 12).
13. The comparison of AgNP@CoTPyP and AuNP@CoTPyP has been changed to “... large particle size of AgNPs (~50 nm), and different CoTPyP-induced aggregation, which are not good for catalytic reactions” in the revised manuscript (Lines 1-2, Page 13).
14. The labels of “Ru(bpy)₂/CoTPyP” and “AuNP@CoTPyP” have been corrected in Figure 4f in the revised manuscript (Page 14).
15. The power of the monochromatic light with different individual wavelengths was included in the revised manuscript. “The power was set as 5.2 W at all wavelengths.” (Line 7, Page 14).
16. The results of AgNP@CoTPyP under monochromatic light illumination were added in the revised manuscript. “Similar results were also observed in AgNP@CoTPyP (Figure S16 in the Supporting Information), double confirming the great contribution of LSPR excitation in HER enhancement.” (Lines 3-5, Page 15)
17. The description of the positive charge of the CoTPyP has been added in the revised manuscript. “The unlinked pyridine groups in CoTPyP may ionize to make the molecule positively charged.” (Lines 6-7, Page 16).
18. Detailed discussion on circuit fitting in EIS spectrum has been added in the revised manuscript. “The model of Randles equivalent circuit (inset in Figure 5a) was used to analyze the charge transfer at interface. R_s and R_{ct} are the solution and charge transfer

resistances, C_w is the Warburg impedance, and C_{DL} is the double-layer capacitance.” (Lines 12-15, Page 16).

19. To avoid possible misunderstanding, the discussion of PL experiment has been revised in the revised manuscript. “... non-radiative decay pathway is improved and thus the hot carriers are more easily separated at the AuNP-CoTPyP interface.” (Line 22, Page 16).
20. The details of the transient absorption experiments have been enriched in the revised manuscript. “All the transient absorption experiments were performed in water solvent added with 5% methanol.” (Lines 1-2, Page 18). “... corresponding fitting of AuNP (at 520 nm) and AuNP@CoTPyP (at 530 nm), respectively.” (Lines 3-4, Page 18).
21. The description on transient absorption spectrum of CoTPyP has been rewritten in the revised manuscript. “... a weak bleaching peak around at ~537 nm (Figure 5d), corresponding to ground state absorption of CoTPyP molecules, was perfectly matched with the second large UV-Vis absorption peak of CoTPyP molecules (Figure 1c). Note that the peak related to the main absorption peak was missing in transient absorption spectra, because it partially overlaps with the pump wavelength (430 nm).” (Lines 10-14, Page 18).
22. The original discussion on CoTPyP spectra has been changed to “A broad and positive absorption band also appeared within the range of 610–720 nm, possibly due to the light absorption of excited state of CoTPyP.” in the revised manuscript (Lines 14-16, Page 18).
23. We have tried our best to improve the discussion on the negative peak at ~630-730 nm in the revised manuscript. “Meanwhile, a new negative peak also appeared at 630–730 nm after 80 ps (Figure 5f). This new peak might be attributed to the emission from the adsorbed CoTPyP molecules.” (Lines 20-22, Page 18).
24. To make it easier to understand, the original discussion on transient absorption results of AuNP@CoTPyP was revised in the revised manuscript. “During the decay of LSPR, hot carriers are formed and then consumed via e-p scattering and chemical reaction. Therefore, a decrease in lifetime of hot carriers usually suggests an inhabitation of radiative decay and a favored chemical reaction. In our case, by fitting the decay curves to a two-term exponential model, it is revealed that the lifetime of plasmon-generated

hot carriers was 3.7 ± 0.13 ps in the naked AuNP sample, and this lifetime decreased to 3.2 ± 0.08 ps when CoTPyP molecules were adsorbed. Before CoTPyP adsorption, the lifetime is mainly affected by the e-p scattering which consumes hot carriers. After CoTPyP adsorption, the plasmon-generated hot carriers can transfer to the adsorbed CoTPyP molecules for catalytic reactions, during which hot carriers are consumed. Therefore, the radiative decay pathway is inhibited and thus the HER rate is increased.” (Lines 24-25, Page 18 and Lines 1-10, Page 19).

25. The synthesis procedures for AuNRs and AgNPs have been included in the revised manuscript. “*Synthesis of AgNPs*. The AgNPs were synthesized by following the procedures in our previous report.” (Lines 23-24, Page 21). “*Synthesis of AuNRs*. The AuNRs were synthesized by following the procedures in previous report.” (Lines 1-2, Page 22).
26. The synthesis of CoTPyP was reported by a previous literature, which has been cited as reference 59 in the revised manuscript. The UV-Vis spectra of CoTPyP have been added in the revised manuscript. “The UV-Vis spectra (Figure S18 in the Supporting Information) showed the typical Soret band red-shifted by 8 nm to 425 nm and the typical Q band at 537 nm showed up after reaction, suggesting a successful metalation.” (Lines 8-10, Page 22).
27. The details of the synthesis of AuNP@CoTPyP have been included in the revised manuscript. “*Synthesis of AuNP@CoTPyP*. CoTPyP powder was dissolved in 1 mL of 0.1 M hydrochloric acid to obtain a final concentration of 2 nM. Then, 140 μ L of the prepared CoTPyP solution was rapidly injected in to a bottle containing 5 mL gold colloid (0.488 mM) and 15 mL water under magnetic stirring at 300 rpm, and this solution was stirred for additional 2 min. The pH value of the final solution was 3.9.” (Lines 11-15, Page 22).
28. The details of photocatalytic reaction have been enriched in the revised manuscript. “The reaction system was prepared in a 40 mL reactor by rapidly mixing above solution and 500 μ L methanol at 300 rpm. Then, photocatalytic hydrogen generation experiments were carried out under illumination of a 300 W Xenon lamp mounted with a long pass filter ($\lambda \geq 400$ nm, UV400CUT).” (Lines 16-19, Page 22).

29. The details of temperature control have been included in the revised manuscript. “In some cases, the temperature of the reaction system was controlled by using a constant-temperature water bath and the reaction suspension was stirred continuously” (Lines 23-24, Page 22).
30. The details of the electrode preparation have been added in the revised manuscript. “The 10 μ L AuNP@CoTPyP was drop-casted on the FTO slide to serve as the working electrode.” (Lines 11-12, Page 23).
31. As suggested, the details of PL measurements have been enriched in the revised Experimental part. “A fluorescence spectrophotometer (F-7100, Hitachi, Japan) was used to record the fluorescence spectra of samples in a standard 10 \times 10 mm² quartz cell and a Xenon lamp mounted with a 430 nm band pass filter was used as the excitation source.” (Lines 14-17, Page 23).

REVIEWER COMMENTS

Reviewer #1 (Remarks to the Author):

The authors have addressed some of the comments, but it still requires major revision because there are still many inconsistencies in the manuscript:

1) The assembly of CoTPyP/AuNP hybrids should be analysed by ICP to confirm the ratio. This can be done by centrifuging the hybrid to wash off the excess CoTP, followed by rinsing with suitable solvents. In figure S9, the authors report the activity of the 'purified' hybrids. This data combined with ICP results, will provide a more accurate TON values.

2) Figure 1c: Why is CoTPyP soret band not observed in the hybrid material? Is it because of the low concentration of the CoTP?

3) Figure 2c: The HER rate in mol/g should be calculated using the total weight of AuNP-CoTPyP hybrid. For a molecular catalyst, it is highly unusual to report the activity as mol/g. To report the activity with respect to CoTPyP, the H₂ production should be converted to TON for each cycle (3h).

4) Figure 2c: If fresh CoTP was added during each cycle, then the recycling experiment essentially indicate recyclability of only the AuNP particles (sensitiser). This should be clarified in the text as well as the experimental section. It is recommended to perform the recycling experiments using the hybrid material collected by centrifugation (without adding extra CoTP during each cycle).

5) Figure 3a: The 620 nm peak has been attributed to Au-NP aggregates, but there is a clear enhancement of this peak from 2nm to 200nm CoTP. This contradicts with the authors' statement that the aggregation is reduced with higher CoTP concentration. Rather the spectra show disappearance of the 620 nm band when the concentration is increased from 200 nm to 2000nm. This section requires major revision.

6) Figure 5: the PL spectra of CoTPyP suggest quenching of photoexcited porphyrin by AuNP. What's the mechanism of the quenching? Is it caused by injection of hot electron from AuNP to CoTP? The authors should clarify this. Which components are donors and acceptors here?

7) Figure 5: The assignment of the peaks in TA spectra (panel d-f) should be complemented by spectroelectrochemistry analysis of CoTPyP.

Reviewer #2 (Remarks to the Author):

The authors answered most of my questions and addressed my comments appropriately. Nevertheless, a few details remain to be clarified before publication.

The authors discuss the negative feature building up in the TA data in Figure 5f to emission. First of all, in a transient absorption pump-probe experiment this is stimulated emission. Second, I am wondering why this feature is spectrally so different from the steady-state emission detected? The peaks in the emission spectra of the aggregate are at different spectral position. How do the authors explain this discrepancy?

Figure 2c, please check the color code, I think the assignment in the legend of the figure is not correct.

In my previous review, I have been asking, whether the authors regarded in the comparison of the emission intensities, that the Au particles are also absorbing light and this needs to be regarded when evaluating the change in emission intensities. The authors answered, that they set the absorbance of the samples to be very low and hence no inner filter effects play a role. This is only true with respect to reabsorption and has nothing to do with the point I was trying to make. Emission can only be originating from photons which are absorbed by the porphyrin (if we ignore for now that also something like energy transfer from Au to the porphyrin might occur). If the authors set the absorbance of a sample of just porphyrin and the aggregates to be e.g. 0.05 at the excitation wavelength each, they will observe even in the case of no interaction less emission intensity for the aggregate, because less porphyrin is in this sample. To be able to directly compare the emission intensity aggregate and just porphyrin and learn about the potential interactions, the concentration of the porphyrin in the samples needs to be identical. If this is experimentally not possible. The emission intensity of the aggregate measured needs to be corrected for the difference in concentration (or the emission intensity of the porphyrin, whatever the authors prefer). I was not able to find any information on how the steady state emission was measured and whether considerations with this respect have been included in the data evaluation.

Reviewer #3 (Remarks to the Author):

The effort made by the authors to respond to the questions raised by the referees is to be acknowledged. Some of the questions were well addressed, however, the key experiments that justify the discoveries made are still not well justified. For all these reasons, I do not recommend its publication in Nature Communications.

1) Authors claim “To the best of our knowledge, our report is the first one on photocatalysis based on composite of plasmonic metal and molecular catalysts”. This is not correct, the field involving plasmonic nanostructures and molecular catalyst it is not novel. E.g. a recent review article already discuss about “plasmonic-Metal/Molecule Heterostructures for CO₂ reduction” Adv. Mater. Interfaces 2022, 9, 2102383.

2) Still remains an issue regarding the aggregation state of the Au NPs under different concentration of the molecular catalyst. The UV-Vis spectra on Figure 2 for the AuNP at 2nM of CoTPyP could support the formation of bigger aggregates but in any case do not fit with the UV-vis spectra in Figure S5 which it is supposed to be the “UV-Vis spectrum of AuNP@CoTPyP initially and after 45 hours of reaction”. Those spectra are clearly different and do not support the statements pointed out by the authors.

3) An interesting result of this work consists in verifying that Au nanoparticles are more efficient than Ag particles, and that the geometry of the particles plays a relevant role. However, this issue is not justified. If we assume that the concentration of Ag and Au is the same, in the case of Ag we will have a smaller number of particles as well as a smaller surface area. All this would mean a lower number of CoTPyP anchored to the surface of the Ag nanoparticles as well as a lower number of hot spots. This fact would be the possible reason for a lower efficiency. For all these reasons, the comparison does not make sense, and it is not well understood why 15 nm Ag nanoparticles were not synthesized attending the available synthetic methods in the literature.

In the case of Au NRs the authors must be sure that they have the same amount of Au in order to establish a discussion. Indeed, in the case of Au NRs the TEM image on Figure R12a, the Au NR are not in close contact to each other, which is rather different than the image of Au NPs with 2nM of CoTPyP. The presence of CTAB could be also the reason for this fact. The full interchange between CTAB and CoTPyP seems to be difficult and should be further proved.

4) The local temperature at the surface of the Au NPs can not be excluded just by using a cooling system for the bulk solution. It will be helpful to address the reaction at different temperatures without illumination.

Reviewer #4 (Remarks to the Author):

The manuscript by Sheng et al presents a hybrid system with enhanced photocatalytic properties towards HER. I feel that the present version has much improved, I have some concerns regarding the theoretical part.

- The first one regards the model they use for their calculations. It is not clear the specific surface of Au they consider, how exposed is this surface in the real sample, how many layers they use for thickness convergence and how big is their supercell in the xy plane (i.e. how their porphyrin coverage matches

the experimental one). I therefore ask the authors to describe in detail all these features and justify their choices.

- Porphyrins are large systems where van Der Waals interactions (molecule-molecule and molecule-surface) can be significant. I assume that dispersion forces are not considered in this work. I therefore ask the authors to include dispersion forces in their optimizations and energetic considerations.

- How many atoms have they considered to calculate Hessian/Frequencies needed for the entropic term? Calculating frequencies for the whole supercell is prohibitive with VASP (it uses expensive finite differences method) and the porphyrin is a very large system so the choice is not obvious. Please describe and justify the choice

- Calculations here are somehow underused, they mainly describe electronic charge transfer but they could be used to discern between a Tafel Volmer or Volmer Heyrovski mechanism. I ask therefore the authors to dig deeper on the mechanism from the theoretical point of view

Response to Referees' Comments

From Referee #1:

The authors have addressed some of the comments, but it still requires major revision because there are still many inconsistencies in the manuscript.

Response: Thank you very much for the valuable comments and suggestions, based on which we have further improved the quality of our manuscript.

***Q1.** The assembly of CoTPyP/AuNP hybrids should be analysed by ICP to confirm the ratio. This can be done by centrifuging the hybrid to wash off the excess CoTPyP, followed by rinsing with suitable solvents. In figure S9, the authors report the activity of the 'purified' hybrids. This data combined with ICP results, will provide a more accurate TON values.*

Response: Thank you very much for the helpful comment. We followed your suggestion. After washing away the excess CoTPyP molecules *via* centrifugation for three times, we obtained the cleaned AuNP@CoTPyP structures and then carried out the ICP measurements (Table R1). The Co:Au atomic ratios in the four samples were not linearly related to the concentrations of CoTPyP, because excess CoTPyP molecules were washed away in the case of high CoTPyP concentration.

At the typical CoTPyP concentration of 2 nM, the Co:Au atomic ratio was 1:1600, which matches perfectly with the calculated Co:Au atomic ratio based on the amount of input CoTPyP. According to the ICP analysis, a more accurate TON for each cycle (3 h) was calculated to be 13950. The related results have been added in the revised manuscript and the TON has been added as another y axis in Figure 2c-d and Figure S9. “After optimization, the HER rate and turn-over frequency (TOF) reach $3.21 \text{ mol} \cdot \text{g}^{-1} \cdot \text{h}^{-1}$ and 4650 h^{-1} ”(Line 10, Page 2) “... and a high TOF of 4650 h^{-1} under visible light illumination.”(Line 23, Page 4) “The TOF of our system was determined as 4650 h^{-1} by using the amount of CoTPyP as the reference.”(Line 14, Page 7) “... and corresponding TON ...”(Line 6-7, Page 8) “Inductively coupled plasma-optical emission spectroscopy (ICP-OES) was used to obtain the accurate Co:Au atomic ratios after washing (Table S2) for evaluation of the accurate TON values. According to the ICP-OES analysis, the Co:Au atomic ratio was 1:1600 for the AuNP@CoTPyP structure prepared at the typical CoTPyP concentration of 2 nM This Co:Au ratio matches perfectly with the one calculated based on the amount of input CoTPyP, since

all molecules were bounded on AuNP surface. Therefore, the previously obtained TON values should be accurate.”(Line 13-19, Page 11)

Table R1. ICP-OES analysis of the AuNP@CoTPyP structures prepared at different concentrations of CoTPyP.

C(CoTPyP)	2000 nM	200 nM	20 nM	2 nM
Co : Au	9.2 : 1000	9.1 : 1000	6.4 : 1000	1 : 1600

Figure S9. HER production and TON of the washed AuNP@CoTPyP samples prepared at different concentrations of CoTPyP.

Q2. Figure 1c: Why is CoTPyP soret band not observed in the hybrid material? Is it because of the low concentration of the CoTPyP?

Response: Yes! The concentration of CoTPyP was as low as low 2 nM in the hybrid structure; therefore, the Soret band of CoTPyP was not observed. We also measured the UV-Vis spectra of pure CoTPyP solution at 2 nM (Figure R1), in which the Soret band was also hardly observed. The related details have been added in the revised manuscript. “UV-Vis extinction spectra of AuNPs, CoTPyP (50 nM) ...” (Line 5, Page 7)

Figure R1. UV–Vis extinction spectra of CoTPyP (concentration = 2 nM) and AuNP@CoTPyP (CoTPyP concentration = 2 nM).

Q3. Figure 2c: The HER rate in mol/g should be calculated using the total weight of AuNP-CoTPyP hybrid. For a molecular catalyst, it is highly unusual to report the activity as mol/g. To report the activity with respect to CoTPyP, the H₂ production should be converted to TON for each cycle (3h).

Response: Thank you very much for the valuable suggestion. The catalytic rate is highly related to the mass of used photocatalyst. Therefore, in many literatures, the mass of catalyst is used for calculating the catalytic rate of HER.¹⁻⁵ In our work, the catalytic activity was provided by CoTPyP, while the AuNPs were confirmed catalytically inert. As a result, we used the mass of CoTPyP as the reference in our work.

As suggested, we calculated the TON for each cycle (3 h) and TOF of our catalytic system and then compared these values with those in literatures. Following equations were used for calculating TON and TOF, $TON = \frac{2n(H_2)}{n(catalyst)}$ and $TOF = TON/t$. In our case, the amount of the catalyst (CoTPyP) was used for calculating the TON and TOF, since the activity is limited by the amount of CoTPyP in our case. The obtained TON was 13950 and TOF was 4650 h⁻¹ for our system. Our TOF value is very high compared with those in literatures (Table R2). The related descriptions has been added in the revised manuscript. “... which corresponds to a turnover number (TON) of 13950 each cycle (3 h).”(Line 10, Page 9)

Table R2. Comparison of TOFs of reported photocatalysts under visible light irradiation.

Catalyst	Light source (W)	HER rate (mmol g ⁻¹ h ⁻¹)	TOF (h ⁻¹)	reference
CoTPyP	300 (>420 nm)	3214	4650	This work
[Co ^{III} (dmgH) ₂ (py)Cl]	500	12	12	6
NiP	1 sun (λ > 300 nm)	0.4	41	3
Cobaloxime	AM 1.5 light	0.8	2.72	2
CoGGH	LED light	N.A.	62.9	5
Co ²⁺ catalysts	AM 1.5G	N.A.	833	7
Cat/PS blend nanofibres	500(>420 nm)	244	1400	8

N.A. refers to not available.

Q4. Figure 2c: If fresh CoTPyP was added during each cycle, then the recycling experiment essentially indicate recyclability of only the AuNP particles (sensitizer). This should be clarified in the text as well as the experimental section. It is recommended to perform the recycling experiments using the hybrid material collected by centrifugation (without adding extra CoTPyP during each cycle).

Response: Thank you very much for the valuable comment. We indeed used the cleaned AuNP@CoTPyP collected by centrifugation for cyclic measurements and no extra CoTPyP was added during each catalytic cycle. The related details have been added in the revised manuscript. “The cleaned AuNP@CoTPyP collected by centrifugation were used for cyclic measurements.” (Line 7-8, Page 9) “After each cycle, the system was degassed and left in dark for one hour.” (Line 15-16, Page 24)

Q5. Figure 3a: The 620 nm peak has been attributed to Au-NP aggregates, but there is a clear enhancement of this peak from 2nm to 200nm CoTPyP. This contradicts with the authors’ statement that the aggregation is reduced with higher CoTPyP concentration. Rather the spectra show disappearance of the 620 nm band when the concentration is increased from 200 nm to 2000nm. This section requires major revision.

Response: Thank you very much for the valuable comment. Generally, new plasmonic band emerges after aggregation of plasmonic nanoparticles and the position and intensity of the newly emerged band are highly related to the degree of aggregation.⁹ At the CoTPyP concentration of 2000 nM, AuNPs cannot aggregate together because the high-concentration CoTPyP molecules could inhibit the aggregation of AuNPs. Decreasing the concentration of CoTPyP led to the aggregation of AuNPs and this aggregation will become more serious at a lower CoTPyP concentration, which has been discussed in main text and Figure 3a. At the CoTPyP concentration of 200 nM, a new peak, except for that at 525 nm, appeared at 630 nm. This new peak red-shifted to 650 nm at a lower CoTPyP concentration of 20 nM. At an ultralow CoTPyP concentration of 2 nM, the aggregation of AuNPs became more serious. Two evidences were provided. First, the aggregation was clearly shown in TEM image (Figure S7b in the Supporting Information). Second, the extinction spectra could reflect the aggregation of the AuNPs. Very broad peaks and strong background showed up at >620 nm, confirming a more serious and random aggregation of AuNPs. Similar aggregation and spectral changes have been reported in many literatures.^{9,10}

Figure S7. (a) UV–Vis extinction spectrum and photograph of the AuNP@CoTPyP prepared at CoTPyP concentration of 2 nM. (b) TEM image of AuNP@CoTPyP (CoTPyP concentration = 2 nM).

Q6. *Figure 5: the PL spectra of CoTPyP suggest quenching of photoexcited porphyrin by AuNP. What's the mechanism of the quenching? Is it caused by injection of hot electron from AuNP to CoTPyP? The authors should clarify this. Which components are donors and acceptors here?*

Response: You made a good point. To effectively excite the photoluminescence of CoTPyP, 430 nm light was used as the excitation source. We realize that this 430 nm laser cannot effectively excite the LSPR of AuNPs, but the fluorescence of CoTPyP molecules. Thus, the spectral change cannot reflect the plasmon-induced charge transfer at the AuNP-CoTPyP interface during catalysis, but the quenching of the fluorescence from CoTPyP. In this quenching process, the excited electrons in CoTPyP transfer to AuNP, and the AuNP is the acceptor. To further investigate the contribution of plasmon excitation, we tried to excite the AuNP-CoTPyP structure with green light, which can excite the LSPR of AuNPs. However, the photoluminescence from AuNPs was too weak to be detected. Then, we tried to use the photocurrent measurement to investigate the charge transfer. We measured the photocurrent response of the samples under illumination of green light (550±25 nm filter was applied to Xenon lamp). Clearly, the photocurrent under illumination for AuNP@CoTPyP was clearly larger than that for AuNPs only, suggesting an improved charge transfer at interfaces under light illumination (Figure R2a). Based on the direction of photocurrent (Figure R2b), it could be concluded that the plasmon-induced hot electrons transfer from AuNP to the adsorbed CoTPyP molecules. This result is highly consistent with the TA results and DFT calculation. The related details have been supplied in the revised manuscript (Figure 5b). **“In the photocurrent response spectra, photocurrent of the prepared**

AuNP@CoTPyP under illumination was clearly larger than that of the bare AuNPs, suggesting an improved charge transfer at the interfaces (Figure 5b).” (Line 20-22, Page 17) “Photocurrent measurements of the AuNPs and AuNP@CoTPyP (CoTPyP concentration = 2 nM). The samples were periodically illuminated with green light (550±25 nm filter was applied to Xenon lamp).” (Line 3-5, Page 18) “The photocurrent test was performed on an electrochemical analyzer (CHI 630E, CH Instrument) in a standard three-electrode electrochemical cell filled with 1 M PBS buffer. Carbon paper was used as the working electrode, and 5 mL of AuNP (0.488 mM) was drop-casted, followed by thermal annealing at 80 °C. The annealed sample was then soaked in CoTPyP solution (typically 2 nM) for ten minutes. A piece of Pt plate served as the counter electrode, while Ag/AgCl served as the reference electrode. A green light beam was used for sample excitation by applying a 550±25 nm filter to a Xenon lamp.” (Line 7-14, Page 25)

Figure R2. (a) Photocurrent measurements of the AuNPs and AuNP@CoTPyP (CoTPyP concentration = 2 nM). The samples were periodically illuminated with monochromatic light of 550 nm. (b) Schematic illustration of the photocurrent measurements for AuNP@CoTPyP.

Q7. Figure 5: The assignment of the peaks in TA spectra (panel d-f) should be complemented by spectroelectrochemistry analysis of CoTPyP.

Response: Thank you very much for the valuable suggestion. As suggested, we carried out spectroelectrochemical measurements for CoTPyP under positive and negative potentials in mixed solvent of acetonitrile and water (Figure R3). The shape of the feature in the UV-Vis differential absorption spectra of CoTPyP molecules under reduction condition (Figure R3b) was similar to that observed in the transient absorption spectra of CoTPyP (Figure 5d), suggesting that the reduction quenching pathway should be a dominant process and the formation of new species may be the

reduced state of CoTPyP.^{11,12} The slight red-shift presented in the UV-Vis differential absorption spectra is possibly due to the solvent of CH₃CN used for spectroelectrochemical measurements. CH₃CN was used as the solvent since it can enhance the conductivity of the solution. The results have been added in the revised manuscript and Supporting Information. “To verify this species, the spectroelectrochemical experiments were performed in N₂ atmosphere. The transient absorption spectra of CoTPyP match well with the shape of the UV-Vis differential absorption spectra of the reduced CoTPyP and were different from that of the oxidized CoTPyP (Figure S18), suggesting that the reduction quenching pathway should be a dominant process and the new species may be the reduced state of CoTPyP.”(Line 16-21, Page 19)

Figure R3. Spectroelectrochemistry of CoTPyP. UV-Vis differential absorption spectra of 2 μ M CoTPyP in mixed solvent of CH₃CN and water (1:1) under (a) oxidation (1 V) and (b) reduction (-1 V) conditions. The solutions were bubbled with N₂ gas before the measurements and the potential was versus Ag/AgCl electrode.

Spectroelectrochemical measurements were also performed for the AuNP and AuNP@CoTPyP structures under positive and negative potentials (Figure R4). Broad and intensive peak was present in all the UV-Vis differential absorption spectra, possibly due to the presence of AuNPs or aggregated AuNPs in AuNP@CoTPyP. The different peak positions in the AuNP and AuNP@CoTPyP samples may be attributed to the aggregation status of AuNPs in these two samples. The peaks observed in the CoTPyP sample could not be observed in the AuNP@CoTPyP sample, since the peaks are relatively weak compared with those from AuNPs. Therefore, only the broad and intensive peak from AuNPs or aggregated AuNPs was observed in the UV-Vis differential absorption spectra of AuNP@CoTPyP, and thus the feature of transient

absorption of AuNP@CoTPyP could not be complemented by spectroelectrochemistry analysis. In transient absorption of AuNP@CoTPyP, the feature from the CoTPyP and its deviations was hardly seen due to the strong peak from AuNPs.

Figure R4. Spectroelectrochemistry of AuNP and AuNP@CoTPyP. UV-Vis differential absorption spectra of AuNP and AuNP@CoTPyP in mixed solvent of CH₃CN and water (1:1) under (a) oxidation (1 V) and (b) reduction (-1 V) conditions. The solutions were bubbled with N₂ gas before the measurements and the potential was versus Ag/AgCl electrode.

From Referee #2:

The authors answered most of my questions and addressed my comments appropriately. Nevertheless, a few details remain to be clarified before publication.

Response: Thanks a lot for your professional comments. We have tried our best to further improve the quality of our manuscript based on the given comments.

Q1. *The authors discuss the negative feature building up in the TA data in Figure 5f to emission. First of all, in a transient absorption pump-probe experiment this is stimulated emission. Second, I am wondering why this feature is spectrally so different from the steady-state emission detected? The peaks in the emission spectra of the aggregate are at different spectral position. How do the authors explain this discrepancy?*

Response: Thank you very much for the valuable comment. As shown in Figure R5, negative peak appeared in the TA spectra of AuNP@CoTPyP from 80 to 1500 ps and the peak shifted slowly from ~670 to ~705 nm. These peaks could be attributed to the stimulated emission from the CoTPyP molecules, in which two peaks at ~665 and ~710 nm showed up (Figure R6). Due to the different lifetimes of these two PL events, the corresponding negative features appeared in the TA spectra at different time scales, well explaining the shift of negative feature in TA spectra. The related descriptions and figure have been added in the revised manuscript and Supporting Information. “Meanwhile, a new negative peak appeared at ~670 nm from 80 ps (Figure 5f) and gradually shifted to ~705 nm from 80 to 1500 ps (Figure 5f and S19). This peak could be attributed to the stimulated emission from the CoTPyP molecules. Note that two peaks showed up at ~665 and ~710 nm in the photoluminescence spectrum of CoTPyP (Figure S20). Due to the different lifetimes of these two photoluminescence events, they showed up in the transient spectra at different time scales, well explaining the observed features in 630-730 nm region.”(Line 25, Page 19 and Line 1-6, Page 20)

Figure R5. Ultrafast transient absorption spectra of AuNP@CoTPyP at 80-1500 ps.

Figure R6. Photoluminescence spectra of CoTPyP (concentration is 2 μ M) and AuNP@CoTPyP (CoTPyP concentration is 2 μ M).

Q2. Figure 2c, please check the color code, I think the assignment in the legend of the figure is not correct.

Response: Thank you very much for the correction. I think you were talking about the legend of Figure 1c, which has been corrected in the revised manuscript (Page 7).

Figure 1c. UV-Vis extinction spectra of AuNPs, CoTPyP (50 nM) and AuNP@CoTPyP (CoTPyP concentration = 2 nM).

Q3. In my previous review, I have been asking, whether the authors regarded in the comparison of the emission intensities, that the Au particles are also absorbing light and this needs to be regarded when evaluating the change in emission intensities. The authors answered, that they set the absorbance of the samples to be very low and hence no inner filter effects play a role. This is only true with respect to reabsorption and has nothing to do with the point I was trying to make. Emission can only be originating from photons which are absorbed by the porphyrin (if we ignore for now that also something like energy transfer from Au to the porphyrin might occur). If the authors set the absorbance of a sample of just porphyrin and the aggregates to be e.g. 0.05 at the excitation wavelength each, they will observe even in the case of no interaction less emission intensity for the aggregate, because less porphyrin is in this sample. To be able to directly compare the emission intensity aggregate and just porphyrin and learn about the potential interactions, the concentration of the porphyrin in the samples needs to be identical. If this is experimentally not possible. The emission intensity of the aggregate measured needs to be corrected for the difference in concentration (or the emission intensity of the porphyrin, whatever the authors prefer). I was not able to find

any information on how the steady state emission was measured and whether considerations with this respect have been included in the data evaluation.

Response: Thank you very much for the valuable comment. You made a good point. The concentration of CoTPyP in the samples needs to be identical for fair comparison. In the PL measurements, the concentration of CoTPyP was always kept as 2 μ M, since the sample at low concentration of 2 nM cannot be detected due to the limited sensitivity of instrument. This information has been included in the figure caption to the PL spectra, which has been moved to Supplementary Information (Figure S20). Based on the comment from Reviewer #1, we realize that the PL spectra cannot confirm the charge transfer from AuNP to CoTPyP during reaction, because the PL was excited by 430 nm light which cannot effectively excite the LSPR of AuNP. Therefore, we have shifted the PL result to the Supplementary Information and added photocurrent results for instead. We measured the photocurrent response of the samples under illumination of green light (550 \pm 25 nm filter was applied to Xenon lamp), which can effectively excite the LSPR of AuNPs. Clearly, the photocurrent under illumination for AuNP@CoTPyP was clearly larger than that for AuNPs only, suggesting an improved charge transfer at interfaces (Figure R2a). Based on the direction of photocurrent, it could be concluded that the plasmon-induced hot electrons transfer from AuNP to the adsorbed CoTPyP molecules (Figure R2b). This result is highly consistent with the TA results and DFT calculation. The related descriptions and preparation process have been supplied in the revised manuscript (Figure 5b). “In the photocurrent response spectra, photocurrent of the prepared AuNP@CoTPyP under illumination was clearly larger than that of the bare AuNPs, suggesting an improved charge transfer at the interfaces (Figure 5b).”(Line 20-22, Page 17) “Photocurrent measurements of the AuNPs and AuNP@CoTPyP (CoTPyP concentration = 2 nM). The samples were periodically illuminated with green light (550 \pm 25 nm filter was applied to Xenon lamp).” (Line 3-5, Page 18) “The photocurrent test was performed on an electrochemical analyzer (CHI 630E, CH Instrument) in a standard three-electrode electrochemical cell filled with 1 M PBS buffer. Carbon paper was used as the working electrode, and 5 mL of AuNP (0.488 mM) was drop-casted, followed by thermal annealing at 80 $^{\circ}$ C. The annealed sample was then soaked in CoTPyP solution (typically 2 nM) for ten minutes. A piece of Pt plate served as the counter electrode, while Ag/AgCl served as the reference electrode. A green light beam was used for sample excitation by applying a 550 \pm 25 nm filter to a Xenon lamp.”(Line 7-14, Page 25)

Figure R2. (a) Photocurrent measurements of the AuNPs and AuNP@CoTPyP (CoTPyP concentration = 2 nM). The samples were periodically illuminated with monochromatic light of 550 nm. (b) Schematic illustration of the photocurrent measurements for AuNP@CoTPyP.

From Referee #3:

The effort made by the authors to respond to the questions raised by the referees is to be acknowledged. Some of the questions were well addressed, however, the key experiments that justify the discoveries made are still not well justified. For all these reasons, I do not recommend its publication in Nature Communications.

Response: We thank the reviewer for the rigorous and valuable comments, which largely improve the quality of our manuscript.

Q1. *Authors claim “To the best of our knowledge, our report is the first one on photocatalysis based on composite of plasmonic metal and molecular catalysts”. This is not correct, the field involving plasmonic nanostructures and molecular catalyst it is not novel. E.g. a recent review article already discuss about “plasmonic-Metal/Molecule Heterostructures for CO₂ reduction” Adv. Mater. Interfaces 2022, 9, 2102383.*

Response: Thank you very much for the valuable comment. The as-mentioned description may be not very accurate and was removed from the revised manuscript. Indeed, someone has composited Au-TiO₂ structure with ruthenium-based molecular catalyst for photocatalytic carbon dioxide reduction by linking the molecules on TiO₂ surface, and CO₂ reduction was investigated. Note that the molecular catalysts were not bound on the surface of AuNPs but the surface of semiconductor TiO₂. They realized a TOF of 1200 h⁻¹, which was also calculated based on the amount of molecular catalyst.

In addition, the contribution of plasmonic effects in the enhanced photocatalysis was not studied. In contrast, in our work, the metalloporphyrin molecules were bound directly on the surface of AuNPs for improvement of photocatalytic HER and a high TOF of 4650 h^{-1} was obtained in our case. In addition, the contributions of three plasmonic effects in catalysis improvements were deeply discussed by using many theoretical and experimental methods, which were missing in the literature. Therefore, our work will deeply inspire the development of plasmon-molecular catalyst photocatalysis systems.

In another plasmon-molecule system mentioned in the literature, the molecular catalyst of RuCY was linked to the surface of p-GaN/AuNP structure and photoelectrochemical catalytic CO_2 reduction was investigated. Rather than photocatalysis, photoelectrochemical CO_2 reduction was studied in the work and the contribution of plasmonic effects was also not clarified. In addition, the RuCY was linked via a N-heterocyclic carbene linker, which may largely inhibit the charge transfer. Therefore, the reaction type in literature is very different to ours and molecular linker was necessary for the system in literature, introducing many complexity.

Q2. Still remains an issue regarding the aggregation state of the Au NPs under different concentration of the molecular catalyst. The UV-Vis spectra on Figure 2 for the AuNP at 2nM of CoTPyP could support the formation of bigger aggregates but in any case do not fit with the UV-vis spectra in Figure S5 which it is supposed to be the “UV-Vis spectrum of AuNP@CoTPyP initially and after 45 hours of reaction”. Those spectra are clearly different and do not support the statements pointed out by the authors.

Response: Thank you very much for the valuable comment. In our case, the aggregation of AuNPs was realized *via* the interlinking of AuNPs with CoTPyP molecules. This process is quite random and may vary slightly from batch to batch. The results in Figure 3 and Figure S5 were from different batches of samples; therefore, the UV-Vis spectra were slightly different. Even though, the UV-Vis spectra hardly changed after 45 h catalytic reaction (Figure S5), indicating the high stability of our catalytic system. To avoid possible misunderstanding, we have updated the spectra in Figure 3 and Figure S5 by using same batch of samples.

Figure S5. UV–Vis extinction spectra of AuNP@CoTPyP initially and after 45 hours of reaction.

Q3. An interesting result of this work consists in verifying that Au nanoparticles are more efficient than Ag particles, and that the geometry of the particles plays a relevant role. However, this issue is not justified. If we assume that the concentration of Ag and Au is the same, in the case of Ag we will have a smaller number of particles as well as a smaller surface area. All this would mean a lower number of CoTPyP anchored to the surface of the Ag nanoparticles as well as a lower number of hot spots. This fact would be the possible reason for a lower efficiency. For all these reasons, the comparison does not make sense, and it is not well understood why 15 nm Ag nanoparticles were not synthesized attending the available synthetic methods in the literature. In the case of Au NRs the authors must be sure that they have the same amount of Au in order to establish a discussion. Indeed, in the case of Au NRs the TEM image on Figure R12a, the Au NR are not in close contact to each other, which is rather different than the image of Au NPs with 2nM of CoTPyP. The presence of CTAB could be also the reason for this fact. The full interchange between CTAB and CoTPyP seems to be difficult and should be further proved.

Response: Thank you very much for the valuable comment. As you said, it is not fair to compare the effectiveness of AuNPs and AgNPs in improving the catalytic activity of CoTPyP. In our case, we just want to demonstrate the versatility of the LSPR-induced enhancement of CoTPyP-catalyzed HER. In addition to AuNPs, silver nanostructures could also be used for improving the catalytic activity of CoTPyP. Therefore, we included the results of AgNPs. We also tried to synthesize the AgNPs with size ~15 nm. However, the ~15 nm AgNPs need to be synthesized in organic solvent, and it easily aggregated when we tried to transfer them to aqueous phase.

Therefore, it was challenging to compare the catalytic activity of AgNP system with ~15 nm diameter. On the other hand, this comparison is out of the focus of our current work. Thus, we prefer not to systematically compare the AuNPs and AgNPs systems in this manuscript. The related description has been changed slightly in the revised manuscript. “The slightly lower HER rate observed here...”(Line 12-13, Page 13)

In terms of AuNR system, the surfactant of CTAB will be difficult to be completely removed, which has been demonstrated by many literatures.¹³ Luckily, in our case, the CoTPyP molecules could be more effectively adsorbed on AuNR surface due to the stronger Au–N bonding compared with the case of CTAB. This information has been confirmed by TEM, EDS mapping, and UV-Vis spectra (Figure R7). In TEM images, the AuNRs aggregation was clearly observed (Figure R7c). Furthermore, the spatial distribution of Co element from CoTPyP overlapped very well with that of Au, demonstrating a strong binding between AuNR and CoTPyP. On the other hand, Br could also be observed in EDS mapping, suggesting that CTAB could not be completely removed. The CoTPyP-induced aggregation was also further confirmed by UV-Vis spectra. As shown in Figure R7b, after adding CoTPyP, the peak at 670 nm redshifted obviously to 678 nm. In addition, this peak obviously broadened and the background at higher wavelength (800-1000 nm) became obviously stronger after adding CoTPyP. These feature changes clearly confirm the aggregation of AuNRs induced by CoTPyP. Similar aggregation of metal nanoparticles has been fully discussed in many literatures.^{9,10}

Figure R7. (a) SEM image of AuNRs. (b) UV–Vis extinction spectrum of AuNRs and AuNR@CoTPyP. (c) STEM image of AuNR@CoTPyP and corresponding EDS element mapping images.

Q4. *The local temperature at the surface of the Au NPs cannot be excluded just by using a cooling system for the bulk solution. It will be helpful to address the reaction at different temperatures without illumination.*

Response: You made a good point. The local temperature could be largely different to that in bulk and it is critically challenging to evaluate the contribution of local temperature. In our case, the temperature of system was controlled by using a constant-temperature water bath, and the reaction suspension was stirred continuously for easier mass transport (Figure R8). Therefore, the temperature of the reaction suspension was kept same during the whole reaction due to the fast heat transfer. The water bath and stirring were used to guarantee a fast heat transfer, minimizing the non-uniform distribution of temperature in the catalytic system. This could be a reasonable approximation, since the power density of our illumination was not very high. In many literatures, researchers also used water bath to minimize the possible contribution from plasmonic heating,^{14,15} which is very similar to what we used here.

As discussed in our manuscript, the excitation of LSPR contributes to the catalytic enhancement of HER. To excite the LSPR and/or photocatalytic HER, a light illumination is necessary. Without light illumination, no HER could happen and of course no hydrogen gas will be produced. As suggested, we performed the reaction at 30, 40, 50, and 60 °C without light illumination, and no hydrogen gas was detected.

Figure R8. Scheme showing the reaction cell for photocatalytic HER.

From Referee #4:

The manuscript by Sheng et al presents a hybrid system with enhanced photocatalytic properties towards HER. I feel that the present version has much improved, I have some concerns regarding the theoretical part.

Response: Thanks a lot for your professional comments. We have tried our best to further improve the quality of our manuscript based on the given comments.

Q1. The first one regards the model they use for their calculations. It is not clear the specific surface of Au they consider, how exposed is this surface in the real sample, how many layers they use for thickness convergence and how big is their supercell in the xy plane (i.e. how their porphyrin coverage matches the experimental one). I therefore ask the authors to describe in detail all these features and justify their choices.

Response: Thank you very much for the valuable suggestion. According to literature,¹⁶ the AuNPs prepared using sodium citrate as the capping agent are surrounded mainly by Au(111) planes. Therefore, the Au(111) was selected for the DFT calculations. On the other hand, the coverage of CoTPyP on AuNP surface in real sample could be estimated based on the ICP results (Figure R9). At low CoTPyP concentration, all CoTPyP molecules are adsorbed on the surface of AuNPs. While at a high CoTPyP concentration, excess CoTPyP molecules will be present in the form of free molecules in solution. The Co:Au atomic ratio in the case of 20 nM CoTPyP was roughly ten times of that in the case of 2 nM CoTPyP, suggesting that all CoTPyP molecules were adsorbed on the surface of AuNPs in these two cases. Otherwise, the Co:Au atomic ratio will be reduced in the case of 20 nM, because the excess CoTPyP molecules will be washed away and give no contribution in ICP results. At the higher CoTPyP concentration of 200 nM, the Co:Au atomic ratio increased to 9.1:1000, which is obviously smaller than ten times of the ratio in 20 nM case, suggesting the existence of unbound CoTPyP molecules, which was washed away *via* the washing process. Based these results, we can conclude that sub-monolayer CoTPyP was adsorbed on gold surface and we can estimate the coverage of CoTPyP molecules on AuNP surface in the case of 2 nM to be ~7%. In the DFT calculation, as shown in Figure R10, the lengths of the x-axis and y-axes of the lattice are 20.18 Å and 19.97 Å, respectively. Thus, the coverage of CoTPyP molecules on gold surface can be estimated to be ~10 %, which is close to that used in typical experiments. The related details have been added in the revised manuscript. “The lengths of the x-axis and y-axes of the lattice were 20.18 Å

and 19.97 Å, respectively.” (Line 8-9, Page 27) “The sub-monolayer CoTPyP molecule was adsorbed on Au(111) surface to obtain a surface coverage of ~10%.” (Line 11-12, Page 27)

Figure R9. Comparison of the Co:Au ratios in the AuNP@CoTPyP samples prepared from different concentrations of CoTPyP obtained from calculation (based on the input CoTPyP solutions) and ICP-OES results.

Figure R10. Schematic illustration of the supercell lattice.

Q2. Porphyrins are large systems where van Der Waals interactions (molecule-molecule and molecule-surface) can be significant. I assume that dispersion forces are not considered in this work. I therefore ask the authors to include dispersion forces in their optimizations and energetic considerations.

Response: Thank you very much for the valuable suggestion. We have considered van der Waals interactions during all the calculations. We set the IVDW=11 in the INCAR file. This information has been added in the description of the computational method. “The DFT-D3 empirical correction method was used to describe van der Waals interactions.” (Line 7-8, Page 27)

Q3. How many atoms have they considered to calculate Hessian/Frequencies needed for the entropic term? Calculating frequencies for the whole supercell is prohibitive with VASP (it uses expensive finite differences method) and the porphyrin is a very large system so the choice is not obvious. Please describe and justify the choice.

Response: Thank you very much for the valuable suggestion. We only considered the vibration of the adsorbent H* without the contribution of catalyst. During the calculation of frequencies, we fixed all the atoms except H* and calculated only the vibration of H*. So, it is affordable to calculate an atom. As suggested, this information has been added in the description of the computational method. “During the calculation of frequencies, we fixed all the atoms except H* and calculated only the vibration of H*.” (Line 18-20, Page 27)

Q4. Calculations here are somehow underused, they mainly describe electronic charge transfer but they could be used to discern between a Tafel Volmer or Volmer Heyrovski mechanism. I ask therefore the authors to dig deeper on the mechanism from the theoretical point of view.

Response: Thank you very much for the valuable suggestion. As suggested, we further explored the Volmer–Heyrovsky and Volmer–Tafel mechanisms of HER process in our system. As shown in Figure R11, by comparing the energy of rate-determining step (RDS) for the Volmer–Heyrovsky and Volmer–Tafel reactions on AuNP@CoTPyP, one can find a considerably lower energy of RDS for the Heyrovsky process (Figure R11a) than that for the Volmer-Tafel process (Figure R11b). This suggests that the Heyrovsky process is much faster, and the Volmer–Heyrovsky mechanism is the main pathway of HER in our system. These results have been included in the revised manuscript and Supporting Information. “We calculated the ΔG for both the Volmer–Heyrovsky and Volmer–Tafel pathways. It was obtained that the rate-determining step (RDS) of Heyrovsky process possesses a lower reaction energy than that of Volmer-Tafel process (Figure S21). Therefore, the Volmer–Heyrovsky mechanism is the main pathway of HER in our system.”(Line 20-23, Page 27 and Figure S21)

Figure R11. Free energy diagram for the (a) Volmer–Heyrovsky route and (b) Volmer–Tafel pathway on AuNP@CoTPyP.

References

1. Tian, J., Zhang, Y., Du, L., He, Y., Jin, X.-H., Pearce, S., *et al.* Tailored self-assembled photocatalytic nanofibres for visible-light-driven hydrogen production. *Nat. Chem.* **12**, 1150-1156 (2020).
2. Banerjee, T., Haase, F., Savasci, G., Gottschling, K., Ochsenfeld, C., Lotsch, B.V. Single-Site Photocatalytic H₂ Evolution from Covalent Organic Frameworks with Molecular Cobaloxime Co-Catalysts. *J. Am. Chem. Soc.* **139**, 16228-16234 (2017).
3. Martindale, B.C.M., Hutton, G.A.M., Caputo, C.A., Reisner, E. Solar Hydrogen Production Using Carbon Quantum Dots and a Molecular Nickel Catalyst. *J. Am. Chem. Soc.* **137**, 6018-6025 (2015).
4. Beyene, B.B., Hung, C.-H. Photocatalytic hydrogen evolution from neutral aqueous solution by a water-soluble cobalt(ii) porphyrin. *Sustain. Energy Fuels* **2**, 2036-2043 (2018).
5. Chakraborty, S., Edwards, E.H., Kandemir, B., Bren, K.L. Photochemical Hydrogen Evolution from Neutral Water with a Cobalt Metallopeptide Catalyst. *Inorg. Chem.* **58**, 16402-16410 (2019).
6. Lazarides, T., Delor, M., Sazanovich, I.V., McCormick, T.M., Georgakaki, I., Charalambidis, G., *et al.* Photocatalytic hydrogen production from a noble metal free system based on a water soluble porphyrin derivative and a cobaloxime catalyst. *Chem. Commun.* **50**, 521-523 (2014).
7. Tritton, D.N., Bodedla, G.B., Tang, G., Zhao, J., Kwan, C.-S., Leung, K.C.-F., *et al.* Iridium motif linked porphyrins for efficient light-driven hydrogen evolution via triplet state stabilization of porphyrin. *J. Mater. Chem. A* **8**, 3005-3010 (2020).
8. Jin, T., Sun, Z., Li, L., Zhang, Q., Zhu, M., Zhang, Z., *et al.* Triboelectric nanogenerator sensors for soft robotics aiming at digital twin applications. *Nat. Commun.* **11**, 5381 (2020).

9. Guo, L., Jackman, J.A., Yang, H.-H., Chen, P., Cho, N.-J., Kim, D.-H. Strategies for enhancing the sensitivity of plasmonic nanosensors. *Nano Today* **10**, 213-239 (2015).
10. Chen, J., Ye, Z., Yang, F., Yin, Y. Plasmonic Nanostructures for Photothermal Conversion. *Small Science* **1**, 2000055 (2021).
11. Wang, P., Guo, S., Wang, H.J., Chen, K.K., Zhang, N., Zhang, Z.M., *et al.* A broadband and strong visible-light-absorbing photosensitizer boosts hydrogen evolution. *Nat. Commun.* **10**, 3155-3157 (2019).
12. Wang, J.-W., Jiang, L., Huang, H.-H., Han, Z., Ouyang, G. Rapid electron transfer via dynamic coordinative interaction boosts quantum efficiency for photocatalytic CO₂ reduction. *Nat. Commun.* **12**, 4276 (2021).
13. Pérez-Juste, J., Pastoriza-Santos, I., Liz-Marzán, L.M., Mulvaney, P. Gold nanorods: Synthesis, characterization and applications. *Coord. Chem. Rev.* **249**, 1870-1901 (2005).
14. Qiu, J., Wei, W.D. Surface Plasmon-Mediated Photothermal Chemistry. *J. Phys. Chem. C* **118**, 20735-20749 (2014).
15. Pan, L., Zhu, Y., Wang, Z., Xu, X., He, H., Du, W., *et al.* Plasmonic Cocatalyst with Electric and Thermal Stimuli Boots Solar Hydrogen Evolution. *Solar RRL* **4**, 2000094 (2020).
16. Yang, T.-H., Shi, Y., Janssen, A., Xia, Y. Surface Capping Agents and Their Roles in Shape-Controlled Synthesis of Colloidal Metal Nanocrystals. *Angew. Chem. Int. Ed.* **59**, 15378-15401 (2020).

List of Changes

1. Based on the comments from reviewers, the determination of TON and TOF has been optimized in the revised manuscript and TON has been added as another y axis in Figure 2c-d. “After optimization, the HER rate and turn-over frequency (TOF) reach 3.21 mol·g⁻¹h⁻¹ and 4650 h⁻¹”(Line 10, Page 2) “... and a high TOF of 4650 h⁻¹ under visible light illumination.”(Line 23, Page 4) “The TOF of our system was determined as 4650 h⁻¹ by using the amount of CoTPyP as the reference.”(Line 14, Page 7) “... and corresponding TON ...”(Line 6-7, Page 8) “... which corresponds to a turnover number (TON) of 13950 each cycle (3 h).”(Line 10, Page 9) “Inductively coupled plasma-optical emission spectroscopy (ICP-OES) was used to obtain the accurate Co: Au atomic ratios after washing (Table S2) for evaluation of the accurate TON values. According to the ICP-OES

analysis, the Co:Au atomic ratio was 1:1600 for the AuNP@CoTPyP structure prepared at the typical CoTPyP concentration of 2 nM. This Co:Au ratio matches perfectly with the one calculated based on the amount of input CoTPyP, since all molecules were bounded on AuNP surface. Therefore, the previously obtained TON values should be accurate.”(Line 13-19, Page 11)

2. The related details have been added in the revised manuscript. “UV–Vis extinction spectra of AuNPs, CoTPyP (50 nM) ...” (Line 5, Page 7)
3. The details on cyclic measurements have been enriched in the revised manuscript. “The cleaned AuNP@CoTPyP collected by centrifugation were used for cyclic measurements.” (Line 7-8, Page 9) “After each cycle, the system was degassed and left in dark for one hour.” (Line 15-16, Page 24)
4. The details on photocurrent tests have been supplied in the revised manuscript (Figure 5b). “In the photocurrent response spectra, photocurrent of the prepared AuNP@CoTPyP under illumination was clearly larger than that of the bare AuNPs, suggesting an improved charge transfer at the interfaces (Figure 5b).” (Line 20-22, Page 17) “Photocurrent measurements of the AuNPs and AuNP@CoTPyP (CoTPyP concentration = 2 nM). The samples were periodically illuminated with green light (550±25 nm filter was applied to Xenon lamp).” (Line 3-5, Page 18) “The photocurrent test was performed on an electrochemical analyzer (CHI 630E, CH Instrument) in a standard three-electrode electrochemical cell filled with 1 M PBS buffer. Carbon paper was used as the working electrode, and 5 mL of AuNP (0.488 mM) was drop-casted, followed by thermal annealing at 80 °C. The annealed sample was then soaked in CoTPyP solution (typically 2 nM) for ten minutes. A piece of Pt plate served as the counter electrode, while Ag/AgCl served as the reference electrode. A green light beam was used for sample excitation by applying a 550±25 nm filter to a Xenon lamp.” (Line 7-14, Page 25)
5. The results on spectroelectrochemical experiments have been added in the revised manuscript and Supporting Information for deeper discussion of transient absorption results. “To verify this species, the spectroelectrochemical experiments were performed in N₂ atmosphere. The transient absorption spectra of CoTPyP match well with the shape of the UV-Vis differential absorption spectra of the reduced CoTPyP and were different from that of the oxidized CoTPyP (Figure S18), suggesting that the reduction quenching pathway should be a dominant process and the new species may be the reduced state of CoTPyP.”(Line 16-21, Page 19)

6. The descriptions and corresponding figure on the reduced CoTPyP species have been added in the revised manuscript and Supporting Information. “Meanwhile, a new negative peak appeared at ~670 nm from 80 ps (Figure 5f) and gradually shifted to ~705 nm from 80 to 1500 ps (Figure 5f and S19). This peak could be attributed to the stimulated emission from the CoTPyP molecules. Note that two peaks showed up at ~665 and ~710 nm in the photoluminescence spectrum of CoTPyP (Figure S20). Due to the different lifetimes of these two photoluminescence events, they showed up in the transient spectra at different time scales, well explaining the observed features in 630-730 nm region.”(Line 25, Page 19 and Line 1-6, Page 20)
7. The legend of Figure 1c has been corrected in the revised manuscript (Page 7).
8. To avoid possible misunderstanding, we have updated the spectra in Figure 2 and Figure S5 by using same batch of samples.
9. The related description on the HER rate of AgNPs has been added in the revised manuscript. “The slightly lower HER rate observed here...”(Line 12-13, Page 13)
10. More details on DFT calculations have been added in the revised manuscript. “The lengths of the x-axis and y-axes of the lattice were 20.18 Å and 19.97 Å, respectively.” (Line 8-9, Page 27) “The sub-monolayer CoTPyP molecule was adsorbed on Au(111) surface to obtain a surface coverage of ~10%.” (Line 11-12, Page 27) “The DFT-D3 empirical correction method was used to describe van der Waals interactions.” (Line 7-8, Page 27) “During the calculation of frequencies, we fixed all the atoms except H* and calculated only the vibration of H*.” (Line 18-20, Page 27)
11. The results on HER mechanism have been included in the revised manuscript and Supplementary Information. “We calculated the ΔG for both the Volmer–Heyrovsky and Volmer–Tafel pathways. It was obtained that the rate-determining step (RDS) of Heyrovsky process possesses a lower reaction energy than that of Volmer–Tafel process (Figure S21). Therefore, the Volmer–Heyrovsky mechanism is the main pathway of HER in our system.”(Line 10-23, Page 27 and Figure S21)

REVIEWER COMMENTS

Reviewer #1 (Remarks to the Author):

The authors have adequately addressed my comments- publication is recommended.

Reviewer #2 (Remarks to the Author):

The authors have really been working very hard on the revision of the manuscript, but I still see here some inconsistencies in the data and discussion of the observed effects. As long as these are not fully resolved or adequately addressed and discussed the manuscript still is not suited for publication.

I still don't understand the argument how from the absorption spectra in figure 3a the conclusion can be derived that aggregation decreases for concentrations larger than 20 nM. The aggregation induced peak is increasing and only for the highest concentration disappears suddenly. This is still not consistent!

For transient absorption excitation at 430 nm was applied. The authors argue that the photoluminescence quenching experiments performed under the same excitation conditions are not valid because this excitation wavelength is not in resonance with the plasmon and that under these condition charge transfer from the porphyrin to the Au particles occurs which leads to the quenching. If this is an issue, this is also valid for the transient absorption experiments, and the transient experiments can not report on what the authors are trying to see in the results.

The results from spectroelectrochemistry show nicely the features of the reduced and oxidized porphyrin. The positive features in the red part of the transient absorption spectra indeed could result from the reduced species but also the just excited porphyrin shows features there: 1) I think from these data with the additional superposition of the signatures by the potential stimulated emission is not suited to distinguish between the reduced porphyrin and the excited porphyrin. Considering the chosen excitation wavelength and the argument concerning the emission quenching, it could be that the positive features are only from the excited porphyrin and the authors observe charge transfer to the Au particle under the chosen excitation conditions. 2) The positive features disappear on a short timescale. If this would be the reduced porphyrin, hence indication of the charge transfer to the porphyrin, then this fast disappearance of the signature would mean either fast charge recombination occurring or

something else, quenching the charge separation, which would be non-beneficial for catalytic activity. Here still the interpretation of the data is not consistent in my opinion, and do not proof the claimed mechanism.

The authors claim that the change in the negative features discussed as stimulated emission is due to different lifetimes of these features. Is there any additional proof for this claim (references from literature or PL lifetime measurements)?

Reviewer #4 (Remarks to the Author):

I think that the authors did a great job on clarifying and adding reaction mechanisms regarding DFT calculations. The only flaw still present regards the frequency calculations: the authors now indicate that they only allow H* to relax, which is formally wrong, because frequencies involve vibration of a covalent bond, so, at least, the adsorbate H* AND the surface atom where H is adsorbed (delivering $3 \times 2 - 5 = 1$ degree of freedom) must be considered in the frequency calculations. I therefore ask the authors to repeat frequency calculations considering those two atoms and report in detail the differences between present calculations with only H*

Response to Reviewers' Comments

From Reviewer #1:

The authors have adequately addressed my comments- publication is recommended.

Response: Thank you very much for the positive comment on our manuscript.

From Reviewer #2:

The authors have really been working very hard on the revision of the manuscript, but I still see here some inconsistencies in the data and discussion of the observed effects. As long as these are not fully resolved or adequately addressed and discussed the manuscript still is not suited for publication.

Response: Thank you very much for the valuable comments and suggestions, based on which we have further improved the quality of our manuscript.

Q1. I still don't understand the argument how from the absorption spectra in figure 3a the conclusion can be derived that aggregation decreases for concentrations larger than 20 nM. The aggregation induced peak is increasing and only for the highest concentration disappears suddenly. This is still not consistent!

Response: Thank you very much for the helpful comment. As discussed, one CoTPyP molecule could be linked to two AuNPs simultaneously (Figure S1 in the Supporting Information). Therefore, at a very low CoTPyP concentration, the CoTPyP molecules will link AuNPs together due to the limited amount of CoTPyP molecules, leading to aggregation of AuNPs (Figure R1). In contrast, at a higher CoTPyP concentration, the surface of each AuNP will be fully adsorbed with many CoTPyP molecules and it is statically difficult for one CoTPyP molecule to simultaneously link two AuNPs. Thus, the aggregation of AuNPs is inhibited. Therefore, increasing the CoTPyP concentration will inhibit the aggregation of AuNPs. In another word, decreasing the CoTPyP concentration will favor the aggregation of AuNPs. Our results, including VU-Vis spectra and TEM images, have clearly demonstrated this theory.

Figure S1. Schematic diagram showing the linking of AuNPs with CoTPyP molecules. The third AuNP (right one) cannot be linked to the CoTPyP molecule (marked green) between first and second AuNPs.

Figure R1. Scheme showing the aggregation of AuNPs at low CoTPyP concentration (left panel) and the inhibition of aggregation at high CoTPyP concentration (right panel).

Generally, new plasmonic band emerges after aggregation of plasmonic nanoparticles and the position and intensity of the newly emerged band are highly related to the degree of aggregation.¹ However, it has to be noted that the aggregation of AuNPs in our cases is quite random and thus the spectroscopic results sometimes may be not straightforward. At the CoTPyP concentration of 2000 nM (Figure 3a), AuNPs cannot aggregate together because the high-concentration CoTPyP molecules could inhibit the aggregation of AuNPs. Decreasing the concentration of CoTPyP led to the aggregation of AuNPs and this aggregation will become more serious at a lower CoTPyP concentration, which has been discussed in main text and Figure 3a. At the CoTPyP concentration of 200 nM, a new peak, except for that at 525 nm, appeared at 630 nm. This new peak red-shifted to 650 nm at a lower CoTPyP concentration of 20 nM. At an ultralow CoTPyP concentration of 2 nM, the aggregation of AuNPs became more serious. This aggregation of AuNPs could be reflected by the extinction spectra (Figure S7a in the Supporting Information). Very broad peaks and strong background showed up at >620 nm, confirming a more serious and random aggregation of AuNPs. Compared with the case of 20 nM CoTPyP, the 2 nM CoTPyP case showed an obvious stronger background at the wavelength >700 nm (Figure R2). Similar aggregation and

spectral changes have been reported in many literatures.^{1,2} In addition, the aggregation was also clearly shown in TEM image (Figure S7b in the Supporting Information). All these results indicate a more serious aggregation of AuNPs at the CoTPyP concentration of 2 nM.

Figure S7. (a) UV–Vis extinction spectrum and photograph of the AuNP@CoTPyP prepared at CoTPyP concentration of 2 nM. (b) TEM image of AuNP@CoTPyP (CoTPyP concentration = 2 nM).

Figure R2. UV–Vis extinction spectra of the AuNP@CoTPyP prepared at CoTPyP concentration of 2 and 20 nM.

Q2. For transient absorption excitation at 430 nm was applied. The authors argue that the photoluminescence quenching experiments performed under the same excitation conditions are not valid because this excitation wavelength is not in resonance with the plasmon and that under these condition charge transfer from the porphyrin to the Au particles occurs which leads to the quenching. If this is an issue, this is also valid for the transient absorption experiments, and the transient experiments cannot report on what the authors are trying to see in the results.

Response: Thank you very much for the valuable comment. You made a very good point. Our original description in previous response was not very accurate. In the PL spectra of AuNP@CoTPyP sample, the CoTPyP molecules were effectively excited by

the applied 430 nm light and the PL signal could be strong (Note that a high concentration of CoTPyP was used for PL measurement to guarantee a detectable signal). While the AuNPs can be excited, although not very effectively, by the same light (confirmed by the considerable absorption of AuNPs at 430 nm, Figure 1c), but the PL signal was too weak possibly due to the ultrashort lifetime of plasmon-generated hot carriers. It is known that the plasmon-generated hot carriers usually recombine very fast³ and thus the PL from AuNPs is very weak. Therefore, we cannot observe the PL from AuNPs (We tried both 430 and 530 nm excitation, and no PL signal was detected in both cases), making it impossible to study the charge transfer from AuNPs to CoTPyP molecules with PL spectra. For instead, we only observed the fluorescence quenching of CoTPyP, which is not the main process during the photocatalysis. We used other evidences to prove that the charge transfer from AuNPs to CoTPyP in the photocatalysis is the main process (discussed in details in response to Q3).

In the transient absorption spectra, the AuNP@CoTPyP sample was excited by a pulsed 430 nm laser (This wavelength was chosen because the CoTPyP, AuNPs, and AuNP@CoTPyP samples can all be excited, favoring the comparison of results), which possess a very high transient power density. As a result, the AuNPs can still be excited even though the excitation wavelength was not optimum (Figure 5c,e). In addition, the transient absorption peak is mainly contributed by the excitation of AuNPs from the ground to excited state. The inverse transition mainly contributes to the decay of the transient absorption peak. Therefore, the weak PL from AuNPs does not affect the peak intensity of the transient absorption peak within a very short period, but reflected by the fast decay of the absorption peak. On the other hand, we used the AuNP@CoTPyP prepared at the CoTPyP concentration of 2 nM, which was optimum for catalytic HER. At this low concentration, the transient absorption signal from CoTPyP was too weak to be observed (We used an elevated concentration for measurement of CoTPyP, Figure 5d). Instead, we observed only the transient absorption signal from AuNPs. The negative peaks at ~530 and ~640 nm in the AuNP@CoTPyP sample originate from the AuNPs and the intensity change of these two peaks follows a similar pattern (Figure 5e). In addition, the left peak and its changes were highly similar to those in AuNPs only (Figure 5c), double confirming this peak is from the AuNPs. Therefore, the negative peaks in the AuNP@CoTPyP system cannot reflect the optical property of the CoTPyP molecules, but mainly the plasmon-induced light absorption. Compared with pristine AuNPs, the AuNP@CoTPyP possesses a shortened lifetime (Figure 5g-h),

suggesting that the plasmon-generated hot carriers possess a shorter lifetime in the AuNP@CoTPyP system. This shortened lifetime is possibly due to the faster transfer and consumption of the hot carriers, and the transferred and consumed hot carriers could be used for catalytic reactions.

Figure 1c. UV-Vis extinction spectra of AuNPs, CoTPyP (50 nM) and AuNP@CoTPyP (CoTPyP concentration = 2 nM).

Figure 5c-h. **c** Ultrafast transient absorption spectra of the AuNP excited by a 430 nm pump beam (pulse density = $17 \mu\text{J}\cdot\text{cm}^{-2}$). **d** Ultrafast transient absorption spectra of the CoTPyP molecules (20 μM) excited by a 430 nm pump beam (pulse density = $90 \mu\text{J}\cdot\text{cm}^{-2}$). **e-f** Ultrafast transient absorption

spectra of AuNP@CoTPyP excited by a 430 nm pump beam (pulse density = $25 \mu\text{J}\cdot\text{cm}^{-2}$). All the transient absorption experiments were performed in water solvent added with 5% methanol. **g-h** Transient absorption decay curves and corresponding fitting of AuNP (at 520 nm) and AuNP@CoTPyP (at 530 nm), respectively.

Q3. *The results from spectroelectrochemistry show nicely the features of the reduced and oxidized porphyrin. The positive features in the red part of the transient absorption spectra indeed could result from the reduced species but also the just excited porphyrin shows features there: 1) I think from these data with the additional superposition of the signatures by the potential stimulated emission is not suited to distinguish between the reduced porphyrin and the excited porphyrin. Considering the chosen excitation wavelength and the argument concerning the emission quenching, it could be that the positive features are only from the excited porphyrin and the authors observe charge transfer to the Au particle under the chosen excitation conditions. 2) The positive features disappear on a short timescale. If this would be the reduced porphyrin, hence indication of the charge transfer to the porphyrin, then this fast disappearance of the signature would mean either fast charge recombination occurring or something else, quenching the charge separation, which would be non-beneficial for catalytic activity. Here still the interpretation of the data is not consistent in my opinion, and do not proof the claimed mechanism.*

Response: Thank you very much for the valuable comment. As you indicated, the charge transfer from porphyrin to AuNPs will be non-beneficial for catalytic activity. However, we observed a significantly enhanced catalysis from the AuNP@CoTPyP structure (Figure 2). This result implies that the quenching of porphyrin is not the main process in our system. We also used the single-molecule fluorescence microscopy to map out the catalysis at a high spatial resolution and the results confirm a significant increase in catalytic activity around the AuNPs (Figure 4c), indicating a plasmon-induced enhancement in catalytic activity. In addition, the catalysis activity was highly related to the wavelength of the excitation light (Figure 4d), and the wavelength dependency of the catalytic activity is in strong consistent with that of the extinction spectrum of AuNPs. The wavelength with strong plasmonic excitation will lead to a higher catalysis activity, indicating a great contribution of plasmonic excitation. In contrast, the wavelength dependency of catalytic activity is very different to that of absorption spectrum of porphyrin, implying the quenching of porphyrin may not be the

main process. Moreover, in broad spectra provided by Xenon lamp, the light absorption of plasmonic AuNPs or AuNP aggregates is much stronger than that of CoTPyP molecules (confirmed by UV–Vis spectra). Therefore, the excitation of AuNPs or AuNP aggregates is dominant and the charge transfer from the AuNP to the adsorbed CoTPyP molecules is the main process.

We also tried to evidence the contribution of plasmon-generated hot carriers in catalysis. First, the EIS spectra suggest a favored charge transfer in the AuNP@CoTPyP system (Figure 5a). Second, the photocurrent results suggest a favored charge transfer and a charge flow from AuNPs to CoTPyP molecules in the AuNP@CoTPyP system (Figure 5b & R3). Third, the DFT calculation results indicate that the charge transfer from the Co atom to H* increases from 0.001 e to 0.013 e when the CoTPyP molecule was adsorbed on gold surface (Figure 6b), suggesting that the charge transfers from the AuNP to adsorbed CoTPyP molecule and then H*. Fourth, the transient absorption spectra show that the lifetime of plasmon-generated hot carriers decreases after the adsorption of CoTPyP molecules (Figure 5g-h). This decrease in lifetime suggests a faster transfer and consumption of the hot carriers, and the transferred and consumed hot carriers may be used for catalytic reactions. All these results suggest that, under plasmonic excitation, the plasmon-generated hot carriers transfer to the Co center in CoTPyP molecule, favoring the CoTPyP-catalyzed HER.

In terms of the positive features in transient absorption spectrum of CoTPyP, the transient absorption spectra of CoTPyP match well with the shape of the UV–Vis differential absorption spectra of the reduced CoTPyP and were different from that of the oxidized CoTPyP (Figure S18 in the Supporting Information), suggesting that the new species may be the reduced state of CoTPyP.^{4,5} Therefore, the CoTPyP-based photocatalysis follows a reductive quenching pathway, which is consistent with the reported work.⁶

Figure 5a. Nyquist plots of CoTPyP and AuNP@CoTPyP in H₂SO₄ solution (pH = 4).

Figure 5b & R3. **5b** Photocurrent measurements of the AuNPs and AuNP@CoTPyP (CoTPyP concentration = 2 nM). The samples were periodically illuminated with green light (550 ± 25 nm filter was applied to Xenon lamp). **R3** Schematic illustration of the photocurrent measurements for AuNP@CoTPyP. The increase in photocurrent indicates a promoted charge flow from AuNPs to the adsorbed CoTPyP molecules.

Figure 6b. Differential charge densities of H* at CoTPyP and AuNP@CoTPyP.

Figure 5g-h. Transient absorption decay curves and corresponding fitting of AuNP (at 520 nm) and AuNP@CoTPyP (at 530 nm), respectively.

Q4. The authors claim that the change in the negative features discussed as stimulated emission is due to different lifetimes of these features. Is there any additional proof for this claim (references from literature or PL lifetime measurements)?

Response: Thank you very much for the valuable comment. It is common for some molecules to have two different lifetimes. It has been reported that some molecules,

such as porphyrins (Figure R4), may have two different lifetimes.⁷⁻⁹ The two lifetimes were attributed to the emission at different wavelengths.

Figure R4. Time-resolved fluorescence decay kinetics of photosensitizing porphyrin (*J. Photochem. Photobiol. B, Biol.* **1993**, *21*, 143-147).

From Reviewer #4:

Q1. I think that the authors did a great job on clarifying and adding reaction mechanisms regarding DFT calculations. The only flaw still present regards the frequency calculations: the authors now indicate that they only allow H to relax, which is formally wrong, because frequencies involve vibration of a covalent bond, so, at least, the adsorbate H* AND the surface atom where H is adsorbed (delivering 3*2-5=1 degree of freedom) must be considered in the frequency calculations. I therefore ask the authors to repeat frequency calculations considering those two atoms and report in detail the differences between present calculations with only H**

Response: We thank the reviewer for the valuable comments, which largely improve the quality of our manuscript. As suggested, we recalculated the frequency of the intermediates considering the adsorbate *H, 2H*, and the Co atoms where intermediates are adsorbed. Based on the updated frequency results, we obtained new ZPE-TS (*H) and ZPE-TS (2H*) values, which are very close to those obtained from the original calculations (Table R1). The related description and Figure S21 have been updated in the revised manuscript. **“For the frequency calculation, we considered the**

adsorbate *H, 2H*, and the Co atoms where intermediates are adsorbed.” (Lines 18-20, Page 27)

Table R1. Results of ZPE-TS (*H) and ZPE-TS (2H*) from original calculation and the repeated calculation.

	ZPE-TS (*H)	ZPE-TS (2H*)
Original Calculation	0.19	0.42
Repeated Calculation	0.19	0.41

Figure S21. Free energy diagram for the (a) Volmer–Heyrovsky route and (b) Volmer–Tafel pathway on AuNP@CoTPyP.

References

1. Guo, L., Jackman, J.A., Yang, H.-H., Chen, P., Cho, N.-J., Kim, D.-H. Strategies for enhancing the sensitivity of plasmonic nanosensors. *Nano Today* **10**, 213-239 (2015).
2. Chen, J., Ye, Z., Yang, F., Yin, Y. Plasmonic Nanostructures for Photothermal Conversion. *Small Science* **1**, 2000055 (2021).
3. Zhang, C., Jia, F., Li, Z., Huang, X., Lu, G. Plasmon-generated hot holes for chemical reactions. *Nano Res.* **13**, 3183-3197 (2020).
4. Wang, J.-W., Jiang, L., Huang, H.-H., Han, Z., Ouyang, G. Rapid electron transfer via dynamic coordinative interaction boosts quantum efficiency for photocatalytic CO₂ reduction. *Nat. Commun.* **12**, 4276 (2021).
5. Wang, P., Guo, S., Wang, H.J., Chen, K.K., Zhang, N., Zhang, Z.M., *et al.* A broadband and strong visible-light-absorbing photosensitizer boosts hydrogen evolution. *Nat. Commun.* **10**, 3155-3157 (2019).
6. Wu, C., Jung, K., Ma, Y., Liu, W., Boyer, C. Unravelling an oxygen-mediated reductive quenching pathway for photopolymerisation under long wavelengths. *Nat. Commun.* **12**, 478 (2021).

7. Zheng, W., Shan, N., Yu, L., Wang, X. UV-visible, fluorescence and EPR properties of porphyrins and metalloporphyrins. *Dyes Pigm.* **77**, 153-157 (2008).
8. Niu, J.-X., Pan, C.-D., Liu, Y.-T., Lou, S.-T., Wu, E., Wu, B.-T., *et al.* Plasmon-enhanced fluorescence of submonolayer porphyrins by silver-polymer core-shell nanoparticles. *Opt. Express* **26**, 3489-3496 (2018).
9. Kumar, P.R., Britto, N.J., Kathiravan, A., Neels, A., Jaccob, M., Mothi, E.M. Synthesis and electronic properties of A3B-thienyl porphyrins: experimental and computational investigations. *New J. Chem.* **43**, 1569-1580 (2019).

List of Changes

1. To avoid possible misunderstanding, a new sentence has been added in the revised manuscript to describe the charge transfer direction, “**The charge flow from AuNPs to adsorbed CoTPyP molecules under illumination could be determined based on the configuration of the measurement setup.**” (bottom part on Page 17)
2. The concentration of bare CoTPyP solution (**20 μ M**) for transient absorption measurement was provided in the caption to Figure 5d.
3. We redid the frequency calculations and updated the ΔG results in Figure S21. As well the description related to frequency calculations has been revised to “**For the frequency calculation, we considered the adsorbate *H, 2H*, and the Co atoms where intermediates are adsorbed**” in the revised manuscript. (bottom part on Page 27)

REVIEWERS' COMMENTS

Reviewer #2 (Remarks to the Author):

I am happy with all changes and explanations and the manuscript can now be published as is.